# Leveraging Offline Data in Linear Latent Contextual Bandits

**Chinmaya Kausik** [1]   **Kevin Tan** [2]   **Ambuj Tewari** [1]

## Abstract

Leveraging offline data is an attractive way to accelerate online sequential decision-making. However, it is crucial to account for latent states in users or environments in the offline data, and latent bandits form a compelling model for doing so. In this light, we design end-to-end latent bandit algorithms capable of handing uncountably many latent states. We focus on a linear latent contextual bandit – a linear bandit where each user has its own high-dimensional reward parameter in $\mathbb{R}^{d_A}$, but reward parameters across users lie in a low-rank latent subspace of dimension $d_K \ll d_A$. First, we provide an offline algorithm to learn this subspace with provable guarantees. We then present two online algorithms that utilize the output of this offline algorithm to accelerate online learning. The first enjoys $\tilde{O}(\min(d_A\sqrt{T}, d_K\sqrt{T}(1 + \sqrt{d_A T/d_K N})))$ regret guarantees, so that the effective dimension is lower when the size $N$ of the offline dataset is larger. We prove a matching lower bound on regret, showing that our algorithm is minimax optimal up to coverage terms. The second is a practical algorithm that enjoys only a slightly weaker guarantee, but is computationally efficient. We also establish the efficacy of our methods using experiments on both synthetic data and real-life movie recommendation data from MovieLens. Finally, we theoretically establish the generality of the latent bandit model by proving a de Finetti theorem for stateless decision processes.

## 1. Introduction

Many sequential-decision making problems can be effectively modeled using the bandit framework. This can span domains as diverse as healthcare (Lu et al., 2021b), randomized clinical trials (Press, 2009), search and recommendation (Li et al., 2010), distributed networks (Kar et al., 2011), and portfolio design (Brochu et al., 2011). There is often a wealth of offline data in such domains, which has led to a growing interest in using offline data to accelerate online learning. However, there often also exist unobserved contexts in the population that influence the distribution of rewards, making it non-trivial to leverage offline data. In Hong et al. (2020), it is shown that this uncertainty can be modeled by a *latent bandit* (or mixture of bandits). This is a bandit where an unobserved latent state determines the reward model for the trajectory. For example, a patient's underlying genetic conditions in healthcare and a user's tastes in recommendation systems are both examples of latent states in sequential decision making. Typically, these latent states are less complex than the actual models underlying users or patients, making it valuable to reduce the online task to learning the latent state (Hong et al., 2020; 2022).

The latent bandit framework therefore has high practical value, and efficient principled algorithms are needed for using offline data to speed up online learning. Using traditional bandit algorithms in this setting does not leverage the offline data that is often available to the agent. Naturally, one also cannot treat the offline data as coming from a single bandit. For example, different user tastes or different underlying genetic conditions require modeling the offline data as coming from a latent bandit. So, we have to develop algorithms *specific* to the latent bandit setting that leverage the offline data to improve online performance.

We note that in bandit literature, it is common to impose a structure on bandit rewards when designing algorithms, the most popular one being a linear structure (Li et al., 2010; Abbasi-Yadkori et al., 2011). In this light, we study a linear contextual bandit setting where each user has its own high-dimensional reward parameter, but reward parameters across users lie in a low-rank subspace. This is a *linear latent contextual bandit*[1], and is much more general than existing models that restrict themselves to finitely many latent states (Hong et al., 2020; 2022). We design a two-pronged algorithm to tackle this setting. First, we provide a method to approximate the low-dimensional subspace spanned by latent states from an offline dataset of unlabeled trajectories collected under some behavior policy $\pi_b$. This is non-trivial since the trajectories are unlabeled, and standard

---

[1]Department of Statistics, University of Michigan, USA [2]Department of Statistics and Data Science, The Wharton School, University of Pennsylvania, USA. Correspondence to: Chinmaya Kausik <ckausik@umich.edu>.

*Proceedings of the 42^{nd} International Conference on Machine Learning*, Vancouver, Canada. PMLR 267, 2025. Copyright 2025 by the author(s).

---

[1]This can also be thought of as a continuous mixture of bandits.

unsupervised learning methods fail. Second, we use this subspace to speed up online learning. However, since the subspace is only learnt *approximately,* we also tackle the non-trivial task of accounting for the uncertainty in the subspace. We design two methods for the latter, facing a trade-off between computational tractability and tightness of guarantees. Experiments show the efficacy of our methods.

While latent bandits have thus shown to be a powerful and tractable framework for accounting for uncertainty in reward models, the extent of their *generality* is unclear. Are there other stateless decision processes that generalize over latent bandits? We end by theoretically demonstrating that under very reasonable assumptions, the answer is no. We show a de Finetti theorem for decision processes, demonstrating that *every* "coherent" and "exchangeable" stateless (contextual) decision process is a latent (contextual) bandit. With this in mind, we outline our contributions below:

- **Offline method:** We present SOLD, a novel offline method for learning low-dimensional subspaces of reward parameters with guarantees, inspired by the novel spectral methods in Kausik et al. (2023).

- **Tight online algorithm:** We present LOCAL-UCB, an online algorithm leveraging the subspace estimated offline to sharpen optimism, achieving $\tilde{O}(\min(d_A\sqrt{T}, d_K\sqrt{T}(1 + \sqrt{d_A T/d_K N})))$ regret.

- **Lower Bound:** We establish a matching lower bound showing that LOCAL-UCB is minimax optimal. To the best of our knowledge, this is the first lower bound in a hybrid (offline-online) sequential decision-making setting.

- **Tractable online algorithm:** Finally, we present ProBALL-UCB, a practical and computationally efficient online algorithm with a slightly looser regret guarantee. This also illustrates a general algorithmic idea for integrating offline subspace estimation into optimistic algorithms.

- **Experiments:** We establish the efficacy of our algorithms outlined above through a simulation study and a demonstration on a real recommendation problem with the MovieLens-1M (Harper and Konstan, 2015) dataset.

- **Theoretical generality:** We are the first, to our knowledge, to prove a de Finetti theorem for decision processes. This establishes the generality of the latent bandit model.

**Related work.** There are three main threads of related work.

- **Latent Bandits.** The line of work most relevant to us has been on latent bandits. The work of Hong et al. (2020; 2022) studies the latent bandit problem under finitely many states. However, they black-box the offline step and do not provide end-to-end guarantees, and their ideas do not extend to infinitely many states. Our work seeks to provide end-to-end guarantees for both the offline and online component under infinitely many latent states.

- **Meta learning, multi-task learning and mixture learning.** A long line of work studies learning with multiple underlying tasks or models. For example, the work of Vempala and Wang (2004); Kong et al. (2020); Anandkumar et al. (2014); Tripuraneni et al. (2022) study learning under latent variable or multi-task models in a supervised setting. The work of Kausik et al. (2023); Chen and Poor (2022) extend some of these ideas to unsupervised but purely offline learning in a time-series setting. On the other hand, work like Yang et al. (2022); Cella et al. (2022) instead focuses on the purely online setting of learning the low-rank structure while simultaneously interacting with multiple finitely many bandit instances. Finally, Zhou et al. (2024); Lu et al. (2021a) work with multiple underlying models in MDPs but in a purely offline and generative model setting respectively. Unlike these papers, we crucially combine the offline and online settings for sequential data and study the problem of using offline data to accelerate online learning in bandits.

- **Hybrid (offline-online) RL.** Work in hybrid RL studies the use of offline data to accelerate online RL, first proposed by Song et al. (2023), with extensions to linear MDPs by Wagenmaker and Pacchiano (2023); Tan et al. (2024). Cai et al. (2024) studies the same problem for contextual bandits. However, all work so far assumes that the offline data is generated by a single model, and does not account for latent states. Our work explores a hybrid offline-online setting while also accounting for the offline data being generated by multiple underlying models.

## 2. Linear Bandits With Latent Structure

We will first introduce latent contextual bandits, and then specialize to our linear model later in this section. A latent contextual bandit is a decision process with contexts $\mathcal{X}$, actions $\mathcal{A}$ and an random latent state $\theta$ that is sampled independently at the beginning of each trajectory. Given a sequence of contexts and actions, the rewards at all steps are independent *conditioned* on $\theta$, and depend on the latent state $\theta$, the context and the action. Since we often have access to an offline dataset of trajectories coming from different kinds of users or patients, it is important to account for a changing latent state $\theta$ between trajectories.

As we will see in Section 8, latent bandits are a powerful and general framework for encoding uncertainty in reward models. However, this generality is both a blessing and a curse. It is hard to design concrete algorithms without further assumptions on the structure and effect of the latent state $\theta$. We therefore focus on a linear structure here. We consider the natural generalization of the linear contextual bandit to the latent bandit setting, where we impose a linear structure on the effect of low-dimensional latent states. We further justify this by noting that in most application domains, it is reasonable to assume that a parsimonious,

low-dimensional latent state affects the reward distribution. This motivates the following definition.

**Definition 1.** A *linear latent contextual bandit* is a linear bandit equipped with a feature map for context-action pairs $\phi : \mathcal{X} \times \mathcal{A} \rightarrow \mathbb{R}^{d_A}$, a latent random variable $\boldsymbol{\theta} \in \mathbb{R}^{d_K}$ with distribution $\mathcal{D}_\theta$ and a map $\mathbf{U}_\star : \mathbb{R}^{d_K} \rightarrow \mathbb{R}^{d_A}$ such that for any $H$ and context-action sequence $((x_1, a_1), \ldots (x_H, a_H))$, the rewards $(Y_1, \ldots Y_H)$ are independent conditioned on $\boldsymbol{\theta}$. Moreover, $Y_h \mid \boldsymbol{\theta} \sim \phi(x_h, a_h)^\top \boldsymbol{\beta} + \epsilon$, where $\epsilon$ is subgaussian noise independent of all actions and all other observations, and $\boldsymbol{\beta} = \mathbf{U}_\star \boldsymbol{\theta}$.

Further, we note that WLOG $\mathbf{U}_\star$ has orthonormal columns: $\mathbf{U}_\star^\top \mathbf{U}_\star = \mathbf{I}_{d_A}$. This is because for any invertible map $A : \mathbb{R}^{d_K} \rightarrow \mathbb{R}^{d_K}$, the observation distribution does not change upon replacing $\boldsymbol{\theta}$ with $A\boldsymbol{\theta}$ and $\mathbf{U}_\star$ with $\mathbf{U}_\star A^{-1}$. One can see this as a generalization of the fact that with finitely many latent states, the observations are not changed by permuting the latent states. That is, the observations are not changed by permuting latent trajectory labels while keeping trajectories with the same label together.

Let us now assume that we have access to a dataset $\mathcal{D}_{\text{off}}$ of $N$ *short* trajectories $\tau_n = ((x_{n,1}, a_{n,1}, r_{n,1}), \ldots, (x_{n,H}, a_{n,H}, r_{n,1}))$ of length $H$, collected by some behavior policy $\pi_b$. The trajectories are short in the sense that in most relevant domains, individual trajectories are not long enough to learn the underlying reward model. Each trajectory $\tau_n$ has a different $\boldsymbol{\beta}_n = \mathbf{U}_\star \boldsymbol{\theta}_n$. In online deployment, a single latent label $\boldsymbol{\theta}_\star$ is chosen and rewards are generated using $\boldsymbol{\beta}_\star = \mathbf{U}_\star \boldsymbol{\theta}_\star$. At each timestep $t$, an agent observes contexts $x_t$ and uses both the offline data and the online data at time $t$ to execute a policy $\pi_t$. Define the optimal action at time $t$ by $a_t^\star := \max_a \phi(x_t, a_t)^\top \boldsymbol{\beta}_\star$ We tackle the problem of minimizing the *frequentist* regret in linear latent contextual bandits, given by

$$\text{Reg}_T := \sum_{t=1}^{T} \phi(x_t, a_t^\star)^\top \boldsymbol{\beta}_\star - \mathbb{E}_{a \sim \pi_t}[\phi(x_t, a)^\top \boldsymbol{\beta}_\star].$$

For example, in medical applications, data from short randomized controlled trials can be used to help an agent suggest treatment decisions for a new patient online. In this case, we would like the algorithm to administer the correct treatments for *each* patient. This means that the *frequentist* regret is the relevant performance metric here, and not the Bayesian regret over some prior. Additionally, any worst-case bound on the frequentist regret is a bound on the Bayesian regret for arbitrary priors.

**Challenges with latent bandits.** Despite the linear assumption, and the dimension reduction obtained in the common case when $d_K \ll d_A$, significant challenges remain. First, the value of the latent state $\boldsymbol{\theta}$ and the map $\mathbf{U}_\star$ are both unknown a priori, making it hard to leverage the low-dimensional structure of the problem. Second, a good choice of dimension $d_K$ is itself unknown a priori, and must be determined from data in a principled manner. Third, even if we learn the low-dimensional structure, our learning will be *approximate*, and the online procedure must account for this uncertainty. In the following sections, we will provide a method to estimate and use latent subspaces given offline data that allows us to overcome these challenges.

**Additional Notation.** We use $\mathbf{V}$ to denote regularized design matrices given by $\mu\mathbf{I} + \sum_{(x,a)} \phi(x, a)\phi(x, a)^\top$. We define $\mathbf{D}_{n,i} = I - \mu\mathbf{V}_{n,i}^{-1}$ and $\overline{\mathbf{D}}_{N,i} = \frac{1}{N}\sum_{n=1}^{N} \mathbf{D}_{n,i}$. Denote by $\hat{\beta}_{n,1}, \hat{\beta}_{n,2}$ independent estimates of $\beta_n$ from $\tau_n$. Let $\mathbf{M}_n = \frac{1}{2}(\hat{\beta}_{n,1}\hat{\beta}_{n,2}^\top + \hat{\beta}_{n,2}\hat{\beta}_{n,1}^\top)$ and $\overline{\mathbf{M}}_N \leftarrow \frac{1}{N}\sum_{n=1}^{N} \mathbf{M}_n$.

## 3. Estimating Latent Subspaces Offline

Although we do not have access to the values of the latent states $\boldsymbol{\theta}$ or to the map $\mathbf{U}_\star$, we can still extract useful information from data. To that effect, recall that we have access to a dataset $\mathcal{D}_{\text{off}}$ of $N$ trajectories $\tau_n = ((x_{n,h}, a_{n,h}, r_{n,h}))_{h=1}^{H}$ of length $H$, collected by some behavior policy $\pi_b$.

**How can offline data help us in online deployment?** To minimize the regret, one must learn the reward parameter $\boldsymbol{\beta}_\star$ online. However, it is much easier to search among all latent states $\boldsymbol{\theta} \in \mathbb{R}^{d_K}$ than to search among all possible reward parameters $\boldsymbol{\beta} \in \mathbb{R}^{d_A}$ since typically, $d_K \ll d_A$. So, it will help to learn some projection matrix $\hat{\mathbf{U}}^\top \approx \mathbf{U}_\star^\top : \mathbb{R}^{d_A} \rightarrow \mathbb{R}^{d_K}$ offline so that for any estimate $\hat{\boldsymbol{\beta}}_t$ of $\boldsymbol{\beta}_\star$, $\hat{\mathbf{U}}^\top \hat{\boldsymbol{\beta}}_t$ is an estimate of $\hat{\boldsymbol{\theta}}_t \in \mathbb{R}^{d_K}$. This amounts to *learning a subspace* of the feature space from logged bandit data. We therefore provide a method for Subspace estimation from Offline Latent bandit Data (SOLD) in Algorithm 1. Recall that since the learnt subspace is approximate, we also need to compute the uncertainty over the subspace to get a subspace confidence set that we can use online.

**On trajectory splitting and corrections.** To extract the $d_K$-dimensional subspace, we aim to estimate $\mathbb{E}[\boldsymbol{\beta}\boldsymbol{\beta}^\top] = \mathbf{U}_\star \mathbb{E}[\boldsymbol{\theta}\boldsymbol{\theta}^\top]\mathbf{U}_\star^\top$, which has the same $d_K$-dimensional span as $\mathbf{U}_\star$. This cannot be achieved by using a single estimator $\hat{\boldsymbol{\beta}}_n$ for each trajectory $\tau_n$ and averaging the outer products $\hat{\boldsymbol{\beta}}_n\hat{\boldsymbol{\beta}}_n^\top$ across all $n$. That is because the per-reward noise $\epsilon$ will be shared by both copies of $\hat{\boldsymbol{\beta}}_n$, and so the variance of $\epsilon$ will make $\mathbb{E}[\hat{\boldsymbol{\beta}}_n\hat{\boldsymbol{\beta}}_n^\top]$ full rank. We therefore split each trajectory $\tau_n$ to obtain two independent estimators $\hat{\boldsymbol{\beta}}_{n,1}, \hat{\boldsymbol{\beta}}_{n,2}$, compute the outer products $\hat{\boldsymbol{\beta}}_{n,1}^\top\hat{\boldsymbol{\beta}}_{n,2}$, and obtain the top $d_K$ eigenvectors of the mean outer product across trajectories.

However, there is a further wrinkle here. We cannot simply take the top $d_K$ eigenvectors of the mean outer product $\overline{\mathbf{M}}_N$. One can compute that $\mathbb{E}[\overline{\mathbf{M}}_N] = \mathbb{E}[\mathbf{D}_{n,1}\boldsymbol{\beta}_n\boldsymbol{\beta}_n^\top\mathbf{D}_{n,2}] =$

**Algorithm 1** Subspace estimation from Offline Latent bandit Data (SOLD)

1: **Input:** Dataset $\mathcal{D}_{\text{off}}$ of collected trajectories $\tau_n = ((x_{n,1}, a_{n,1}, r_{n,1}), ..., (x_{n,H}, a_{n,H}, r_{n,1}))$ under a behavior policy $\pi_b$, dimension of latent subspace $d_K$.

2: **Divide** each $\tau_n$ into odd and even steps, giving trajectory halves $\tau_{n,1}$ and $\tau_{n,2}$.

3: **Estimate** reward parameters $\hat{\beta}_{n,i} \leftarrow \mathbf{V}_{n,i}^{-1}\mathbf{b}_{n,i}$, where $\mathbf{V}_{n,i} \leftarrow \mu\mathbf{I} + \sum_{(x,a,r)\in\tau_{n,i}} \phi(x,a)\phi(x,a)^\top$ and $\mathbf{b}_{n,i} \leftarrow \sum_{(x,a,r)\in\tau_{n,i}} \phi(x,a)r$ for $i = 1, 2$.

4: **Compute** $\mathbf{M}_n \leftarrow \frac{1}{2}(\hat{\beta}_{n,1}\hat{\beta}_{n,2}^\top + \hat{\beta}_{n,2}\hat{\beta}_{n,1}^\top)$ and compute $\overline{\mathbf{M}}_N \leftarrow \frac{1}{N}\sum_{n=1}^N \mathbf{M}_n$.

5: **Compute** $\overline{\mathbf{D}}_{N,i} \leftarrow \frac{1}{N}\sum_{n=1}^N (I - \mu\mathbf{V}_{n,i}^{-1})$, $i = 1, 2$.

6: **Obtain** $\hat{\mathbf{U}}$, the top $d_K$ eigenvectors of $\overline{\mathbf{D}}_{N,1}^{-1}\overline{\mathbf{M}}_N\overline{\mathbf{D}}_{N,2}^{-1}$.

7: **return** Projection matrix $\hat{\mathbf{U}}\hat{\mathbf{U}}^\top$, $\Delta_{\text{off}}$ as in Theorem 1

$\mathbb{E}[\mathbf{D}_{n,1}\mathbf{U}_\star\theta_n\theta_n^\top\mathbf{U}_\star^\top\mathbf{D}_{n,2}]$. To separate $\mathbb{E}[\beta_n\beta_n^\top]$ from this, we need $\mathbf{D}_{n,1}, \beta_n, \mathbf{D}_{n,2}$ to be independent. If $\pi_b$ does not use $\theta$ and contexts are generated independently of each other and of $\theta$, then this is satisfied. Intuitively, we need the offline trajectories to be non-adaptive. In fact, we show in the lemma below that if any of these three conditions is violated, then it is in fact impossible to determine the latent subspace $\mathbf{U}^\star$ using *any* method, even with infinitely many infinitely long trajectories.

**Lemma 1** (Contexts, $\theta$, and $\pi_b$ cannot be dependent). *For each of these conditions:*

*1. Contexts in a trajectory are dependent but do not depend on $\theta$, and $\pi_b$ also does not use $\theta$,*

*2. Contexts are generated independently using $\theta$, while $\pi_b$ does not use $\theta$,*

*3. Contexts are generated independently without using $\theta$, while $\pi_b$ uses $\theta$,*

*there exist two different linear latent contextual bandits with orthogonal latent subspaces satisfying the condition, and a behavior policy $\pi_b$ so that the offline data distributions are indistinguishable and cover all $(x, a)$ pairs with probability at least $1/4$. Since the latent subspaces are orthogonal, an action that gives the maximum reward on one latent bandit gives reward $0$ on the other.*

To estimate the latent subspace, one is thus forced to make the following assumption.

**Assumption 1** (Unconfounded Offline Actions). The offline behavior policy $\pi_b$ does not use $\theta$ to choose actions, and contexts $x_{n,h}$ are stochastic and generated independently of each other and of $\theta$.

This is satisfied when the offline data comes from random-

ized controlled trials or A/B testing, which are common sources of offline datasets. Even if this is not satisfied, Algorithm 1 can learn a good subspace whenever $\overline{\mathbf{D}}_{N,1}^{-1}\overline{\mathbf{M}}_N\overline{\mathbf{D}}_{N,2}^{-1}$ has eigenspace close to the span of $\mathbf{U}_\star$, e.g. when high-reward actions contribute heavily to $\mathbf{D}_{n,i}$. This can happen if offline trajectories were collected to maximize rewards.

Returning to our scrutiny of $\overline{\mathbf{M}}_N$, let the covariance matrix of $\theta$ be $\Lambda$ and let its mean be $\mu_\theta$. Then we have that $\mathbb{E}[\overline{\mathbf{M}}_N] = \mathbb{E}[\mathbf{D}_{n,1}\beta_n\beta_n^\top\mathbf{D}_{n,2}] = \mathbb{E}[\mathbf{D}_{n,1}]\mathbf{U}_\star(\Lambda + \mu_\theta\mu_\theta^\top)\mathbf{U}_\star^\top\mathbb{E}[\mathbf{D}_{n,2}]$. So, we still cannot merely consider the top $d_K$ eigenvectors of $\overline{\mathbf{M}}_N$ without accounting for $\mathbf{D}_{n,1}$. Intuitively, $\mathbf{D}_{n,1}$ captures the distortion in reward estimation caused by regularization in ridge regression[2]. We therefore construct correction matrices $\overline{\mathbf{D}}_{N,i}$ and use them to "neutralize" the distortion from regularization. In particular, $\overline{\mathbf{D}}_{N,1}^{-1}\overline{\mathbf{M}}_N\overline{\mathbf{D}}_{N,2}^{-1}$ is an estimator for $\mathbf{U}_\star(\Lambda + \mu_\theta\mu_\theta^\top)\mathbf{U}_\star^\top$. Crucially, this allows us to aggregate information across many trajectories to overcome the challenge of learning from short trajectories. We can now take the top $d_K$ eigenvectors of $\overline{\mathbf{D}}_{N,1}^{-1}\overline{\mathbf{M}}_N\overline{\mathbf{D}}_{N,2}^{-1}$ to estimate the subspace determined by $\mathbf{U}_\star$. To give guarantees, we must make a coverage assumption. Unlike in standard offline RL, where only coverage along actions is needed, we also need coverage along latent states.

**Assumption 2** (Boundedness and Coverage). Rewards $|r_{n,h}| \leq R$ [3] for all $n, h$, $\|\phi(x,a)\|_2 \leq 1$ and $\|\beta\|_2 \leq R$. Also, $\lambda_A := \min_{i=1,2} \lambda_{\min}(\mathbb{E}[\mathbf{D}_{n,i}]) > 0$ and $\lambda_\theta := \frac{1}{R^2}\lambda_{\min}(\Lambda) > 0$.

Intuitively, $\lambda_A$ measures coverage along actions, while $\lambda_\theta$ measures coverage along latent states $\theta$. Both must be non-zero to expect satisfactory estimation of the subspace. Unlike the setting of Yang et al. (2022), whose setting is purely online, we work with an offline dataset of trajectories spanning multiple bandit instances. The learner has no control over the behavior policy that collected the data. Without structural assumptions on the dataset, estimating a useful subspace becomes infeasible, and the regret degenerates to the standard $d_A\sqrt{T}$. Similar coverage assumptions are commonplace within the offline linear MDP literature (Jin et al., 2021; Duan and Wang, 2020).

We can then use confidence bounds for $\overline{\mathbf{M}}_N$ and $\overline{\mathbf{D}}_{N,i}$ to give a data-dependent confidence bound $\Delta_{\text{off}}$ for the projection matrix $\hat{\mathbf{U}}\hat{\mathbf{U}}^\top$, as in Theorem 1 below. In one instantiation, Propositions 1 and 2 in Appendix C derive simple data-dependent bounds for $\overline{\mathbf{M}}_N$ and $\overline{\mathbf{D}}_{N,i}$ respectively. Under this choice, we control the growth of $\Delta_{\text{off}}$ in terms of the unknown problem parameters at the end of Theorem 1.

---

[2]This is not unique to regularization. Pseudo-inverses cause an analogous problem of distortion caused by unseen actions.

[3]$R$-bounded rewards are automatically $R$-subgaussian. We can easily extend our results to more general subgaussian rewards, but stick to bounded rewards for simplicity of proofs.

**Theorem 1** (Computing and Bounding $\Delta_{\text{off}}$). *Let* $\|\overline{\boldsymbol{M}}_N - \mathbb{E}[\boldsymbol{M}_1]\|_2 \leq \Delta_M$ *and* $\|\overline{\boldsymbol{D}}_{N,i} - \mathbb{E}[\boldsymbol{D}_{n,i}]\|_2 \leq \Delta_D$ *for* $i = 1, 2$ *with probability* $1 - \delta/3$ *each. Then, with probability* $1 - \delta$, $\|\hat{\boldsymbol{U}}\hat{\boldsymbol{U}}^\top - \boldsymbol{U}_\star \boldsymbol{U}_\star^\top\|_2 \leq \Delta_{\text{off}}$, *where for* $B_D = \|\overline{\boldsymbol{D}}_N^{-1}\|_2$ *and* $\hat{\lambda} := \lambda_{d_K}(\overline{\boldsymbol{M}}_N) - \lambda_{d_K+1}(\overline{\boldsymbol{M}}_N)$,

$$
\Delta_{\text{off}} = \frac{2\sqrt{2d_K}}{\hat{\lambda}} \left( \frac{B_D^3(2 - B_D\Delta_D)}{(1 - B_D\Delta_D)^2}(R^2 + \Delta_M)\Delta_D \right.
$$
$$
\left. + \left( \frac{B_D}{1 - B_D\Delta_D} \right)^2 \Delta_M \right).
$$

*Obtaining* $\Delta_M$ *and* $\Delta_D$ *from Propositions 1 and 2,* $\Delta_{\text{off}} = \widetilde{O}(\frac{1}{\lambda_\theta \lambda_A^3} N^{-1/2}\sqrt{d_K d_A \log(d_A/\delta)})$.

**Estimating $d_K$ offline.** As our estimator $\overline{\mathbf{D}}_{N,1}^{-1}\overline{\mathbf{M}}_N \overline{\mathbf{D}}_{N,2}^{-1}$ is approximately rank-$d_K$, the number of nonzero eigenvalues of the estimator is a principled heuristic for determining $d_K$.

**Insufficiency of PCA and PMF for subspace estimation.** Naively performing PCA on the raw rewards or on single reward estimates $\hat{\beta}_n$ can lead to erroneous subspaces – as while the PCA target is linear-algebraically similar to $\overline{\mathbf{M}}_N$, it is statistically different. The PCA target (e.g. $\mathbb{E}[\hat{\beta}_n \hat{\beta}_n^\top]$) is typically full rank due to the variance of the per-reward noise $\epsilon$. On the other hand, PMF (Mnih and Salakhutdinov, 2007a) offers neither confidence bounds on the estimated subspace, nor a principled method for determining $d_K$.

## 4. Offline Data Sharpens Online Optimism

Here, we motivate and describe LOCAL-UCB, a natural algorithm that accelerates LinUCB with offline data. The core idea is **sharpening optimism** by being optimistic over the intersection of two confidence sets – one obtained using offline and online data and another purely from online data.

We geometrically motivate our update rule here, and illustrate it in Figure 1. After any $t$ steps, we can construct a $d_K$-dimensional confidence ellipsoid for every subspace in the subspace confidence set obtained from SOLD. The union of all these ellipsoids gives us our "offline confidence set"[4], called $\mathcal{C}_{\text{off}}^t(\boldsymbol{\beta})$. The usual $d_A$-dimensional ellipsoid forms our "online confidence set." We call this $\mathcal{C}_{\text{on}}^t(\boldsymbol{\beta})$. Since the true parameter lies in both confidence sets with high probability, being optimistic over their intersection allows us to sharpen or "further localize" optimism. Even though the offline confidence set uses both offline and online data, it will never shrink to a point due to the frozen subspace confidence set. So, we need the intersection of both sets to be sharply optimistic. This is the intuition behind LOCAL-UCB.

[4]The set is not only dependent on offline data, since online data is used to construct the $d_K$-dimensional ellipsoids.

---

**Algorithm 2** Latent Offline subspace Constraints for Accelerating Linear UCB (LOCAL-UCB)

---

1: **Input:** Projection matrix $\hat{\mathbf{U}}\hat{\mathbf{U}}^\top$, confidence bound $\Delta_{\text{off}}$ from an offline uncertainty-aware method, e.g. SOLD.
2: **Initialize** $\mathbf{V}_1 \leftarrow \mathbf{I}_{d_A}$, $\mathbf{b}_1 \leftarrow 0$, $\alpha_t$
3: **for** $t = 1, \ldots T$ **do**
4:     **Play** action $a_t$ and receive reward $r_t$ according to:

$$
a_t, \widetilde{\boldsymbol{\beta}}_t, \widetilde{\mathbf{U}}_t \leftarrow \arg\max_{a,\boldsymbol{\beta},\mathbf{U}} \phi(x_t, a)^\top \boldsymbol{\beta} \text{ such that}
$$
$$
\hat{\boldsymbol{\beta}}_{1,t} \leftarrow \mathbf{U}(\mathbf{U}^T\mathbf{V}_t\mathbf{U})^{-1}\mathbf{U}^\top \mathbf{b}_t,
$$
$$
\|\mathbf{U}^\top(\boldsymbol{\beta} - \hat{\boldsymbol{\beta}}_{1,t})\|_{(\mathbf{U}^\top\mathbf{V}_t\mathbf{U})^{-1}} \leq \alpha_{1,t}
$$
$$
\hat{\boldsymbol{\beta}}_{2,t} \leftarrow \mathbf{V}_t^{-1}\mathbf{b}_t, \|(\boldsymbol{\beta} - \hat{\boldsymbol{\beta}}_{2,t})\|_{\mathbf{V}_t^{-1}} \leq \alpha_{2,t}
$$
$$
\|\boldsymbol{\beta}\|_2 \leq R, \mathbf{U}^\top\mathbf{U} = \mathbf{I}_{d_K}, \mathbf{U}\mathbf{U}^\top\boldsymbol{\beta} = \boldsymbol{\beta},
$$
$$
\|\hat{\mathbf{U}}^\top\mathbf{U}\|_F \geq \sqrt{d_K - \Delta_{\text{off}}^2/2}
$$

5:     **Compute** $\mathbf{b}_{t+1} \leftarrow \mathbf{b}_t + \phi(x_t, a)r_t$, $\mathbf{V}_{t+1} \leftarrow \mathbf{V}_t + \phi(x_t, a)\phi(x_t, a)^\top$, $\alpha_{t+1}$
6: **end for**

---

We formalize this intuition in Algorithm 2 by formulating the sharpened optimism as an optimization problem in step 4. The first two constraints represent the low dimensional confidence ellipsoids in the subspace spanned by a given $\mathbf{U}$, while the next two merely represent the usual high dimensional ellipsoid. The remaining constraints let $\mathbf{U}$ range over our subspace confidence set.

We provide the following guarantee for LOCAL-UCB. Notice that our guarantee shows that for enough offline data with $N \gg T$, the effective dimension of the problem is $d_K$. It increases to $d_A$ as $T$ gets closer to $N$. The quality of the offline data is reflected in the coverage constants $\lambda_\theta$ and $\lambda_A$.

**Theorem 2** (LOCAL-UCB Regret). *Under Assumptions 1 and 2, if* $\alpha_{1,t} = R\sqrt{\mu} + CR\sqrt{d_K \log(2T/\delta)}$ *and* $\alpha_{2,t} = R\sqrt{\mu} + CR\sqrt{d_A \log(2T/\delta)}$ *for a universal constant $C$, then with probability at least $1 - \delta$ over offline data and online rewards, LOCAL-UCB has regret* $\text{Reg}_T$ *bounded by*

$$
O\left( \min\left( Rd_A\sqrt{T}, Rd_K\sqrt{T}\left( 1 + \frac{1}{\lambda_\theta \lambda_A^3}\sqrt{\frac{d_A T}{d_K N}} \right) \right) \right).
$$

However, the subspace constraint $\|\hat{\mathbf{U}}^\top\mathbf{U}\|_F \geq \sqrt{d_K - \Delta_{\text{off}}^2/2}$ is *nonconvex*. In fact, we lower bound a convex function in the constraint, making us search for $\widetilde{\mathbf{U}}_t$ over a complicated star-shaped set. So, it is unclear if LOCAL-UCB can be made computationally efficient.

## 5. Lower Bound

We now establish that LOCAL-UCB is in fact minimax optimal up to the coverage constants $\lambda_A, \lambda_\theta$ defined in Assumption 2. While we provide a full statement and proof of our lower bound in Appendix F, we provide an informal version here. Much like how we generate families of reward parameters in lower bound proofs for purely online regret, we are now generating a family of tuples of latent bandits (for the offline data) and reward parameters represented in the latent bandit (for the online interaction).

**Theorem 3.** *There exists a family of tuples $(F, \boldsymbol{\beta})$, where $F$ is a latent bandit with a rank $d_K$ latent subspace and $\boldsymbol{\beta}$ is a reward parameter in its support, so that for any offline behavior policy $\pi_b$ and any learner, $(i)$ $\lambda_\theta$ is uniformly bounded from below for all $F$, $(ii)$ there exists a $(F, \boldsymbol{\beta})$ such that the regret $\mathrm{Reg}(T, \boldsymbol{\beta})$ of the learner under offline data from $\pi_b$ and $F$ and online reward parameter $\boldsymbol{\beta}$ is bounded below by*

$$\Omega\left(\min\left(d_A\sqrt{T}, d_K\sqrt{T}\left(1 + \sqrt{\frac{d_A T}{d_K N}}\right)\right)\right)$$

To the best of our knowledge, this is the first lower bound in a hybrid (offline-online) sequential decision-making setting.[5] The key challenge is in selecting an instance space that yields an informative lower bound. When the offline data has insufficient coverage, one can show a trivial $d_A\sqrt{T}$ lower bound. Assuming $\lambda_\theta$ is uniformly bounded from below for all $F$ models scenarios with sufficient offline coverage, and our lower bound shows that even in these non-trivial settings, no algorithm can achieve a regret better than $\min\left(d_A\sqrt{T}, d_K\sqrt{T}\left(1 + \sqrt{\frac{d_A T}{d_K N}}\right)\right)$. While we analyze worst-case performance over a meaningful class of instances where the offline data is of sufficiently high quality, there remains room for future work on sharper, instance-dependent lower bounds that reflect explicit dependence on both $\lambda_\theta$ and $\lambda_A$.

## 6. Practical Optimism with ProBALL-UCB

While LOCAL-UCB is minimax-optimal, it is not computationally efficient due to the non-convex constraint discussed in Section 4. We address this by introducing ProBALL-UCB (Algorithm 3), a practical and computationally efficient algorithm. In this section, we first sketch the algorithm and then describe how it can be geometrically motivated as a relaxation of LOCAL-UCB.

ProBALL-UCB works in the subspace estimated by SOLD until the online confidence set is small enough. The algo-

---

[5]Pal et al. (2023) give lower bounds on the cumulative regret for a structure type of latent bandits (with hidden clusters). Their setting is purely online, although they rely on an offline matrix completion oracle during online learning.

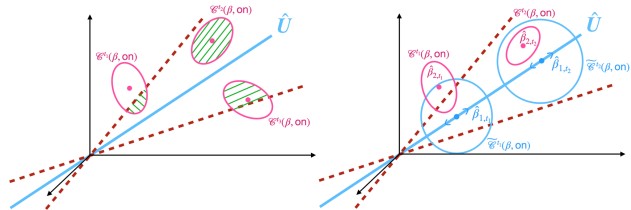

Figure 1: **Left:** Geometric interpretations of LOCAL-UCB. Showing $\mathcal{C}_{\mathrm{on}}^t(\boldsymbol{\beta}) \cap \mathcal{C}_{\mathrm{off}}^t(\boldsymbol{\beta})$ in green for three timepoints $t = t_1, t_2, t_3$. The dotted lines delineate the subspace confidence set. **Right:** Geometric interpretation of ProBALL-UCB. $\mathcal{C}_{\mathrm{on}}^{t_1}(\boldsymbol{\beta}) \not\subset \widetilde{\mathcal{C}}_{\mathrm{off}}^{t_1}(\boldsymbol{\beta})$, so we continue to use projections; but by time $t_2$, $\mathcal{C}_{\mathrm{on}}^{t_2}(\boldsymbol{\beta}) \subset \widetilde{\mathcal{C}}_{\mathrm{off}}^{t_2}(\boldsymbol{\beta})$, so we stop using projections.

rithm maintains a low-dimensional confidence set, a high-dimensional confidence set, and swaps between them to achieve acceleration. Once the cumulative error of using the low-dimensional confidence set ($\approx \Delta_{\mathrm{off}} T$ in ProBALL-UCB) exceeds the cumulative error of using the high-dimensional confidence set ($\approx d_A\sqrt{T}$), we stop using the former. This is instantiated with LinUCB, but the same idea can be immediately applied to other algorithms like SupLin-UCB or Bayesian algorithms like Thompson sampling.

We geometrically motivate our update rule here as a relaxation of LOCAL-UCB, and illustrate it in Figure 1, like in Section 4. We go through three stages of simplification over LOCAL-UCB, which suprisingly only leads to a minor degradation in provable guarantees.

- Cruder offline confidence sets are used. We take the subspace estimated by SOLD, compute a point estimate for $\beta_\star$ within the subspace, and construct a ball that contains the LOCAL-UCB offline confidence set. The online confidence set is still the standard $d_A$-dimensional ellipsoid.

- We wait for the offline confidence ball to contain the online confidence set, instead of taking intersections.

- We use a computable proxy for this subset condition instead of explicitly checking it.

As a final note before presenting the regret bound proper, there is a technical challenge with analyzing ProBALL-UCB. Since $\hat{\boldsymbol{\beta}}_{1,t}$ lies in $\hat{\mathbf{U}}$ but $\boldsymbol{\beta}_\star$ might not, the $d_K$-dimensional confidence ellipsoid bound no longer applies. We therefore prove our own confidence ellipsoid bound in Lemma 6, to bypass this issue.

**Theorem 4** (Regret for ProBALL-UCB). *Let $\alpha_{1,t} = R\sqrt{\mu} + \tau' R\Delta_{\mathrm{off}}\kappa_t + CR\sqrt{d_K\log(T/\delta)}$ and let $\alpha_{2,t} = R\sqrt{\mu} + CR\sqrt{d_A\log(T/\delta)}$. Let $S$ be the first timestep when Algorithm 3 does not play Line 6 and let $S = T$ if no such timestep exists. For $\tau = \tau' = 1$ we have that*

$$\mathrm{Reg}_T = \widetilde{O}\left(\min\left(\mathrm{Reg}_{on,T}, \mathrm{Reg}_{hyb,T}\right)\right).$$

---

**Algorithm 3** Projection and Bonuses for Accelerating Latent bandit Linear UCB (ProBALL-UCB)

---

1: **Input:** Projection matrix $\hat{\mathbf{U}}\hat{\mathbf{U}}^\top$, confidence bound $\Delta_{\text{off}}$. Hyperparameters $\alpha_{1,t}, \alpha_{2,t}, \tau, \tau'$.
2: **Initialize** $\mathbf{V}_1 \leftarrow I$, $\mathbf{b}_1 \leftarrow 0$, $\mathbf{C}_t \leftarrow 0$
3: **for** $t = 1, \ldots T$ **do**
4:    **if** $\Delta_{\text{off}}\tau\sqrt{t} + \Delta_{\text{off}}\tau'\sqrt{d_K \sum_{s=1}^t \kappa_s^2/t} \le d_A$ **then**
5:       **Compute** $\hat{\boldsymbol{\beta}}_{1,t} \leftarrow \hat{\mathbf{U}}(\hat{\mathbf{U}}^\top \mathbf{V}_t \hat{\mathbf{U}})^{-1}\hat{\mathbf{U}}^\top \mathbf{b}_t$
6:       **Play** $a_t \leftarrow \arg\max_a \phi(x_t, a)^\top \hat{\mathbf{U}}\hat{\mathbf{U}}^\top \hat{\boldsymbol{\beta}}_{1,t} + \alpha_{1,t}\|\phi(x_t,a)^\top \hat{\mathbf{U}}\|_{(\hat{\mathbf{U}}^\top \mathbf{V}_t \hat{\mathbf{U}})^{-1}}$
7:    **else**
8:       **Compute** $\hat{\boldsymbol{\beta}}_{2,t} \leftarrow \mathbf{V}_t^{-1}\mathbf{b}_t$
9:       **Play** $a_t \leftarrow \arg\max_a \phi(x_t, a)^\top \hat{\boldsymbol{\beta}}_{2,t} + \alpha_{2,t}\|\phi(x_t,a)\|_{\mathbf{V}_t^{-1}}$
10:    **end if**
11:    **Observe** reward $r_t$ and update $\mathbf{b}_{t+1} \leftarrow \mathbf{b}_t + \phi(x_t,a)r_t$, $\mathbf{V}_{t+1} \leftarrow \mathbf{V}_t + \phi(x_t,a)\phi(x_t,a)^\top$
12:    **Update** $\mathbf{C}_{t+1} \leftarrow \mathbf{C}_t + \hat{\mathbf{U}}^\top\phi(x_t,a_t)\phi(x_t,a_t)^\top$, $\kappa_{t+1} \leftarrow \|\mathbf{C}_{t+1}\|_{(\hat{\mathbf{U}}^\top \mathbf{V}_{t+1}\hat{\mathbf{U}})^{-1}}$
13: **end for**

---

*where* $\text{Reg}_{on,T} = Rd_A\sqrt{T}$ *and* $\text{Reg}_{hyb,T}$ *is defined as*

$$Rd_K\sqrt{T}\left(1 + \frac{1}{\lambda_A^3 \lambda_\theta}\left(\sqrt{\frac{d_A T}{d_K N}} + \sqrt{\frac{d_A}{SN}\sum_{t=1}^S \kappa_t^2}\right)\right).$$

*In the worst case, $\kappa_t = O(t)$ and so $\frac{1}{S}\sum_{t=1}^S \kappa_t^2 = O(T^2)$, but if all features $\phi(x_t, a_t)$ lie in the span of $\hat{U}$ for $t \le S$, then $\frac{1}{S}\sum_{t=1}^S \kappa_t^2 = O(T)$.*

While the regret bound looks weaker in the worst case, we emphasize that the "good case" in Theorem 4 is quite common. As an illustrative example, if the feature set $\mathcal{F}_t = \{\phi(x_t, a) \mid a \in \mathcal{A}\}$ is an $\ell_2$ ball, then the maximization problem in Step 6 will always choose $a_t$ with $\phi(x_t, a_t)$ in the span of $\hat{U}$. This can also approximately hold if the features are roughly isotropic or close to the span of $\hat{U}$. We direct the reader to Appendix E.2.1 for further discussion.

Furthermore, Theorem 4 shows that ProBALL-UCB performs no worse than LinUCB, and can significantly outperform it both in theory and in practice, as we will see in the following section.

## 7. Experiments

We now establish the practical efficacy of SOLD (Algorithm 1) and ProBALL-UCB (Algorithm 3) for linear latent contextual bandits through a series of numerical experiments.[6] We perform a simulation study and a demonstration using

---

[6]See https://github.com/hetankevin/probono for source code.

real-life data. While specific details of the experiments and many ablation studies are in Appendix H, we sketch our experiments and discuss key observations in this section.

In all experiments, we obtain confidence bounds $\Delta_{\text{off}}$ using three different concentration inequalities – (1) Hoeffding as in Proposition 2 (H-ProBALL), (2) empirical Bernstein as in Proposition 1 (E-ProBALL), and (3) the martingale Bernstein concentration inequalities of (Waudby-Smith and Ramdas, 2023) (M-ProBALL). We use a simpler expression for $\Delta_{\text{off}}$, set $\tau' = 0$, and choose a suitable value of the hyperparameter $\tau$ to adjust for overly conservative $\Delta_{\text{off}}$[7]. We later vary $\tau$ in ablation experiments to demonstrate that our results are not a consequence of our choice of hyperparameters. Finally, for the MovieLens experiments, we additionally design a natural Thompson sampling version of ProBALL-UCB to highlight the applicability of the ProBALL idea called ProBALL-TS in Appendix G. All experiments for ProBALL-TS are in Appendix H.3.2.

**Simulation study.** We first perform a simulation study on a latent linear bandit with $d_A = 50$ and $d_K = 2$, with 5000 trajectories generated offline. Further details are provided in Appendix H, and the results are presented in Figure 2. Note that ProBALL-UCB (Algorithm 3) performs no worse than LinUCB, no matter what we choose for $\tau$ and $\Delta_{\text{off}}$. However, we see a clear benefit from using tighter confidence bounds – as $\Delta_{\text{off}}$ gets smaller, Algorithm 3 chooses to utilize the projected estimate $\hat{\boldsymbol{\beta}}_{1,t}$ more often, resulting in better performance. Note that the kinks in the regret curves correspond to points where Algorithm 3 switches over to the higher dimensional optimism in step 7.

**MovieLens dataset.** In line with (Hong et al., 2020), we assess the performance of our algorithms on real data using the MovieLens dataset. Like them, we filter the dataset to include only movies rated by at least 200 users and vice versa, and apply probabilistic matrix factorization (PMF) to the rating matrix to generate ground truth user preferences for online experiments. Applying PMF gives $d_K = 18$ and we choose $d_A = 200$, generating 5000 trajectories offline. For baselines, we reproduce the methods of (Hong et al., 2020; 2022) and implement LinUCB with canonical hyperparameter choices.

We initialize ProBALL-UCB with a subspace estimated with an unregularized variant of SOLD (see Appendix G) that uses pseudo-inverses instead of inverses. This is due to difficulties in finding an appropriate regularization parameter for this large, noisy, and high-dimensional dataset. Figure 2 depicts the result of this experiment. Once again, ProBALL-UCB performs no worse than LinUCB, no matter what we choose for $\tau$ and $\Delta_{\text{off}}$, and the benefit of using tighter con-

---

[7]Namely, we set $\Delta_D = 0$ in $\Delta_{\text{off}}$. Also, Lemma 6 and some thought reveal that choosing $\tau' = 0$ recovers the "good" version of ProBALL-UCB guarantees, if features are isotropic enough.

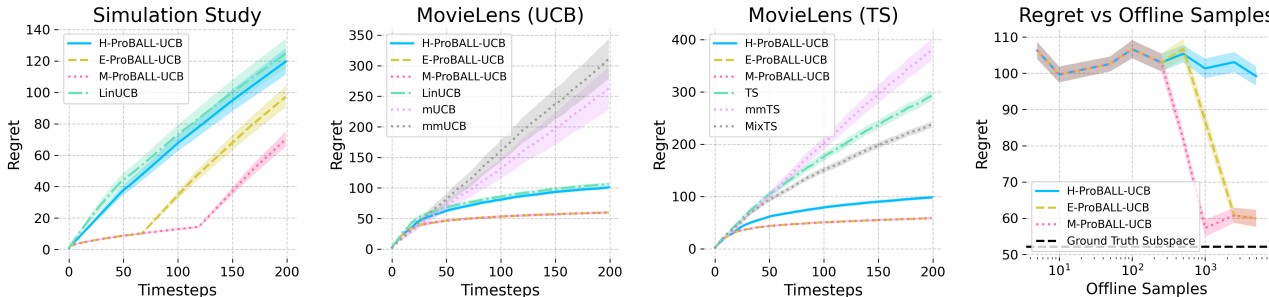

Figure 2: Left to Right. **First:** Simulation study comparison of ProBALL-UCB against LinUCB for $\tau = 5$. **Second/Third:** Comparison of ProBALL-UCB initialized with SOLD against {LinUCB, mUCB, and mmUCB, TS, mmTS, and MixTS}, for $\tau = 0.1$ and various confidence bound constructions. ProBALL-UCB outperforms all other algorithms, and approaches the performance of LinUCB when Hoeffding confidence sets are used. **Fourth**: ProBALL-UCB regret on MovieLens against offline samples used in SOLD, compared to LinUCB on ground-truth low-dimensional features. Here, $\tau = 0.1, T = 200$. As the number of offline samples increases, SOLD recovers a low-rank subspace almost as good as ground-truth. The shaded area in each sub-figure depicts 1-s.e. confidence intervals over 30 trials with fresh $\boldsymbol{\theta}$, accounting for the variation in frequentist regret for changing $\boldsymbol{\theta}$.

fidence bounds remains. With $\tau = 0.5$, ProBALL-UCB with martingale Bernstein confidence bands stops using the projected estimates at around timestep 70, but still continues to outperform LinUCB. Although mUCB and mmUCB perform slightly better than ProBALL-UCB and LinUCB at the beginning, the model misspecification incurred by discretizing the features into $d_K$ clusters ensures that it typically suffers linear regret in this scenario. The lower initial performance of ProBALL-UCB and Lin-UCB is a consequence of their higher initial exploration.

**Ablation study.** While the end-to-end performance of ProBALL-UCB significantly improves over existing algorithms, we also address further questions about various components of our method in Appendix H. We first show that the rank $d_K$ can be determined from offline data via the procedure outlined in Section 3 of using the eigenvalues of $\overline{\mathbf{D}}_{N,1}^{-1}\overline{\mathbf{M}}_N\overline{\mathbf{D}}_{N,2}^{-1}$ to determine the rank of our subspace. Second, we study the effect of varying the hyperparameter $\tau$ and note that our method stably outperforms existing methods at all reasonable values of $\tau$. Third, we compare different combinations of algorithms in our figures above, side by side. Finally, we evaluate the effect of offline data by plotting the online regret against the number of offline samples used to estimate the latent subspace.

## 8. How General Are Latent Bandits?

While have established that latent bandits are a powerful framework for accounting for uncertainty in reward models, the extent of their generality is unclear. Are there other stateless decision processes that generalize over latent bandits? We cap off our contributions by establishing the generality of latent bandits. In this section, we show a de Finetti

theorem for decision processes, demonstrating that *every* "coherent" and "exchangeable" stateless (contextual) decision process is a latent (contextual) bandit. We first define a stateless decision process at a high level of generality.[8]

**Definition 2.** A *stateless decision process* (SDP) with action set $\mathcal{A}$ is a probability space $(\Omega, \mathcal{G}, \mathbb{P})$ with a family of random maps $\mathcal{F}_H : \Omega \rightarrow (\mathcal{A}^H \rightarrow \mathbb{R}^H)$ for $H \in \mathbb{N} \cup \{\infty\}$.

That is, given a sequence of actions $(a_1, \ldots a_H)$, an SDP generates a random sequence of rewards $(Y_1, \ldots Y_H)$. As such, we abuse notation to denote by $\mathcal{F}_H(a_1, \ldots a_H)$ the random variable $\omega \mapsto \mathcal{F}_H(\omega)(a_1, \ldots a_H)$. Without any coherence between $\mathcal{F}_H$ across $H$, a stateless process can behave arbitrarily for different horizons $H$. We present a natural coherence condition below, essentially requiring that a given action sequence should produce consistent rewards.

**Definition 3.** A stateless decision process is *coherent* if for any $h \leq k \leq H, H' \in \mathbb{N} \cup \{\infty\}$ and for any two action sequences $\tau, \tau'$ of lengths $H$ and $H'$ sharing the same actions $(a_h, \ldots a_k)$ from index $h$ to $k$, with $\mathcal{F}_H(\tau) = (Y_1, \ldots Y_H)$ and $\mathcal{F}_{H'}(\tau') = (Y_1', \ldots Y_{H'}')$, we have $(Y_h, \ldots Y_k) = (Y_h', \ldots Y_k')$, viewed as functions of $\Omega$.

It is natural to require equality in value and not just in distribution, since after taking an extra action, the *values* of past rewards stay the same, not just their *distribution*. For example, if we pull 10 different jackpot levers and then pull a new one, the previous 10 outcomes stay the same in value, not just in distribution. We also give a natural definition for exchangeability of a stateless decision process – namely that

---

[8](Liu et al., 2023) work with a much more restrictive notion of a generalized bandit and use the original de Finetti theorem in some of their lemmas. See Appendix B.1 for a discussion.

exchanging any two rewards should lead to the distribution obtained by exchanging the corresponding actions.

**Definition 4.** A stateless decision process is *exchangeable* if for any permutation $\pi : [H] \to [H]$ and $\mathcal{F}_H(a_1, \ldots a_H) = (Y_1, \ldots Y_H)$, we have $\mathcal{F}_h(a_{\pi(1)}, \ldots a_{\pi(H)}) \sim (Y_{\pi(1)}, \ldots Y_{\pi(H)})$.

Finally, a latent bandit is an SDP that behaves like a bandit *conditioned* on a random latent state $F$ that determines the reward distribution. As $F$ determines a distribution, it is a random measure-valued function on $\mathcal{A}$.

**Definition 5.** A latent bandit is a stateless decision process equipped with a random measure-valued function $F : \Omega \to (\mathcal{A} \to \mathcal{P}(\mathbb{R}))$ so that for any $H$ and action sequence $(a_1, \ldots a_H)$, the rewards $(Y_1, \ldots Y_H) := \mathcal{F}_H(a_1, \ldots a_H)$ are independent conditioned on $F$. Moreover, the conditional distribution $\mathcal{L}[Y_h \mid F] = F(a_h)$ for all $h \leq H$.[9]

As such, the latent bandit is indeed a special case of an SDP, where the function $\mathcal{F}_H$ is induced by the latent state random variable $F$.

While exchangeability and coherence are reasonable conditions on an SDP and are clearly satisfied by latent bandits, it is a-priori unclear if they are sufficient to ensure that the SDP is a latent bandit. As only exchanging rewards from the *same action* preserves the distribution, standard de Finetti proof ideas do not immediately apply. After all, it is possible that an SDP could be cleverly designed to choose rewards adaptively across time and satisfy these properties. Reassuringly, no such counterexamples exist, guaranteed by the following theorem.

**Theorem 5** (De Finetti Theorem for Stateless Decision Processes). *Every exchangeable and coherent stateless decision process is a latent bandit.*

We show in Lemma 2 in Appendix A that coherence is not a consequence of exchangeability – it is a necessary condition for being a latent bandit. Finally, we analogously consider contexts and define "transition-agnostic contextual decision processes" (TACDPs) in Appendix B.2. We define coherence and coherence and exchangeability for TACDPs, and define latent *contextual* bandits by simply replacing $\mathcal{A}$ with $\mathcal{X} \times \mathcal{A}$ in the definitions above. We then show an analogous de Finetti theorem, as a corollary of our proof of Theorem 5. See Appendix B.6 for more details.

**Linear latent contextual bandits and SDPs.** Finally, note that this section is faithful to the rest of this paper.

A linear latent contextual bandit is a latent contextual bandit where the random measure-valued function $F : \Omega \to ((\mathcal{X} \times \mathcal{A}) \to \mathcal{P}(\mathbb{R}))$ is defined by setting $F(\omega)(x, a)$ to be the distribution given by $\phi(x, a)^\top \mathbf{U}_\star \boldsymbol{\theta}(\omega) + \epsilon$, for any $\omega \in \Omega$ and $(x, a) \in (\mathcal{X} \times \mathcal{A})$. We have seen that every coherent and exchangeable stateless contextual decision process is a latent contextual bandits, of which the linear latent contextual bandit is an important special case.

## 9. Discussion, Limitations and Further Work

In this paper, we have addressed the problem of leveraging offline data to accelerate online learning in linear latent bandits. Our work has a few limitations. First, while ProBALL-UCB is practical and computationally efficient, it has a slightly weaker worse-case guarantee than LOCAL-UCB. Second, when the data is noisy, it can be hard to tune the regularization $\mu$. We use a pseudoinverse-based version of SOLD in such a case (Appendix G), implemented in our code. This variant is easy to tune and performs well empirically. Third, the offline uncertainty sets computed using $\Delta_{\text{off}}$ can be overly conservative, and a discount hyperparameter $\tau$ for deciding when to switch between using the offline confidence ball and the online confidence set within ProBALL-UCB must be fine-tuned online.

Despite these limitations, our work enjoys strong theoretical guarantees and convincing empirical performance. We hope that this method opens the door for developing other efficient and scalable algorithms for sequential decision-making with continuous latent states. One can use ideas presented in this paper to design similar algorithms for MDPs, linear MDPs, and RL or bandits with general function approximation.

## Impact Statement

This paper presents work whose goal is to advance the field of Machine Learning. There are many potential societal consequences of our work, none which we feel must be specifically highlighted here.

## Acknowledgements

CK would like to acknowledge the support of the Rackham International Student Fellowship (the Indian Alumni Fellowship) for this work. KT is supported in part by the NSF grant CCF-2106778.

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

# Contents

# A. Additional Lemmas and Discussion

## A.1. Broader Impacts

Often, protected groups with private identities have temporal behavior correlated with their private identities. Latent state estimation methods have the potential to identify such private identities online, and must be used with care. Use of such methods should comply with the relevant privacy and data protection acts, and corporations with access to a large customer base as well as governments should be mindful of the impact of the use of latent state methods on their customers and citizens respectively.

## A.2. Coherence with Equality is Necessary

**Lemma 2** (Coherence with Equality is Necessary). *There exists a stateless decision process that is exchangeable and not coherent but satisfies the following property: for any two action sequences $\tau, \tau'$ of lengths $H$ and $H'$ sharing the same actions $(a_h, \ldots a_k)$ from index $h$ to $k$, with $\mathcal{F}_H(\tau) = (Y_1, \ldots Y_H)$ and $\mathcal{F}_{H'}(\tau') = (Y_1', \ldots Y_{H'}')$, we have $(Y_h, \ldots Y_k) \sim (Y_h', \ldots Y_k')$. Additionally, since the SDP is not coherent, it is not a latent bandit.*

*Proof.* Consider an SDP with action set $\mathcal{A} = \{0, 1\}$ equipped with a random variable $\theta \in \{0, 1\}$ distributed as $Ber(1/2)$. If the first action is 0, then all rewards are given by $\theta$. If the first action is 1, then the first reward $X_1 = \theta$ while all future rewards are $1 - \theta$.

Notice now that for any timestep $h$, its rewards are $Ber(1/2)$. We note that this process is clearly exchangeable. Moreover, this process is coherent in a weaker sense – for any two action sequences $\tau, \tau'$ of lengths $H$ and $H'$ sharing the same actions $(a_h, \ldots a_k)$ from index $h$ to $k$, with $\mathcal{F}_H(\tau) = (Y_1, \ldots Y_H)$ and $\mathcal{F}_{H'}(\tau') = (Y_1', \ldots Y_{H'}')$, we have $(Y_h, \ldots Y_k) \sim (Y_h', \ldots Y_k')$. That is, the reward subsequences are only equal *in distribution*. This is easy to see, since if $a_1$ is not included in the action subsequence or if $a_1 = 0$, the reward subsequence is always given by $(X, \ldots X)$ for $X \sim Ber(1/2)$. If $a_1 = 1$ and $a_1$ is in the action subsequence, then the reward subsequence is always given by $(-X, X, \ldots X)$ for $X \sim Ber(1/2)$.

However, the reward subsequences are not equal *in value*. If the action subsequence is inside two action sequences $\tau = (a_1, \ldots a_H)$ and $\tau' = (a_1', \ldots a_H')$ with $a_1 = 0$ and $a_1' = 1$, then their rewards are negatives of each other. So, this SDP is not coherent. This means that this SDP is not a latent bandit. $\square$

## A.3. Contexts, Latent State and Behavior Policy cannot be Dependent

**Lemma 1** (Contexts, $\boldsymbol{\theta}$, and $\pi_b$ cannot be dependent). *For each of these conditions:*

1. *Contexts in a trajectory are dependent but do not depend on $\boldsymbol{\theta}$, and $\pi_b$ also does not use $\boldsymbol{\theta}$,*

2. *Contexts are generated independently using $\boldsymbol{\theta}$, while $\pi_b$ does not use $\boldsymbol{\theta}$,*

3. *Contexts are generated independently without using $\boldsymbol{\theta}$, while $\pi_b$ uses $\boldsymbol{\theta}$,*

*there exist two different linear latent contextual bandits with orthogonal latent subspaces satisfying the condition, and a behavior policy $\pi_b$ so that the offline data distributions are indistinguishable and cover all $(x, a)$ pairs with probability at least $1/4$. Since the latent subspaces are orthogonal, an action that gives the maximum reward on one latent bandit gives reward 0 on the other.*

*Proof.* Let $e_1, e_2$ be the standard basis of $\mathbb{R}^2$. For all these examples, our two latent bandits will satisfy the following:

- **Latent contextual bandit** 1: Let $\boldsymbol{\theta}$ take values $e_1 + e_2$ and $-e_1 - e_2$ with probability $1/2$ each.

- **Latent contextual bandit** 2: Let $\boldsymbol{\theta}$ take values $e_1 - e_2$ and $e_2 - e_1$ with probability $1/2$ each.

### A.3.1. CONTEXTS CANNOT BE DEPENDENT ON EACH OTHER

We consider two actions, so $\mathcal{A} = \{0, 1\}$ We have two contexts $x, y$ so that $\phi(x, 0) = e_1$, $\phi(x, 1) = 2e_1$ while $\phi(y, 0) = e_2$, $\phi(y, 1) = 2e_2$. We design them to be dependent so that any trajectory either only sees $x$ or only sees $y$. However, $x$ and $y$ are both seen with probability $1/2$. Pick $\pi_b$ so that it takes each action with probability $1/2$. Now note that the mean reward of $x, 0$ in either latent bandit takes values $\pm 1$ with probability $1/2$. So, the offline data distributions are also indistinguishable and every context-action pair is seen with probability at least $1/4$.

A.3.2. CONTEXTS AND LATENT STATE CANNOT BE DEPENDENT

We consider two actions, so $\mathcal{A} = \{0, 1\}$ Again, consider two contexts $x, y$ so that $\phi(x, 0) = e_1$, $\phi(x, 1) = 2e_1$ while $\phi(y, 0) = e_2$, $\phi(y, 1) = 2e_2$. Let us say that for either latent bandit and for any $\boldsymbol{\theta}$, the context distribution is a Dirac-$\delta$ over the context whose features have a positive dot product with $\boldsymbol{\theta}$. Then, note that either context is seen in both latent bandits with probability $1/2$. Again, let $\pi_b$ take either action with probability $1/2$. So, the offline data distributions are indistinguishable and every context-action pair is seen with probability at least $1/4$.

A.3.3. LATENT STATE AND BEHAVIOR POLICY CANNOT BE DEPENDENT

We consider four actions, so that $\mathcal{A} = \{0, 1, 2, 3\}$. Let there be a single context $x$ with $\phi(x, 0) = e_1, \phi(x, 1) = e_2, \phi(x, 2) = -e_1, \phi(x, 3) = -e_2$. For any latent state $\boldsymbol{\theta}$ in either bandit, let $\pi_b$ be uniform over the actions that have a positive dot product with $\boldsymbol{\theta}$. It is then easy to see that the offline data distributions are indistinguishable and every context-action pair is seen with probability at least $1/4$. $\qquad\square$

# B. A de Finetti Theorem for decision processes

## B.1. Discussion on (Liu et al., 2023)'s smaller class of generalized bandits

We prove the de Finetti theorem for a very general formulation of decision processes. However, past work (Liu et al., 2023) has studied a simpler generalization of bandits, namely a stochastic process valued in $\mathbb{R}^{\mathcal{A}}$. A sample point in this space is a *sequence of functions* in $\mathcal{A} \to \mathbb{R}$, which rules out the possibility of adaptivity. In contrast, a sample point in our space is a *function of sequences*, which subsumes all sample points of their space, but allows for adaptivity.

(Liu et al., 2023) are able to use the original de Finetti theorem on their random process directly, but work with a much more restrictive kind of decision process. We show that even when considering much more general stateless decision processes, latent bandits are the "right" objects produced by a de Finetti theorem for stateless decision processes.

## B.2. Proof of the De Finetti Theorem for Stateless Decision Processes

**A note on $F$, measurability and well-definedness in the definition of a latent bandit.** Recall that $F$ in the definition of a latent bandit needs to be a measurable map $\Omega \to (\mathcal{A} \to \mathcal{P}(\mathbb{R}))$. To define measurability for the output space of functions $(\mathcal{A} \to \mathcal{P}(\mathbb{R}))$, we endow $\mathcal{P}(\mathbb{R})$ with the topology of weak convergence, endow the space $\mathcal{P}(\mathbb{R})^{\mathcal{A}}$ of maps $\mathcal{A} \to \mathcal{P}(\mathbb{R})$ with the topology of point-wise convergence, and require $F$ to be measurable w.r.t. the induced Borel $\sigma$-algebra. We also recall two abuses of notation in the definition of a latent bandit. First, we abuse notation to define the random measure $F(a) := (\omega \mapsto F(\omega)(a))$. Second, we also abuse notation to conflate $F(a)$ with the curried map $\kappa_a(\omega, B) := F(a)(\omega)(B)$, which is a map $\Omega \times \mathcal{B} \to [0,1]$. This map $\kappa_a$ will turn out to be a kernel by the construction of $F$. Equating $F(a)$ to a regular conditional distribution in the definition of a latent bandit *requires* that $\kappa_a$ be a kernel.

We recall our de Finetti theorem for stateless decision processes here.

**Theorem 5** (De Finetti Theorem for Stateless Decision Processes). *Every exchangeable and coherent stateless decision process is a latent bandit.*

*Proof.* Consider an exchangeable and coherent stateless decision process. We will establish that there is a latent bandit with the same reward distribution as this process for any sequence of actions.

For any sequence $\tau = (a_1, \ldots a_H)$, denote by $(Y_{\tau,1}, \ldots Y_{\tau,H}) := \mathcal{F}_H(a_1, \ldots a_H)$. We intend to establish that there is a random measure-valued function $F$ such that $(Y_{\tau,1}, \ldots Y_{\tau,H})$ are independent given $F$ and $\mathcal{L}[Y_{\tau,h} \mid F] = F(a_h)$ for all $h \leq H$ almost surely. Since conditional independence is a property of finite subsets of a set of random variables, it suffices to show this for finite $H$. The version for $H = \infty$ will immediately follow by coherence.

First recall that regular conditional distributions $\mathcal{L}[Y \mid F]$ are almost surely unique under if the $\sigma$-algebra of the output space of $Y$ is countably generated. Since the Borel $\sigma$-algebra on $\mathbb{R}$ is countably generated, this is true for our case. We also recall for the rusty reader that conditioning on a random variable is the same as conditioning on the induced $\sigma$-algebra in the domain.

## B.3. Constructing $F$

Fix any finite trajectory $\tau = (a_1, \ldots a_H)$ and index $h$.

**The trick:** Consider the infinite sequence $\tau_\infty := (a_h, a_h, \ldots)$. By coherence, $Y_{\tau,h} = Y_{\tau_\infty,h}$. Now $\tau_\infty$ is a sequence where exchanging any finite set of rewards preserves the reward distribution, since all actions are identical. Since $\mathbb{R}$ is locally compact, we can apply the usual de Finetti representation theorem (Theorem 12.26 in (Klenke, 2008)) to conclude that:

- The random measure $\Xi_{a_h} = \mathrm{wlim}_{n \to \infty} \frac{1}{n} \sum_{j=1}^n \delta_{Y_{\tau_\infty,j}}$ is well defined. Here $\mathrm{wlim}$ is the weak limit of measures.

- The regular conditional distribution $\mathcal{L}[Y_{\tau_\infty,j} \mid \Xi_{a_h}] = \Xi_{a_h}$ for all $j$.

Since $Y_{\tau,h} = Y_{\tau_\infty,h}$, we conclude that the conditional distribution $\mathcal{L}[Y_{\tau,h} \mid \Xi_{a_h}] = \mathcal{L}[Y_{\tau_\infty,h} \mid \Xi_{a_h}] = \Xi_{a_h}$.

**Constructing $F$:** For every action, consider such a random measure $\Xi_{a_h}$ and define a random measure-valued function $F : \Omega \to (\mathcal{A} \to \mathcal{P}(\mathbb{R}))$ in the following manner: for any $\omega \in \Omega$, define $F(\omega)(a) := \Xi_{a_h}(\omega)$. We know that for any $a$, $\Xi_{a_h}$ is measurable w.r.t. the topology of weak convergence on $\mathcal{P}(\mathbb{R})$. It is now tedious but straightforward to verify that $F$ is measurable w.r.t. the Borel $\sigma$-algebra generated by the topology of pointwise convergence on $\mathcal{P}(\mathbb{R})^{\mathcal{A}}$.

**B.4. Establishing that $\mathcal{L}[Y_{\tau,h} \mid F] = F(a_h)$ almost surely for all $h$**

Again, fix any finite trajectory $\tau = (a_1, \ldots a_H)$ and index $h$. Recall that we abuse notation to denote by $F(a)$ the random-measure $\omega \mapsto F(\omega)(a)$. In particular, for any measurable set $B \subset \mathbb{R}$, we conflate $F(a)(\omega)(B) = F(\omega)(a)(B) = F(a)(\omega, B)$, where the last equality holds since $F$ is a regular conditional distribution. Note that $F(a_h) = \Xi_{a_h}$ by the construction of $F$. Since $\Xi_{a_h} = F(a_h)$, we have $\mathcal{L}[Y_{\tau,h} \mid F(a_h)] = F(a_h)$. Thus, it suffices to show that $\mathcal{L}[Y_{\tau,h} \mid F] = \mathcal{L}[Y_{\tau,h} \mid F(a_h)]$.

**Lemma 3.** $\mathcal{L}[Y_{\tau,h} \mid F] = \mathcal{L}[Y_{\tau,h} \mid F(a_h)]$ *almost surely.*

*Proof.* First note that $F(a_h)$ is measurable w.r.t. $F$. For showing this, view the set of maps $\mathcal{P}(\mathbb{R})^{\mathcal{A}}$ as the product set $\prod_{a'} \mathcal{P}(\mathbb{R})_{a'}$. Now merely note that $F^{-1}(E \times \prod_{a' \neq a} \mathcal{P}(\mathbb{R})_{a'}) = (F(a_h))^{-1}(E)$ for any measurable subset $E \subset \mathcal{P}(\mathbb{R})_a$. Hence, $F(a_h)$ is measurable w.r.t. $F$.

Let $\mathcal{B}$ be the Borel $\sigma$-algebra on $\mathbb{R}$. Now recall that the regular conditional distribution $\mathcal{L}[X \mid G]$ for a real-valued random variable $X$ is the almost surely unique kernel $\kappa_{G,X} : \Omega \times \mathcal{B} \to \mathbb{R}$ such that:

- $\omega \to \kappa_{G,X}(\omega, B)$ is $G$-measurable for any set $B \in \mathcal{B}$

- $B \to \kappa_{G,X}(\omega, B)$ is a probability measure on $\mathbb{R}$ for any sample point $\omega \in \Omega$.

- For any measurable set $B \subset \mathbb{R}$ and any $G$-measurable set $A$,

$$\mathbb{E}[\mathbb{1}_B(Y)\mathbb{1}_A] = \int \mathbb{1}_B(Y(\omega))\mathbb{1}_A(\omega)d\mathbb{P}(\omega) = \int \kappa_{G,X}(\omega, B)\mathbb{1}_A(\omega)d\mathbb{P}(\omega)$$

We will show our claim using the definition and a.s. uniqueness of the regular conditional distribution in our case. Consider any $F$-measurable set $A \subset \Omega$ and Borel-measurable set $B \subset \mathbb{R}$, $B \in \mathcal{B}$. Denote by $\kappa_F := \mathcal{L}[Y_{\tau,h} \mid F]$ and by $\kappa_{a_h} := \mathcal{L}[Y_{\tau,h} \mid F(a_h)] = F(a)$. Note that by the coherence and exchangeability in section B.3,

$$\int \kappa_F(\omega, B)\mathbb{1}_A(\omega)d\mathbb{P}(\omega) = \mathbb{E}[\mathbb{1}_B(Y_{\tau,h})\mathbb{1}_A] = \mathbb{E}[\mathbb{1}_B(Y_{\tau_\infty,h})\mathbb{1}_A] = \mathbb{E}[\mathbb{1}_B(Y_{\tau_\infty,j})\mathbb{1}_A]$$

for all $j$. Averaging all these equations and taking a limit, we get that

$$\int \kappa_F(\omega, B)\mathbb{1}_A(\omega)d\mathbb{P}(\omega) = \mathbb{E}[\mathbb{1}_B(Y_{\tau,h})\mathbb{1}_A] = \mathbb{E}[\mathbb{1}_B(Y_{\tau_\infty,h})\mathbb{1}_A]$$

$$= \lim_{n \to \infty} \frac{1}{n} \sum_{j=1}^{n} \mathbb{E}[\mathbb{1}_B(Y_{\tau_\infty,j})\mathbb{1}_A]$$

$$= \mathbb{E}\left[\left(\lim_{n \to \infty} \frac{1}{n} \sum_{j=1}^{n} \mathbb{1}_B(Y_{\tau_\infty,j})\right)\mathbb{1}_A\right]$$

where the last equality holds by the dominated convergence theorem, if the limit exists. To establish that the limit exists and compute it, we apply the usual de Finetti theorem – specificaly point $(i)$ of remark 12.27 in (Klenke, 2008) with $f$ set to the identity map. This gives us that $\lim_{n \to \infty} \frac{1}{n} \sum_{j=1}^{n} \mathbb{1}_B(Y_{\tau_\infty,j}) = \mathbb{E}[\mathbb{1}_B(Y_{\tau_\infty,h}) \mid \Xi_{a_h}] = \Xi_{a_h}(B)$, where $\Xi_{a_h}(B)$ is the random variable $\omega \mapsto \Xi_{a_h}(\omega)(B)$. This in turn satisfies $\Xi_{a_h}(\omega)(B) = F(a_h)(\omega)(B) = \kappa_{a_h}(\omega, B)$. This establishes that for any $F$-measurable set $A$ and Borel set $B$,

$$\int \kappa_F(\omega, B)\mathbb{1}_A(\omega)d\mathbb{P}(\omega) = \int \kappa_{a_h}(\omega, B)\mathbb{1}_A(\omega)d\mathbb{P}(\omega)$$

. In conclusion, $\kappa_{a_h}$ satisfies:

- $F(a_h)(B) := \omega \to \kappa_{a_h}(\omega, B)$ is $F$-measurable for any set $B \in \mathcal{B}$ since $F(a_h)(B)$ is $F(a_h)$ measurable by definition of $\kappa_{a_h}$ and $F(a_h)$ is $F$-measurable from above.

- $B \to \kappa_{a_h}(\omega, B)$ is a probability measure on $\mathbb{R}$ for any sample point $\omega \in \Omega$ by definition of $\kappa_{a_h}$

- For any measurable set $B \subset \mathbb{R}$ and any $F$-measurable set $A$, by the argument above,

$$\mathbb{E}[\mathbb{1}_B(Y_{\tau,h})\mathbb{1}_A] = \int \kappa_F(\omega, B)\mathbb{1}_A(\omega)d\mathbb{P}(\omega) = \int \kappa_{a_h}(\omega, B)\mathbb{1}_A(\omega)d\mathbb{P}(\omega)$$

By the definition as well as a.s. uniqueness of regular conditional distributions in our case, this establishes that $\mathcal{L}[Y_{\tau,h} \mid F] = \kappa_F = \kappa_{a_h} = \mathbb{E}[\mathbb{E}[Y_{\tau,h} \mid F(a_h)]$ almost surely.

$\square$

### B.5. Establishing conditional independence of rewards

This is the trickier to establish. It suffices to show that for any finite length $h$ trajectory $\tau = (a_1, \ldots a_H)$ and any tuple $(f_1, \ldots f_H)$ of bounded measurable functions, $\mathbb{E}\left[\prod_{h=1}^{H} f_h(Y_{\tau,h}) \mid F\right] = \prod_{h=1}^{H} \mathbb{E}[f_h(Y_{\tau,h}) \mid F]$. Equivalently, it suffices to show that for any bounded $F$-measurable real-valued random variable $U$, the following holds.

$$\mathbb{E}\left[U \prod_{h=1}^{H} f_h(Y_{\tau,h})\right] = \mathbb{E}\left[U \prod_{h=1}^{H} \mathbb{E}[f_h(Y_{\tau,h}) \mid F]\right]$$

We will show this by induction on $H$. This clearly holds for $H = 1$ by section B.3 above. Now assume that this holds for $H = l - 1$.

**Replacing the last term, $f_l(Y_{\tau,l})$, by an empirical average:** Fix any trajectory $\tau = (a_1, \ldots a_l)$. First consider the infinite trajectory $\tau' = (a_1, \ldots a_{l-1}, a_l, a_l, a_l, \ldots)$. This creates a random sequence of rewards $Y_{\tau',1}, Y_{\tau',2}, \ldots$. Define $\tau'_j$ by switching indices $l$ and $l + j$ in $\tau'$ and considering the first $l$ actions. In particular, $\tau'_j$ gives the sequence of rewards $(Y_{\tau',1}, \ldots Y_{\tau',l-1}, Y_{\tau',l+j})$.

First note that $\tau'_0 = \tau$. By coherence, $(Y_{\tau,1}, \ldots Y_{\tau,l}) = (Y_{\tau',1}, \ldots Y_{\tau',l})$. By exchangeability and coherence, $(Y_{\tau',1}, \ldots Y_{\tau',l-1}, Y_{\tau',l}) \sim (Y_{\tau',1}, \ldots Y_{\tau',l-1}, Y_{\tau',l+j})$. This means that for any $j \geq 0$,

$$\mathbb{E}\left[U \prod_{h=1}^{l} f_h(Y_{\tau,h})\right] = \mathbb{E}\left[U \left(\prod_{h=1}^{l-1} f_h(Y_{\tau',h})\right) f_l(Y_{\tau',l+j})\right]$$

We can then consider the average over all these equations for $j = 0 \to n - 1$ and get that for all $n \geq 1$,

$$\mathbb{E}\left[U \prod_{h=1}^{l} f_h(Y_{\tau,h})\right] = \mathbb{E}\left[U \left(\prod_{h=1}^{l-1} f_h(Y_{\tau',h})\right) \left(\frac{1}{n}\sum_{j=0}^{n-1} f_l(Y_{\tau',l+j})\right)\right]$$

Taking limits and using the dominated convergence theorem, we get that

$$\mathbb{E}\left[U \prod_{h=1}^{l} f_h(Y_{\tau,h})\right] = \mathbb{E}\left[U \left(\prod_{h=1}^{l-1} f_h(Y_{\tau',h})\right) \left(\lim_{n\to\infty}\frac{1}{n}\sum_{j=0}^{n-1} f_l(Y_{\tau',l+j})\right)\right] \tag{1}$$

if the limit on the right side exists.

**Showing that the empirical average is the conditional expectation:** Again, consider a different infinite trajectory $\tau_l = (a_l, a_l, \ldots)$. Again, by coherence, $Y_{\tau_l,l+j} = Y_{\tau',l+j}$ for all $j \geq 0$. From the usual de Finetti theorem, specifically point (i) of remark 12.27 in (Klenke, 2008), we have that $\frac{1}{m}\sum_{j=1}^{m} f_l(Y_{\tau_l,j}) \to \mathbb{E}[f_l(Y_{\tau_l,l}) \mid F]$. We can then observe that by general properties of convergence, the following holds.

$$\mathbb{E}[f_l(Y_{\tau_l,l}) \mid F] = \lim_{m\to\infty}\frac{1}{m}\sum_{j=1}^{m} f_l(Y_{\tau_l,j}) = \lim_{m\to\infty}\frac{1}{m-l}\sum_{j=l}^{m-1} f_l(Y_{\tau_l,j})$$

$$= \lim_{m\to\infty}\frac{1}{m-l}\sum_{j=l}^{m-1} f_l(Y_{\tau',j}) = \lim_{n\to\infty}\frac{1}{n}\sum_{j=0}^{n-1} f_l(Y_{\tau',l+j})$$

We can combine this with equation 1 to get that

$$\mathbb{E}\left[U \prod_{h=1}^{l} f_h(Y_{\tau,h})\right] = \mathbb{E}\left[U \left(\prod_{h=1}^{l-1} f_h(Y_{\tau',h})\right) \mathbb{E}[f_l(Y_{\tau_l,l}) \mid F]\right]$$

Since both $U$ and $\mathbb{E}[f_l(Y_{\tau_l,l}) \mid F]$ are now $F$ measurable, we can apply the induction hypothesis to $\tau'$ truncated at $l-1$ and conclude that

$$\mathbb{E}\left[U \prod_{h=1}^{l} f_h(Y_{\tau,h})\right] = \mathbb{E}\left[U\mathbb{E}[f_l(Y_{\tau_l,l}) \mid F] \prod_{h=1}^{l-1} f_h(Y_{\tau,h})\right] = \mathbb{E}\left[U \left(\prod_{h=1}^{l} \mathbb{E}[f_h(Y_{\tau',h}) \mid F]\right)\right]$$

Thus, the induction step holds and the claim holds for all finite $H$. We discussed at the beginning of the section how this implies conditional independence for $H = \infty$ as well. □

### B.6. A de Finetti theorem for TACDPs

We consider contexts and define contextual decision processes in a manner agnostic to context transitions. That is, the process only carries the data of how rewards are generated from given sequences of contexts and actions, while the context transitions themselves may be generated by a different process.

**Definition 6.** A *transition-agnostic contextual decision process* (TACDP) with action set $\mathcal{A}$ and context set $\mathcal{X}$ is a probability space $(\Omega, \mathcal{G}, \mathbb{P})$ equipped with a family of random maps $\mathcal{F}_H : \Omega \to ((\mathcal{X} \times \mathcal{A})^H \to \mathbb{R}^H)$ for $H \in \mathbb{N}$ as well as a map $\mathcal{F}_\infty : \Omega \to ((\mathcal{X} \times \mathcal{A})^{\mathbb{N}} \to \mathbb{R}^{\mathbb{N}})$.

It is natural to isolate away context dynamics for processes like stochastic contextual bandits, in which actions do not affect the context transitions – in contrast to transition-aware definitions in (Jiang et al., 2016).

**Definition 7.** A TACDP is said to be *coherent* if for any $h \leq k \leq H, H' \in \mathbb{N} \cup \{\infty\}$ and for any two context-action sequences $\tau, \tau'$ of lengths $H$ and $H'$ sharing the same context-action pairs $((x_h, a_h), \dots (x_k, a_k))$ from index $h$ to $k$, with $\mathcal{F}_H(\tau) = (Y_1, \dots Y_H)$ and $\mathcal{F}_{H'}(\tau') = (Y_1', \dots Y_{H'}')$, we have $(Y_h, \dots Y_k) = (Y_h', \dots Y_k')$, viewed as functions of $\Omega$.

**Definition 8.** A stateless decision process is said to be *exchangeable* if for any permutation $\pi : [H] \to [H]$ and $\mathcal{F}_H((x_1, a_1), \dots (x_H, a_H)) = (Y_1, \dots Y_H)$, we have $\mathcal{F}_h((x_{\pi(1)}, a_{\pi(1)}), \dots (x_{\pi(H)}, a_{\pi(H)})) \sim (Y_{\pi(1)}, \dots Y_{\pi(H)})$.

**Definition 9.** A latent contextual bandit is a stateless decision process equipped with a random measure-valued function $F : \Omega \to ((\mathcal{X} \times \mathcal{A}) \to \mathcal{P}(\mathbb{R}))$ so that for any $H$ and context-action sequence $((x_1, a_1), \dots (x_H, a_H))$, the rewards $(Y_1, \dots Y_H) := \mathcal{F}_H((x_1, a_1), \dots (x_H, a_H))$ are independent conditioned on $F$. Moreover, the conditional distribution $\mathcal{L}[Y_h \mid F] = F((x_h, a_h))$ for all $h \leq H$.[10]

**Theorem 6** (De Finetti Theorem for Stateless Decision Processes). *Every exchangeable and coherent TACDP is a latent contextual bandit.*

*Proof.* The proof is verbatim the same as that for Theorem 5 after merely replacing $\mathcal{A}$ with $\mathcal{X} \times \mathcal{A}$ and $a$ with $(x, a)$. □

---

[10]We abuse notation twice here. First, we write $F((x_h, a_h)) := (\omega \mapsto F(\omega)((x_h, a_h)))$. Second, as the regular conditional distribution $\mathcal{L}[Y_h \mid F]$ is a kernel that maps from $\Omega \times \mathcal{B} \to \mathbb{R}$, we view $F((x_h, a_h))$ as its curried map $(\omega, B) \mapsto F((x_h, a_h))(\omega)(B)$. A discussion of issues like measurability and well-definedness is in Appendix B.2.

# C. Proofs for SOLD

Recall that $\mu_\theta := \mathbb{E}[\boldsymbol{\theta}]$ and define $\mu_\beta := \mathbb{E}[\boldsymbol{\beta}] = \mathbf{U}_\star \mu_\theta$. Also recall that $\Lambda := \mathbb{E}[\boldsymbol{\theta}_n \boldsymbol{\theta}_n^\top]$. For a trajectory with index $n$, denote by $\boldsymbol{\beta}_n := \mathbf{U}_\star \boldsymbol{\theta}_n$. Denote by $X_n := [\phi(x_1, a_1), \dots \phi(x_H, a_H)]^\top$. Denote by $\eta_n := [\epsilon_{n,1}, \epsilon_{n,2} \dots \epsilon_{n,H}]^\top$ the vector of subgaussian noises with subgaussian parameter $\sigma^2$. Since rewards are bounded by $R$, we know that $\sigma^2 \leq R^2$. Denote by $X_{n,i}$ and $\eta_{n,i}$ the action matrix and noise vector corresponding to the trajectory halves $\tau_{n,i}$ for $i = 1, 2$. Since rewards are bounded by $R$, $\epsilon_{n,h}$ is subgaussian for all $h$ and so $\eta_{n,i}$ is $\sigma^2$-subgaussian for $i = 1, 2$. Also note that

$$
\begin{aligned}
\hat{\boldsymbol{\beta}}_{n,i} &= (\mu I + X_{n,i}^\top X_{n,i})^{-1} X_{n,i}^\top r_{n,i} \\
&= (\mu I + X_{n,i}^\top X_{n,i})^{-1} X_{n,i}^\top (X_{n,i}\boldsymbol{\beta}_n + \eta_{n,i}) \\
&= (I - \mu(\mu I + X_{n,i}^\top X_{n,i})^{-1})\boldsymbol{\beta}_n + (\mu I + X_{n,i}^\top X_{n,i})^{-1} X_{n,i}^\top \eta_{n,i} \\
&= (I - \mu(\mu I + X_{n,i}^\top X_{n,i})^{-1})\boldsymbol{\beta}_n + (\mu I + X_{n,i}^\top X_{n,i})^{-1} X_{n,i}^\top \eta_{n,i}
\end{aligned}
$$

Note that $\hat{\boldsymbol{\beta}}_{n,1}$ and $\hat{\boldsymbol{\beta}}_{n,2}$ are identically distributed. Now recall that

$$
\mathbf{M}_n = \frac{1}{2}(\hat{\boldsymbol{\beta}}_{n,1}\hat{\boldsymbol{\beta}}_{n,2}^\top + \hat{\boldsymbol{\beta}}_{n,2}\hat{\boldsymbol{\beta}}_{n,1}^\top)
$$

are i.i.d. random matrices. Denote by $\mathbf{D}_{n,i} := (I - \mu(\mu I + X_{n,i}^\top X_{n,i})^{-1})$ and denote by $u_i := (\mu I + X_{n,i}^\top X_{n,i})^{-1} X_{n,i}^\top \eta_{n,i}$.

**Lemma 4.** *If the per-reward noise is $\sigma^2$-subgaussian, then the following inequalities hold*

$$
\|\boldsymbol{M}_n\|_2 \leq R^2 \left(2 + \frac{H}{2\mu}\right)
$$

$$
\|\mathbb{E}[\boldsymbol{M}_n^2]\|_2 \leq R^4 + \frac{\sigma^4(d_A + 1)}{8\mu^2}
$$

*Proof.* We prove various bounds and assemble them.

**Bounding** $\|(\mu I + X_{n,i}^\top X_{n,i})^{-1} X_{n,i}^\top\|_2$: Consider any $X$ with SVD $X = U^\top \Sigma V$. This means that

$$
\begin{aligned}
\|(\mu I + X^\top X)^{-1} X^\top\|_2 &= V^\top (\Sigma^2 + \mu)^{-1} \Sigma U \\
&= \|V^\top (\Sigma^2 + \mu)^{-1} \Sigma\|_2 = \|(\Sigma^2 + \mu)^{-1} \Sigma\|_2 \\
&\leq \max_a \frac{a}{a^2 + \mu} \\
&= \frac{1}{2\sqrt{\mu}}
\end{aligned}
$$

We can now apply this to $X = X_{n,i}$ and conclude that $\|(\mu I + X_{n,i}^\top X_{n,i})^{-1} X_{n,i}^\top\|_2 \leq \frac{1}{2\sqrt{\mu}}$

$u_i$ **are independent,** $\frac{\sigma^2}{4\mu}$**-subgaussian and** $\frac{\sigma\sqrt{H}}{2\sqrt{\mu}}$**-bounded:** We claim that $u_i$ are independent, $\frac{\sigma^2}{4\mu}$-subgaussian and $\|u_i\|_2^2 \leq \frac{\sigma\sqrt{H}}{2\sqrt{\mu}}$. Recall that $\eta_{n,i}$ are $\sigma^2$-subgaussian vectors and $\|(\mu I + X_{n,i}^\top X_{n,i})^{-1} X_{n,i}^\top\|_2 \leq \frac{1}{2\sqrt{\mu}}$. So, we have that $u_i = (\mu I + X_{n,i}^\top X_{n,i})^{-1} X_{n,i}^\top \eta_{n,i}$ is $\frac{\sigma^2}{4\mu}$-subgaussian. Also recall that $u_1$ and $u_2$ are independent since both contexts and reward-noise are generated independently at each timestep. Finally, since $|\epsilon_{n,h}| \leq R$, we also have that $\|\eta_{n,i}\|_2^2 \leq R^2 H$, so $\|u_i\|_2^2 \leq \frac{R^2 H}{4\mu}$ since $\|(\mu I + X_{n,i}^\top X_{n,i})^{-1} X_{n,i}^\top\|_2 \leq \frac{1}{2\sqrt{\mu}}$.

**Bounding** $\|\mathbf{M}_n\|_2$: Note that $\|\mathbf{D}_{n,i}\|_2 \leq 1$, $\|\boldsymbol{\beta}_2\|_n \leq R$ and $\|u_i\|_2 \leq \frac{\sigma\sqrt{H}}{2\sqrt{\mu}}$. So, we have that

$$
\|\mathbf{M}_n\|_2 \leq \left(R + \frac{R\sqrt{H}}{2\sqrt{\mu}}\right)^2 \leq R^2 \left(2 + \frac{H}{2\mu}\right)
$$

**Bounding** $\|\mathbb{E}[\mathbf{M}_n^2]\|_2$: We compute that

$$
\mathbb{E}[\mathbf{M}_n^2] = \frac{1}{4}\Big(\mathbb{E}[\mathbf{D}_{n,1}\boldsymbol{\beta}_n\boldsymbol{\beta}_n^\top \mathbf{D}_{n,1}]\mathbb{E}[\boldsymbol{\beta}_n^\top \mathbf{D}_{n,2}^2 \boldsymbol{\beta}_n] + \mathbb{E}[\mathbf{D}_{n,2}\boldsymbol{\beta}_n\boldsymbol{\beta}_n^\top \mathbf{D}_{n,2}]\mathbb{E}[\boldsymbol{\beta}_n^\top \mathbf{D}_{n,1}^2 \boldsymbol{\beta}_n]
$$

$$+ \mathbb{E}[(\boldsymbol{\beta}_n^\top \mathbf{D}_{n,2}\mathbf{D}_{n,1}\boldsymbol{\beta}_n)\mathbf{D}_{n,1}\boldsymbol{\beta}_n\boldsymbol{\beta}_n^\top \mathbf{D}_{n,2}]] + \mathbb{E}[(\boldsymbol{\beta}_n^\top \mathbf{D}_{n,1}\mathbf{D}_{n,2}\boldsymbol{\beta}_n)\mathbf{D}_{n,2}\boldsymbol{\beta}_n\boldsymbol{\beta}_n^\top \mathbf{D}_{n,1}]]$$

$$\mathbb{E}[\|u_1\|_2^2]\mathbb{E}[u_2 u_2^\top] + \mathbb{E}[\|u_2\|_2^2]\mathbb{E}[u_1 u_1^\top] + \mathbb{E}[u_1 u_2^\top u_1 u_2^\top] + \mathbb{E}[u_2 u_1^\top u_2 u_1^\top]\Big)$$

Now since $\|\mathbf{D}_{n,i}\|_2 \leq 1$ and $\|\beta_n\|_2 \leq R$, the norm of the first four terms is bounded by $R^4$. Now note that

$$\|\mathbb{E}[u_i u_i^\top]\|_2 = \max_{\mathbf{v}, \|\mathbf{v}\|_2 \leq 1} \mathbb{E}[\mathbf{v}^\top u_i u_i^\top \mathbf{v}] = \max_{\mathbf{v}, \|\mathbf{v}\|_2 \leq 1} \mathbb{E}[(u_i^\top \mathbf{v})^2] \leq \frac{\sigma^2}{2\mu}$$

$$\|\mathbb{E}[\|u_i\|_2^2]\|_2 = \text{Tr}(\mathbb{E}[u_i u_i^\top]) \leq \|\mathbb{E}[u_i u_i^\top]\|_2 d_A \leq \frac{\sigma^2 d_A}{2\mu}$$

$$\|\mathbb{E}[u_2 u_1^\top u_2 u_1^\top]\|_2 = \|\mathbb{E}[u_1 u_2^\top u_1 u_2^\top]\|_2 = \max_{\mathbf{v}, \|\mathbf{v}\|_2 \leq 1} \mathbb{E}[\mathbf{v}^\top u_1 u_2^\top u_1 u_2^\top \mathbf{v}]$$

$$\max_{\mathbf{v}, \|\mathbf{v}\|_2 \leq 1} \mathbb{E}[\mathbf{v}^\top u_1 u_2^\top u_1 u_2^\top \mathbf{v}] = \max_{\mathbf{v}, \|\mathbf{v}\|_2 \leq 1} \text{Tr}(\mathbf{v}^\top \mathbb{E}[u_1 u_1^\top]\mathbb{E}[u_2 u_2^\top]\mathbf{v}) = \|\mathbb{E}[u_1 u_1^\top]\mathbb{E}[u_2 u_2^\top]\|_2 \leq \frac{\sigma^4}{4\mu^2}$$

Combining all of these, we get that

$$\|\mathbb{E}[\mathbf{M}_n^2]\|_2 \leq R^4 + \frac{\sigma^4(d_A + 1)}{8\mu^2}$$

$\square$

**Proposition 1** (Confidence Bound for $\overline{\boldsymbol{M}}_N$). *With probability at least $1 - \delta/2$, we have that $\|\overline{\boldsymbol{M}}_N - \mathbb{E}[\boldsymbol{M}_1]\|_2 \leq \Delta_M$ with*

$$\Delta_M := \sqrt{2\left\|\sum_{n=1}^N \boldsymbol{M}_n^2\right\|_2 \frac{\log(4d_A/\delta)}{N}} + 2R^2\left(2 + \frac{H}{2\mu}\right)\left(\frac{2\log(4d_A/\delta)}{N}\right)^{3/4}$$
$$+ 4R^2\left(2 + \frac{H}{2\mu}\right)\frac{\log(4d_A/\delta)}{3N}$$

*Proof.* Now since $\|\mathbf{M}_n^2\|_2 \leq \|\mathbf{M}_n\|_2^2 \leq R^4\left(2 + \frac{H}{4\mu}\right)^2$, we have that $\|\mathbf{M}_n^2 - \mathbb{E}[\mathbf{M}_1^2]\|_2 \leq 2R^4\left(2 + \frac{H}{4\mu}\right)^2$ by the matrix Hoeffding bound, we have that with probability $1 - \delta/2$,

$$\left\|\frac{1}{N}\sum_{n=1}^N \mathbf{M}_n^2 - \mathbb{E}[\mathbf{M}_1^2]\right\|_2 \leq 4R^4\left(2 + \frac{H}{2\mu}\right)^2\sqrt{\frac{\log(d_A/\delta)}{N}}$$

Further, by the matrix Bernstein inequality, we have that with probability $1 - \delta/2$,

$$\left\|\frac{1}{N}\sum_{n=1}^N \mathbf{M}_n - \mathbb{E}[\mathbf{M}_1]\right\|_2 \leq \sqrt{2\|\mathbb{E}[\mathbf{M}_1^2]\|_2\frac{\log(d_A/\delta)}{N}} + 4R^2\left(2 + \frac{H}{2\mu}\right)\frac{\log(d_A/\delta)}{3N}$$

Combining the two results and using a union bound, we get that with probability $1 - \delta/2$

$$\|\overline{\mathbf{M}}_N - \mathbb{E}[\mathbf{M}_1]\|_2 = \left\|\frac{1}{N}\sum_{n=1}^N \mathbf{M}_n - \mathbb{E}[\mathbf{M}_1]\right\|_2$$
$$\leq \sqrt{2\left\|\sum_{n=1}^N \mathbf{M}_n^2\right\|_2 \frac{\log(2d_A/\delta)}{N}} + 2R^2\left(2 + \frac{H}{2\mu}\right)\left(\frac{2\log(2d_A/\delta)}{N}\right)^{3/4}$$
$$+ 4R^2\left(2 + \frac{H}{2\mu}\right)\frac{\log(d_A/\delta)}{3N}$$

$\square$

**Proposition 2** (Confidence Bound for $\overline{\mathbf{D}}_{N,i}$)**.** *With probability* $1 - \delta/4$*, for* $i = 1, 2$*, we have that* $\|\overline{\mathbf{D}}_{N,i} - \mathbb{E}[\boldsymbol{D}_{n,i}]\|_2 = \left\|\frac{1}{N}\sum_{n=1}^{N}\boldsymbol{D}_{n,i} - \mathbb{E}[\boldsymbol{D}_{n,i}]\right\|_2 \leq \Delta_D$ *with*

$$\Delta_D \leq \sqrt{\frac{8\log(4d_A/\delta)}{N}}$$

*Proof.* Since $\|\mathbf{D}_{n,i}\|_2 \leq 1$, this immediately follows by the matrix Hoeffding inequality. $\square$

**Lemma 5** (Confidence Bound for $\overline{\mathbf{D}}_{N,1}^{-1}\overline{\mathbf{M}}_N\overline{\mathbf{D}}_{N,2}^{-1}$)**.** *We have that with probability* $1 - \delta$

$$\|\overline{\boldsymbol{D}}_{N,1}^{-1}\overline{\boldsymbol{M}}_N\overline{\boldsymbol{D}}_{N,2}^{-1} - \mathbb{E}[\boldsymbol{D}_{N,1}]^{-1}\mathbb{E}[\boldsymbol{M}_{N,1}]\mathbb{E}[\boldsymbol{D}_{N,2}]^{-1}\|_2 \leq \left(\frac{B_D^3(2 - B_D\Delta_D)}{(1 - B_D\Delta_D)^2}\right)(R^2 + \Delta_M)\Delta_D$$
$$+ \left(\frac{B_D}{1 - B_D\Delta_D}\right)^2\Delta_M$$

*where* $B_D = \max_{i=1,2}\|\overline{\boldsymbol{D}}_{N,i}^{-1}\|_2$.

*Proof.* For brevity, just for this proof, we define $\mathbf{D}_i = \mathbb{E}[\overline{\mathbf{D}}_{N,i}]$ for $i = 1, 2$ and by $\mathbf{M} := \mathbb{E}[\mathbf{M}_1] = \mathbb{E}[\overline{\mathbf{M}}_N]$. By a union bound, Propositions 1 and 2 hold with probability at least $1 - \delta$. The statements in the rest of this proof thus hold with probability at least $1 - \delta$. Now note that

$$\|\overline{\mathbf{D}}_{N,1}^{-1}\overline{\mathbf{M}}_{N,1}\overline{\mathbf{D}}_{N,2}^{-1} - \mathbf{D}_1^{-1}\mathbf{M}\mathbf{D}_2^{-1}\|_2 \leq \|\overline{\mathbf{D}}_{N,1}^{-1}\|_2\|\overline{\mathbf{M}}_{N,1}\|_2\|\overline{\mathbf{D}}_{N,2}^{-1} - \mathbf{D}_2^{-1}\|_2$$
$$+ \|\overline{\mathbf{D}}_{N,1}^{-1} - \mathbf{D}_1^{-1}\|_2\|\overline{\mathbf{M}}_{N,1}\|_2\|\mathbf{D}^{-1}\|_2$$
$$+ \|\mathbf{D}_1^{-1}\|_2\|\overline{\mathbf{M}}_{N,1} - \mathbf{M}\|_2\|\mathbf{D}_2^{-1}\|_2$$

Now note that by inequality (1.1) from Wei et al. (2005), we have that

$$\|\overline{\mathbf{D}}_{N,i}^{-1} - \mathbf{D}_i^{-1}\|_2 \leq \frac{B_D^2\Delta_D}{1 - B_D\Delta_D}$$

This means that

$$\|\mathbf{D}_i^{-1}\|_2 \leq \|\overline{\mathbf{D}}_{N,i}^{-1}\|_2 + \|\overline{\mathbf{D}}_{N,i}^{-1} - \mathbf{D}_i^{-1}\|_2 \leq \frac{B_D}{1 - B_D\Delta_D}$$

Also, since contexts in both trajectory halves have the same distribution and contexts are generated independently, $\mathbf{M} = \mathbb{E}[\overline{\mathbf{M}}_1] = \mathbf{D}_i\mathbb{E}[\beta_1\beta_1^\top]\mathbf{D}_i$. So, $\|\mathbf{M}\|_2 \leq \|\mathbf{D}_1\|_2\|\mathbf{D}_2\|_2R^2 \leq R^2$. This implies that

$$\|\overline{\mathbf{M}}_{N,1}\|_2 \leq R^2 + \|\overline{\mathbf{M}}_{N,1} - \mathbf{M}\|_2 \leq R^2 + \Delta_M$$

Combining all these with the bound above, we get that

$$\|\overline{\mathbf{D}}_{N,1}^{-1}\overline{\mathbf{M}}_N\overline{\mathbf{D}}_{N,2}^{-1} - \mathbb{E}[\overline{\mathbf{D}}_{N,1}]^{-1}\mathbb{E}[\mathbf{M}_{n,1}]\mathbb{E}[\overline{\mathbf{D}}_{N,2}]^{-1}\|_2 \leq \left(\frac{B_D^3(2 - B_D\Delta_D)}{(1 - B_D\Delta_D)^2}\right)(R^2 + \Delta_M)\Delta_D$$
$$+ \left(\frac{B_D}{1 - B_D\Delta_D}\right)^2\Delta_M$$

$\square$

*Proof.* First note that $\mathbb{E}[\mathbf{D}_{N,1}]^{-1}\mathbb{E}[\mathbf{M}_{N,1}]\mathbb{E}[\mathbf{D}_{N,2}]^{-1} = \mathbf{U}_\star(\Lambda + \mu_\theta\mu_\theta^\top)\mathbf{U}_\star^\top$. Also recall that $\hat{\mathbf{U}}$ is given by the top-$d_K$ eigenvectors for $\mathbb{E}[\overline{\mathbf{D}}_{N,1}^{-1}\mathbf{M}_{N,1}\mathbf{D}_{N,2}^{-1}]$. This means that there is a $d_K \times d_K$ unitary matrix $\mathbf{W}$ such that $\mathbf{U}_\star\mathbf{W}$ forms the eigenvectors for $\mathbf{U}_\star(\Lambda + \mu_\theta\mu_\theta^\top)\mathbf{U}_\star^\top$. This means that by the Davis-Kahan theorem for statisticians (Yu et al., 2014) and Lemma 5, we have that with probability $1 - \delta$

$$\|\hat{\mathbf{U}}\hat{\mathbf{U}}^\top - \mathbf{U}_\star\mathbf{U}_\star^\top\|_2 = \|\hat{\mathbf{U}}\hat{\mathbf{U}}^\top - \mathbf{U}_\star\mathbf{W}\mathbf{W}^\top\mathbf{U}_\star^\top\|_2 = \sqrt{2}(d_K - \|\hat{\mathbf{U}}^\top\mathbf{U}_\star\|_F^2)$$

$$\leq \frac{2\sqrt{2d_K}}{\hat{\lambda}} \|\mathbb{E}[\mathbf{D}_{N,1}]^{-1}\mathbb{E}[\mathbf{M}_{N,1}]\mathbb{E}[\mathbf{D}_{N,2}]^{-1} - \mathbf{D}_{N,1}^{-1}\mathbf{M}_{N,1}\mathbf{D}_{N,2}^{-1}\|_2$$

$$\leq \frac{2\sqrt{2d_K}}{\hat{\lambda}} \left( \left( \frac{B_D^3(2 - B_D\Delta_D)}{(1 - B_D\Delta_D)^2} \right)(R^2 + \Delta_M)\Delta_D + \left( \frac{B_D}{1 - B_D\Delta_D} \right)^2 \Delta_M \right)$$

We thus set $\Delta_{\text{off}}$ to be the value above.

**Bounding** $\Delta_{\text{off}}$**:** Now note that for large enough $N$, $\|\mathbb{E}[\mathbf{D}_{n,1}]^{-1}\|_2\Delta_D = \sqrt{\frac{8\log(d_A/\delta)}{\lambda_A N}} \leq \frac{1}{2}$. In particular, by applying inequality (1.1) from Wei et al. (2005), we have that $B_D \leq 2\|\mathbb{E}[\mathbf{D}_{n,1}]^{-1}\|_2 = \frac{2}{\lambda_A}$. This already gives us the much simpler expression

$$\Delta_{\text{off}} \leq \frac{2\sqrt{2d_K}}{\hat{\lambda}} \left( \frac{64}{\lambda_A^3}(R^2 + \Delta_M)\Delta_D + \frac{16}{\lambda_A^2}\Delta_M \right)$$

Also note that by Lemma 4 and the matrix Hoeffding inequality, we have that

$$\|\frac{1}{N}\sum_{n=1}^{N}\mathbf{M}_n^2\| \leq \|\mathbb{E}[\mathbf{M}_1^2]\| + 4R^4\left(2 + \frac{H}{2\mu}\right)^2\sqrt{\frac{\log(d_A/\delta)}{N}}$$

$$\leq R^4 + \frac{\sigma^4(d_A + 1)}{8\mu^2} + 4R^4\left(2 + \frac{H}{2\mu}\right)^2\sqrt{\frac{\log(d_A/\delta)}{N}}$$

We combine this with Proposition 1 to get that

$$\Delta_M = O\left( \left(R^2 + \frac{\sigma^2\sqrt{d_A}}{\mu}\right)\sqrt{\frac{\log(d_A/\delta)}{N}} \right)$$

Since $\sigma^2 \leq R^2$, we get that

$$\Delta_M = O\left( R^2\sqrt{\frac{d_A\log(d_A/\delta)}{N}} \right) = O(R^2)$$

Also recall from Proposition 2, we get that

$$\Delta_D = \sqrt{\frac{8\log(d_A/\delta)}{N}}$$

Also recall that $\lambda_\theta$ is the minimum eigenvalue of $\frac{1}{R^2}\Lambda$, and so the minimum eigenvalue of $\mathbf{U}_\star(\Lambda + \mu_\theta\mu_\theta^\top)\mathbf{U}_\star^\top$ is larger than $R^2\lambda_\theta$. We then conclude that $\hat{\lambda} \geq R^2\lambda_\theta - 2\Delta_M \geq \frac{\lambda_\theta}{2}$ for large enough $N$. Combining all these, we get that

$$\Delta_{\text{off}} = O\left( \frac{\sqrt{d_K}}{R^2\lambda_\theta}\left(\frac{R^2}{\lambda_A^3} + \frac{R^2}{\lambda_A^2}\sqrt{d_A}\right)\sqrt{\frac{\log(d_A/\delta)}{N}} \right)$$

$$= O\left( \frac{1}{\lambda_\theta\lambda_A^3}\sqrt{\frac{d_Kd_A\log(d_A/\delta)}{N}} \right)$$

$$\square$$

# D. Proofs for LOCAL-UCB

**Theorem 2** (LOCAL-UCB Regret). *Under Assumptions 1 and 2, if $\alpha_{1,t} = R\sqrt{\mu} + CR\sqrt{d_K \log(2T/\delta)}$ and $\alpha_{2,t} = R\sqrt{\mu} + CR\sqrt{d_A \log(2T/\delta)}$ for a universal constant $C$, then with probability at least $1 - \delta$ over offline data and online rewards, LOCAL-UCB has regret $\mathrm{Reg}_T$ bounded by*

$$O\left(\min\left(Rd_A\sqrt{T}, Rd_K\sqrt{T}\left(1 + \frac{1}{\lambda_\theta \lambda_A^3}\sqrt{\frac{d_A T}{d_K N}}\right)\right)\right).$$

*Proof.* Let the true latent state for the given trajectory be $\boldsymbol{\theta}_\star$, so that the reward parameter $\boldsymbol{\beta}_\star = \mathbf{U}_\star \boldsymbol{\theta}_\star$.

**Showing that $\mathbf{U}_\star, \boldsymbol{\beta}_\star$ are in our confidence set:** We check all our constraints:

- Note that with probability $1 - \delta/3$, $\mathbf{U}_\star$ satisfies $\|\hat{\mathbf{U}}^\top \mathbf{U}_\star\|_F \geq \sqrt{d_K - \Delta_{\mathrm{off}}^2/2}$.

- From the standard confidence ellipsoid bound for linear models applied to dimension $d_A$, with probability $1 - \delta/3T$, $\|\hat{\boldsymbol{\beta}}_{2,t} - \boldsymbol{\beta}_\star\|_{\mathbf{V}_t^{-1}} \leq \alpha_{2,t}$ for all $t$.

- Note that $\mathbf{U}_\star \mathbf{U}_\star^\top \boldsymbol{\beta}_\star = \boldsymbol{\beta}_\star$.

- Finally, we apply the standard confidence ellipsoid bound for linear models to dimension $d_K$ instead of $d_A$ with model $r_t = (\phi(x_t, a_t)^\top \mathbf{U}_\star)(\mathbf{U}_\star^\top \boldsymbol{\beta}_\star) + \epsilon_t$. This means that $\|\mathbf{U}_\star^\top \boldsymbol{\beta}_\star - \mathbf{U}_\star^\top \hat{\boldsymbol{\beta}}_{1,t}\|_{\mathbf{V}_{1,t}^{-1}} \leq \alpha_{1,t}$ where $\mathbf{V}_{1,t} = (\mathbf{I}_{d_K} + \sum_{s=1}^{t-1} \mathbf{U}_\star^\top \phi(x_t, a_t)\phi(x_t, a_t)^\top \mathbf{U}_\star)$. Note that since $\mathbf{U}_\star \mathbf{U}_\star^\top = \mathbf{I}_{d_K}$, we have that $\mathbf{V}_{1,t} = (\mathbf{U}_\star^\top \mathbf{V}_t \mathbf{U}_\star)$. So, $\|\mathbf{U}_\star^\top(\boldsymbol{\beta} - \hat{\boldsymbol{\beta}}_{1,t})\|_{(\mathbf{U}_\star^T \mathbf{V}_t \mathbf{U}_\star)^{-1}} \leq \alpha_{1,t}$ holds with probability at least $1 - \delta/3T$ for all $t$.

So, by a union bound over all events, we get that for all actions $a$, the tuple $(a, \boldsymbol{\beta}_\star, \mathbf{U}_\star)$ satisfies our conditions with probability $1 - \delta$.

**Leveraging optimism in low dimension:** From above and by the optimistic design of the algorithm, with probability $1 - \delta$, $\phi(x_t, a_t)^\top \tilde{\boldsymbol{\beta}}_t$ is an upper bound on $\phi(x_t, a)^\top \boldsymbol{\beta}_\star$ for any action $a$. Thus, we have the following regret decomposition with probability at least $1 - \delta$:

$$\mathrm{Reg}_T = \sum_{t=1}^T \phi(x_t, a_t^\star)^\top \boldsymbol{\beta}_\star - \phi(x_t, a_t)^\top \boldsymbol{\beta}_\star$$

$$\leq \sum_{t=1}^T \phi(x_t, a_t)^\top \tilde{\boldsymbol{\beta}}_t - \phi(x_t, a_t)^\top \boldsymbol{\beta}_\star$$

$$\stackrel{(i)}{=} \sum_{t=1}^T \phi(x_t, a_t)^\top \tilde{\mathbf{U}}_t \tilde{\mathbf{U}}_t^\top \tilde{\boldsymbol{\beta}}_t - \phi(x_t, a_t)^\top \mathbf{U}_\star \mathbf{U}_\star^\top \boldsymbol{\beta}_\star$$

$$\leq \sum_{t=1}^T \phi(x_t, a_t)^\top (\tilde{\mathbf{U}}_t \tilde{\mathbf{U}}_t^\top - \mathbf{U}_\star \mathbf{U}_\star^\top)\tilde{\boldsymbol{\beta}}_t + \phi(x_t, a_t)^\top \mathbf{U}_\star \mathbf{U}_\star^\top(\boldsymbol{\beta}_t - \boldsymbol{\beta}_\star)$$

$$\leq RT\|\mathbf{U}_t \tilde{\mathbf{U}}_t^\top - \mathbf{U}_\star \mathbf{U}_\star^\top\|_2 + \sum_{t=1}^T \phi(x_t, a_t)^\top \mathbf{U}_\star \mathbf{U}_\star^\top(\boldsymbol{\beta}_t - \boldsymbol{\beta}_\star)$$

$$\leq RT\Delta_{\mathrm{off}} + \sum_{t=1}^T \phi(x_t, a_t)^\top \mathbf{U}_\star \mathbf{U}_\star^\top(\boldsymbol{\beta}_t - \boldsymbol{\beta}_\star)$$

$$\leq RT\Delta_{\mathrm{off}} + \sum_{t=1}^T \|\phi(x_t, a_t)^\top \mathbf{U}_\star\|_{(\mathbf{U}_\star^\top \mathbf{V}_t \mathbf{U}_\star)^{-1}} \|\mathbf{U}_\star^\top(\boldsymbol{\beta}_t - \boldsymbol{\beta}_\star)\|_{(\mathbf{U}_\star^\top \mathbf{V}_t \mathbf{U}_\star)^{-1}}$$

$$\leq RT\Delta_{\mathrm{off}} + \sum_{t=1}^T \alpha_{2,t}\|\phi(x_t, a_t)^\top \mathbf{U}_\star\|_{(\mathbf{U}_\star^\top \mathbf{V}_t \mathbf{U}_\star)^{-1}}$$

$$\stackrel{(ii)}{=} RT\Delta_{\mathrm{off}} + \sum_{t=1}^T \alpha_{2,t}\|\phi(x_t, a_t)^\top \mathbf{U}_\star\|_{\mathbf{V}_{1,t}^{-1}}$$

$$\overset{(iii)}{\leq} RT\Delta_{\text{off}} + O\left(d_K \sqrt{T \log\left(T/\delta\right)}\right)$$

$$\overset{(iv)}{=} O\left(Rd_K \sqrt{T \log\left(T/\delta\right)} + \frac{R\lambda_D^3}{\lambda_{\min}} \sqrt{\frac{T^2 d_K d_A \log(d_A/\delta)}{N}}\right)$$

$$= O\left(Rd_K \sqrt{T \log\left(T/\delta\right)} \left(1 + \frac{\lambda_D^3}{\lambda_{\min}} \sqrt{\frac{T d_A \log(d_A)}{d_K N}}\right)\right)$$

$$= \tilde{O}\left(Rd_K \sqrt{T} \left(1 + \sqrt{\frac{T d_A}{d_K N}}\right)\right)$$

where $(i)$ holds since $\tilde{\mathbf{U}}_t \tilde{\mathbf{U}}_t^\top \tilde{\boldsymbol{\beta}}_t = \tilde{\boldsymbol{\beta}}_t$ and $\mathbf{U}_\star \mathbf{U}_\star^\top \boldsymbol{\beta}_\star = \boldsymbol{\beta}_\star$ $(ii)$ holds since $\mathbf{V}_{1,t}^{-1} = (\mathbf{U}_\star^\top \mathbf{V}_t \mathbf{U}_\star)^{-1}$, $(iii)$ holds by the usual proof of LinUCB applied to dimension $d_K$, and $(iv)$ holds by Theorem 1.

**Leveraging optimism in high dimension:** This is merely the proof of LinUCB. From above and by the optimistic design of the algorithm, with probability $1 - \delta$, $\phi(x_t, a_t)^\top \tilde{\boldsymbol{\beta}}_t$ is an upper bound on $\phi(x_t, a)^\top \boldsymbol{\beta}_\star$ for any action $a$. Thus, we have the following regret decomposition with probability at least $1 - \delta$:

$$\text{Reg}_T = \sum_{t=1}^T \phi(x_t, a_t^\star)^\top \boldsymbol{\beta}_\star - \phi(x_t, a_t)^\top \tilde{\boldsymbol{\beta}}_\star$$

$$\leq \sum_{t=1}^T \phi(x_t, a_t)^\top \tilde{\boldsymbol{\beta}}_t - \phi(x_t, a_t)^\top \boldsymbol{\beta}_\star$$

$$= \sum_{t=1}^T \|\phi(x_t, a_t)^\top\|_{\mathbf{V}_t^{-1}} \|\tilde{\boldsymbol{\beta}}_t - \boldsymbol{\beta}_\star\|_{\mathbf{V}_t}$$

$$= O(Rd_A \sqrt{T \log(T/\delta)})$$

$$= \tilde{O}(Rd_A \sqrt{T})$$

where the last line follows from the standard regret bound for LinUCB applied to dimension $d_A$.

Combining the two bounds, we have our result.

$$\text{Reg}_T = O\left(\min\left(Rd_A \sqrt{T \log(T/\delta)}, Rd_K \sqrt{T \log\left(T/\delta\right)} \left(1 + \frac{\lambda_D^3}{\lambda_{\min}} \sqrt{\frac{T d_A \log(d_A)}{d_K N}}\right)\right)\right)$$

$$= \tilde{O}\left(\min\left(Rd_A \sqrt{T}, Rd_K \sqrt{T} \left(1 + \sqrt{\frac{T d_A}{d_K N}}\right)\right)\right)$$

$\square$

# E. Proofs for ProBALL-UCB

## E.1. Confidence bound for the low-dimensional reward parameter

**Lemma 6** (Confidence Bound for $\hat{\boldsymbol{\beta}}_{1,t}$). *If for all timesteps $t$, $\phi(x_t, a_t)$ lies in the span of $\hat{\boldsymbol{U}}$, we have that for a universal constant $C$,*

$$\left\| \hat{\boldsymbol{U}}^\top \hat{\boldsymbol{\beta}}_{1,t} - \hat{\boldsymbol{U}}^\top \boldsymbol{\beta}_\star \right\|_{\boldsymbol{V}_{1,t}^{-1}} \le R\sqrt{\mu} + CR\sqrt{d_A \log(t/\delta)}$$

*Otherwise, we have that*

$$\left\| \hat{\boldsymbol{U}}^\top \hat{\boldsymbol{\beta}}_{1,t} - \hat{\boldsymbol{U}}^\top \boldsymbol{\beta}_\star \right\|_{\boldsymbol{V}_{1,t}^{-1}} \le R\sqrt{\mu} + R\kappa_t + \Delta_{\text{off}} CR\sqrt{d_A \log(t/\delta)}$$

*where $\kappa_t = \|\hat{\boldsymbol{U}}^\top \boldsymbol{X}_t^\top \boldsymbol{X}_t\|_{(\hat{\boldsymbol{U}}^\top \boldsymbol{V}_t \hat{\boldsymbol{U}})^{-1}}$, with notation $\|\boldsymbol{A}\|_{\boldsymbol{C}} := \sqrt{\|\boldsymbol{A}^\top \boldsymbol{C} \boldsymbol{A}\|_2}$*

*Proof.* For brevity, in this proof we will denote by $\mathbf{X}_t := [\phi(x_1, a_1), \dots \phi(x_{t-1}, a_{t-1})]^\top$, which is a $t \times d_A$ matrix. Recall that $\mathbf{V}_t = \mu \mathbf{I}_{d_A} + \mathbf{X}_t^\top \mathbf{X}_t$. We will also denote $\mathbf{V}_{1,t} = \hat{\mathbf{U}}^\top \mathbf{V}_t \hat{\mathbf{U}}$. Note that the vector of rewards is given by $\mathbf{X}_t \boldsymbol{\beta}_\star + \eta_t$, where $\eta_t$ is a random vector of $t$ independent $R^2$-subgaussian entries. So, $\mathbf{b}_t = \mathbf{X}_t^\top (\mathbf{X}_t \boldsymbol{\beta}_\star + \eta_t)$. Also define the notation $\Delta_U := \hat{\mathbf{U}} \hat{\mathbf{U}}^\top - \mathbf{U}_\star \mathbf{U}_\star^\top$.

**If all actions lie in the span of $\hat{\mathbf{U}}$.** Note that since all actions taken lie in the span of $\hat{\mathbf{U}}$, $\mathbf{X}_t = \mathbf{X}_t \hat{\mathbf{U}} \hat{\mathbf{U}}^\top$. *This is the key observation.* Now note that

$$\begin{aligned}
\hat{\boldsymbol{\beta}}_{1,t} &= \hat{\mathbf{U}} \mathbf{V}_{1,t}^{-1} \hat{\mathbf{U}}^\top \mathbf{X}_t^\top (\mathbf{X}_t \boldsymbol{\beta}_\star + \eta_t) \\
&= \hat{\mathbf{U}} \mathbf{V}_{1,t}^{-1} \hat{\mathbf{U}}^\top \mathbf{X}_t^\top \mathbf{X}_t \boldsymbol{\beta}_\star + \hat{\mathbf{U}} \mathbf{V}_{1,t}^{-1} \hat{\mathbf{U}}^\top \mathbf{X}_t^\top \eta_t \\
&= \hat{\mathbf{U}} \mathbf{V}_{1,t}^{-1} \hat{\mathbf{U}}^\top \mathbf{X}_t^\top \mathbf{X}_t \hat{\mathbf{U}} \hat{\mathbf{U}}^\top \boldsymbol{\beta}_\star + \hat{\mathbf{U}} \mathbf{V}_{1,t}^{-1} \hat{\mathbf{U}}^\top \mathbf{X}_t^\top \eta_t \\
&= \hat{\mathbf{U}} \mathbf{V}_{1,t}^{-1} (\mathbf{V}_{1,t} - \mu \mathbf{I}_{d_K}) \hat{\mathbf{U}}^\top \boldsymbol{\beta}_\star + \hat{\mathbf{U}} \mathbf{V}_{1,t}^{-1} \hat{\mathbf{U}}^\top \mathbf{X}_t^\top \eta_t \\
&= \hat{\mathbf{U}} \hat{\mathbf{U}}^\top \boldsymbol{\beta}_\star - \mu \hat{\mathbf{U}} \mathbf{V}_{1,t}^{-1} \hat{\mathbf{U}}^\top \boldsymbol{\beta}_\star + \hat{\mathbf{U}} \mathbf{V}_{1,t}^{-1} \hat{\mathbf{U}}^\top \mathbf{X}_t^\top \eta_t
\end{aligned}$$

The rest of the proof is similar to Theorem 2 in (Abbasi-Yadkori et al., 2011). We first note that for any $x$, we have that

$$\begin{aligned}
x^\top (\hat{\boldsymbol{\beta}}_{1,t} - \hat{\mathbf{U}} \hat{\mathbf{U}}^\top \boldsymbol{\beta}_\star) &\le \left| (x^\top \hat{\mathbf{U}}) \mathbf{V}_{1,t}^{-1} (-\mu \hat{\mathbf{U}}^\top \boldsymbol{\beta}_\star + \hat{\mathbf{U}}^\top \mathbf{X}_t^\top \eta_t) \right| \\
&\le \|\hat{\mathbf{U}}^\top x\|_{\mathbf{V}_{1,t}^{-1}} \left\| (-\mu \hat{\mathbf{U}}^\top \boldsymbol{\beta}_\star + \hat{\mathbf{U}}^\top \mathbf{X}_t^\top \eta_t) \right\|_{\mathbf{V}_{1,t}^{-1}} \\
&\le \|\hat{\mathbf{U}}^\top x\|_{\mathbf{V}_{1,t}^{-1}} \left( \mu \|\hat{\mathbf{U}}^\top \boldsymbol{\beta}_\star\|_{\mathbf{V}_{1,t}^{-1}} + \|\hat{\mathbf{U}}^\top \mathbf{X}_t^\top \eta_t\|_{\mathbf{V}_{1,t}^{-1}} \right)
\end{aligned}$$

Now note that $\mu \|\hat{\mathbf{U}}^\top \boldsymbol{\beta}_\star\|_{\mathbf{V}_{1,t}^{-1}} \le \|\hat{\mathbf{U}}^\top \boldsymbol{\beta}_\star\|_2 \sqrt{\mu} \le R\sqrt{\mu}$ and by the self normalized martingale concentration inequality from (Abbasi-Yadkori et al., 2011) applied to $d_K$ dimensional vectors $\hat{\mathbf{U}}^\top \mathbf{X}_t$, we have that with probability at least $1 - \delta$, $\|\hat{\mathbf{U}}^\top \mathbf{X}_t^\top \eta_t\|_{\mathbf{V}_{1,t}^{-1}} \le CR\sqrt{d_A \log(t/\delta)}$ for some universal constant $C$. So we have with probability at least $1 - \delta$ that

$$= \left\| \hat{\mathbf{U}}(\hat{\boldsymbol{\beta}}_{1,t} - \hat{\mathbf{U}} \hat{\mathbf{U}}^\top \boldsymbol{\beta}_\star) \right\|_{\mathbf{V}_{1,t}^{-1}}^2 \le \left\| \hat{\boldsymbol{\beta}}_{1,t} - \hat{\mathbf{U}} \hat{\mathbf{U}}^\top \boldsymbol{\beta}_\star \right\|_{\mathbf{V}_{1,t}^{-1}} \left( R\sqrt{\mu} + CR\sqrt{d_A \log(t/\delta)} \right)$$

This means that

$$\begin{aligned}
\left\| \hat{\mathbf{U}}^\top \hat{\boldsymbol{\beta}}_{1,t} - \hat{\mathbf{U}}^\top \boldsymbol{\beta}_\star \right\|_{\mathbf{V}_{1,t}^{-1}} &= \left\| \hat{\boldsymbol{\beta}}_{1,t} - \hat{\mathbf{U}} \hat{\mathbf{U}}^\top \boldsymbol{\beta}_\star \right\|_{\mathbf{V}_{1,t}^{-1}} \\
&\le R\sqrt{\mu} + CR\sqrt{d_A \log(t/\delta)}
\end{aligned}$$

**If all actions don't lie in the span of $\hat{\mathbf{U}}$.** This time note that

$$
\begin{aligned}
\hat{\boldsymbol{\beta}}_{1,t} &= \hat{\mathbf{U}}\mathbf{V}_{1,t}^{-1}\hat{\mathbf{U}}^{\top}\mathbf{X}_t^{\top}\left(\mathbf{X}_t\boldsymbol{\beta}_{\star}+\eta_t\right) \\
&= \hat{\mathbf{U}}\mathbf{V}_{1,t}^{-1}\hat{\mathbf{U}}^{\top}\mathbf{X}_t^{\top}\mathbf{X}_t\boldsymbol{\beta}_{\star}+\hat{\mathbf{U}}\mathbf{V}_{1,t}^{-1}\hat{\mathbf{U}}^{\top}\mathbf{X}_t^{\top}\eta_t \\
&= \hat{\mathbf{U}}\mathbf{V}_{1,t}^{-1}\hat{\mathbf{U}}^{\top}\mathbf{X}_t^{\top}\mathbf{X}_t\hat{\mathbf{U}}_{\star}\mathbf{U}_{\star}^{\top}\boldsymbol{\beta}_{\star}+\hat{\mathbf{U}}\mathbf{V}_{1,t}^{-1}\mathbf{U}^{\top}\mathbf{X}_t^{\top}\eta_t \\
&= \hat{\mathbf{U}}\mathbf{V}_{1,t}^{-1}\hat{\mathbf{U}}^{\top}\mathbf{X}_t^{\top}\mathbf{X}_t\hat{\mathbf{U}}\hat{\mathbf{U}}^{\top}\boldsymbol{\beta}_{\star}+\hat{\mathbf{U}}\mathbf{V}_{1,t}^{-1}\hat{\mathbf{U}}^{\top}\mathbf{X}_t^{\top}\mathbf{X}_t\Delta_U\boldsymbol{\beta}_{\star}+\hat{\mathbf{U}}\mathbf{V}_{1,t}^{-1}\hat{\mathbf{U}}^{\top}\mathbf{X}_t^{\top}\eta_t \\
&= \hat{\mathbf{U}}\mathbf{V}_{1,t}^{-1}(\mathbf{V}_{1,t}-\mu\mathbf{I}_{d_K})\hat{\mathbf{U}}^{\top}\boldsymbol{\beta}_{\star}+\hat{\mathbf{U}}\mathbf{V}_{1,t}^{-1}\hat{\mathbf{U}}^{\top}\mathbf{X}_t^{\top}\mathbf{X}_t\Delta_U\boldsymbol{\beta}_{\star}+\hat{\mathbf{U}}\mathbf{V}_{1,t}^{-1}\hat{\mathbf{U}}^{\top}\mathbf{X}_t^{\top}\eta_t \\
&= \hat{\mathbf{U}}\mathbf{U}^{\top}\boldsymbol{\beta}_{\star}-\mu\hat{\mathbf{U}}\mathbf{V}_{1,t}^{-1}\hat{\mathbf{U}}^{\top}\boldsymbol{\beta}_{\star}+\hat{\mathbf{U}}\mathbf{V}_{1,t}^{-1}\hat{\mathbf{U}}^{\top}\mathbf{X}_t^{\top}\mathbf{X}_t\Delta_U\boldsymbol{\beta}_{\star}+\hat{\mathbf{U}}\mathbf{V}_{1,t}^{-1}\hat{\mathbf{U}}^{\top}\mathbf{X}_t^{\top}\eta_t
\end{aligned}
$$

The rest of the proof is similar to Theorem 2 in (Abbasi-Yadkori et al., 2011). We first note that for any $x$, we have that

$$
\begin{aligned}
x^{\top}(\hat{\boldsymbol{\beta}}_{1,t}-\hat{\mathbf{U}}\hat{\mathbf{U}}^{\top}\boldsymbol{\beta}_{\star}) &\leq \left|(x^{\top}\hat{\mathbf{U}})\mathbf{V}_{1,t}^{-1}(-\mu\hat{\mathbf{U}}^{\top}\boldsymbol{\beta}_{\star}+\hat{\mathbf{U}}^{\top}\mathbf{X}_t^{\top}\mathbf{X}_t\Delta_U\boldsymbol{\beta}_{\star}+\hat{\mathbf{U}}^{\top}\mathbf{X}_t^{\top}\eta_t)\right| \\
&\leq \|\hat{\mathbf{U}}^{\top}x\|_{\mathbf{V}_{1,t}^{-1}}\left\|(-\mu\hat{\mathbf{U}}^{\top}\boldsymbol{\beta}_{\star}+\hat{\mathbf{U}}^{\top}\mathbf{X}_t^{\top}\mathbf{X}_t\Delta_U\boldsymbol{\beta}_{\star}+\hat{\mathbf{U}}^{\top}\mathbf{X}_t^{\top}\eta_t)\right\|_{\mathbf{V}_{1,t}^{-1}} \\
&\leq \|\hat{\mathbf{U}}^{\top}x\|_{\mathbf{V}_{1,t}^{-1}}\left(\mu\|\hat{\mathbf{U}}^{\top}\boldsymbol{\beta}_{\star}\|_{\mathbf{V}_{1,t}^{-1}}+\|\hat{\mathbf{U}}^{\top}\mathbf{X}_t^{\top}\mathbf{X}_t\Delta_U\boldsymbol{\beta}_{\star}\|_{\mathbf{V}_{1,t}^{-1}}+\|\hat{\mathbf{U}}^{\top}\mathbf{X}_t^{\top}\eta_t\|_{\mathbf{V}_{1,t}^{-1}}\right) \\
&\stackrel{(i)}{\leq} \|\hat{\mathbf{U}}^{\top}x\|_{\mathbf{V}_{1,t}^{-1}}\left(\mu\|\hat{\mathbf{U}}^{\top}\boldsymbol{\beta}_{\star}\|_{\mathbf{V}_{1,t}^{-1}}+\|\Delta_U\boldsymbol{\beta}_{\star}\|_2\|\hat{\mathbf{U}}^{\top}\mathbf{X}_t^{\top}\mathbf{X}_t\|_{\mathbf{V}_{1,t}^{-1}}+\|\hat{\mathbf{U}}^{\top}\mathbf{X}_t^{\top}\eta_t\|_{\mathbf{V}_{1,t}^{-1}}\right) \\
&\leq \|\hat{\mathbf{U}}^{\top}x\|_{\mathbf{V}_{1,t}^{-1}}\left(\mu\|\hat{\mathbf{U}}^{\top}\boldsymbol{\beta}_{\star}\|_{\mathbf{V}_{1,t}^{-1}}+R\|\Delta_U\|_2\|\hat{\mathbf{U}}^{\top}\mathbf{X}_t^{\top}\mathbf{X}_t\|_{\mathbf{V}_{1,t}^{-1}}+\|\hat{\mathbf{U}}^{\top}\mathbf{X}_t^{\top}\eta_t\|_{\mathbf{V}_{1,t}^{-1}}\right) \\
&\leq \|\hat{\mathbf{U}}^{\top}x\|_{\mathbf{V}_{1,t}^{-1}}\left(\mu\|\hat{\mathbf{U}}^{\top}\boldsymbol{\beta}_{\star}\|_{\mathbf{V}_{1,t}^{-1}}+R\Delta_{\text{off}}\|\hat{\mathbf{U}}^{\top}\mathbf{X}_t^{\top}\mathbf{X}_t\|_{\mathbf{V}_{1,t}^{-1}}+\|\hat{\mathbf{U}}^{\top}\mathbf{X}_t^{\top}\eta_t\|_{\mathbf{V}_{1,t}^{-1}}\right)
\end{aligned}
$$

Here, $(i)$ holds because for any matrices $\mathbf{A},\mathbf{C}$ and any vector $\mathbf{v}$, we have that $\|\mathbf{A}\mathbf{v}\|_{\mathbf{C}} = \sqrt{\mathbf{v}^{\top}\mathbf{A}^{\top}\mathbf{C}\mathbf{A}\mathbf{v}} \leq \|\mathbf{v}\|_2\sqrt{\|\mathbf{A}^{\top}\mathbf{C}\mathbf{A}\|_2} = \|\mathbf{v}\|_2\|\mathbf{A}\|_{\mathbf{C}}$, where we recall that $\|\mathbf{A}\|_{\mathbf{C}} := \sqrt{\|\mathbf{A}^{\top}\mathbf{C}\mathbf{A}\|_2}$.

Now note that $\mu\|\hat{\mathbf{U}}^{\top}\boldsymbol{\beta}_{\star}\|_{\mathbf{V}_{1,t}^{-1}} \leq \|\hat{\mathbf{U}}^{\top}\boldsymbol{\beta}_{\star}\|_2\sqrt{\mu} \leq R\sqrt{\mu}$ and by the self normalized martingale concentration inequality from (Abbasi-Yadkori et al., 2011) applied to $d_K$ dimensional vectors $\hat{\mathbf{U}}^{\top}\mathbf{X}_t$, we have that with probability at least $1-\delta$, $\|\hat{\mathbf{U}}^{\top}\mathbf{X}_t^{\top}\eta_t\|_{\mathbf{V}_{1,t}^{-1}} \leq CR\sqrt{d_A\log(t/\delta)}$ for some universal constant $C$. Also recall that $\|\hat{\mathbf{U}}^{\top}\mathbf{X}_t^{\top}\mathbf{X}_t\|_{\mathbf{V}_{1,t}^{-1}} = \kappa_t$. So we have with probability at least $1-\delta$ that

$$
\left\|\hat{\mathbf{U}}(\hat{\boldsymbol{\beta}}_{1,t}-\hat{\mathbf{U}}\hat{\mathbf{U}}^{\top}\boldsymbol{\beta}_{\star})\right\|_{\mathbf{V}_{1,t}^{-1}}^2 \leq \left\|\hat{\mathbf{U}}^{\top}(\hat{\boldsymbol{\beta}}_{1,t}-\hat{\mathbf{U}}\hat{\mathbf{U}}^{\top}\boldsymbol{\beta}_{\star})\right\|_{\mathbf{V}_{1,t}^{-1}}\left(R\sqrt{\mu}+R\Delta_{\text{off}}\kappa_t+CR\sqrt{d_A\log(t/\delta)}\right)
$$

This means that

$$
\begin{aligned}
\left\|\hat{\mathbf{U}}^{\top}\hat{\boldsymbol{\beta}}_{1,t}-\hat{\mathbf{U}}^{\top}\boldsymbol{\beta}_{\star}\right\|_{\mathbf{V}_{1,t}^{-1}} &= \left\|\hat{\mathbf{U}}(\hat{\boldsymbol{\beta}}_{1,t}-\hat{\mathbf{U}}\hat{\mathbf{U}}^{\top}\boldsymbol{\beta}_{\star})\right\|_{\mathbf{V}_{1,t}^{-1}} \\
&\leq R\sqrt{\mu}+R\Delta_{\text{off}}\kappa_t+CR\sqrt{d_A\log(t/\delta)}
\end{aligned}
$$

Note that $\kappa_t = \left\|\sum_{s=1}^{t-1}\hat{\mathbf{U}}^{\top}\phi(x_s,a_s)\phi(x_s,a_s)^{\top}\right\|_{\mathbf{V}_{1,t}^{-1}} = \frac{1}{\sqrt{\mu}}\left\|\sum_{s=1}^{t-1}\hat{\mathbf{U}}^{\top}\phi(x_s,a_s)\phi(x_s,a_s)^{\top}\right\|_2 = O(t)$, but this is a worst case bound. $\qquad\square$

We also state a lemma, borrowed from the standard proof of LinUCB regret.

**Lemma 7.** *For any sequence of actions and contexts $x_t, a_t$ and $\boldsymbol{V}_t = \mu\boldsymbol{I}+\sum_{s=1}^{t-1}\phi(x_t,a_t)\phi(x_t,a_t)^{\top}$, we have that*

$$
\sum_{t=1}^{T}\min\left(\|\phi(x_t,a_t)\|_{\boldsymbol{V}_t^{-1}}^2,1\right) = O(\sqrt{d_A})
$$

$$
\sum_{t=1}^{T}\min\left(\|\hat{\boldsymbol{U}}^{\top}\phi(x_t,a_t)\|_{(\hat{\boldsymbol{U}}^{\top}\boldsymbol{V}_t\hat{\boldsymbol{U}})^{-1}}^2,1\right) = O(\sqrt{d_K})
$$

*Proof.* This follows immediately from Lemma 11 of (Abbasi-Yadkori et al., 2011). □

### E.2. Proof of the theorem

**Theorem 4** (Regret for ProBALL-UCB). *Let* $\alpha_{1,t} = R\sqrt{\mu} + \tau' R\Delta_{\text{off}}\kappa_t + CR\sqrt{d_K \log(T/\delta)}$ *and let* $\alpha_{2,t} = R\sqrt{\mu} + CR\sqrt{d_A \log(T/\delta)}$. *Let* $S$ *be the first timestep when Algorithm 3 does not play Line 6 and let* $S = T$ *if no such timestep exists. For* $\tau = \tau' = 1$ *we have that*

$$\text{Reg}_T = \widetilde{O}\left(\min\left(\text{Reg}_{on,T}, \text{Reg}_{hyb,T}\right)\right).$$

*where* $\text{Reg}_{on,T} = Rd_A\sqrt{T}$ *and* $\text{Reg}_{hyb,T}$ *is defined as*

$$Rd_K\sqrt{T}\left(1 + \frac{1}{\lambda_A^3 \lambda_\theta}\left(\sqrt{\frac{d_A T}{d_K N}} + \sqrt{\frac{d_A}{SN}\sum_{t=1}^{S}\kappa_t^2}\right)\right).$$

*In the worst case,* $\kappa_t = O(t)$ *and so* $\frac{1}{S}\sum_{t=1}^{S}\kappa_t^2 = O(T^2)$, *but if all features* $\phi(x_t, a_t)$ *lie in the span of* $\hat{U}$ *for* $t \le S$, *then* $\frac{1}{S}\sum_{t=1}^{S}\kappa_t^2 = O(T)$.

*Proof.* We will first show that our bonuses give optimistic estimates of the true reward whp and then leverage the optimism.

**Showing optimism when** $\tau\Delta_{\text{off}}\sqrt{T} \le d_A$: In this case, we are inside the "if" statement and are running a projected and modified version of LinUCB. In that case, for all timesteps the projected version is run, all features will lie in the span of $\hat{U}$. This is because the maximization problem is given by

$$a_t = \arg\max_{a, \|\phi(x_t,a)\|_2 \le 1} \phi^\top \hat{U}\hat{U}^\top \hat{\beta}_{1,t} + \|\hat{U}^\top \phi(x_t,a)\|_{(\hat{U}^\top V_{1,t}\hat{U})^{-1}}$$

If our features are isotropic, then this is only maximized by a feature in the span of $\hat{U}$. So, the conditions of the first bound in Lemma 6 are fulfilled. We will denote $V_{1,t} = \hat{U}^\top V_t\hat{U}$. We have that with probability $1 - \delta/2$, for all $x, a, t$, the following holds.

$$
\begin{aligned}
|\phi(x,a)^\top \beta_\star - \phi(x,a)^\top \hat{U}\hat{U}^\top \hat{\beta}_{1,t}| &\le |\phi(x,a)^\top \hat{U}(\hat{U}^\top \beta_\star - \hat{U}^\top \hat{\beta}_{1,t})| + |\phi(x,a)^\top (U_\star U_\star^\top - \hat{U}\hat{U}^\top)\beta_\star| \\
&\le \|\hat{U}^\top \phi(x,a)\|_{V_{1,t}^{-1}}\|\hat{U}^\top \beta_\star - \hat{U}^\top \hat{\beta}_{1,t}\|_{V_{1,t}^{-1}} + R\|U_\star U_\star^\top - \hat{U}\hat{U}^\top\|_2 \\
&\le \alpha_{1,t}\|\hat{U}^\top \phi(x,a)\|_{V_{1,t}^{-1}} + R\Delta_{\text{off}}
\end{aligned}
$$

This shows that with probability at least $1 - \delta$ and for all $x, a, t$,

$$\phi(x,a)^\top \beta_\star \le \phi(x,a)^\top \hat{U}\hat{U}^\top \hat{\beta}_{1,t} + \alpha_{1,t}\|\hat{U}^\top \phi(x,a)\|_{V_{1,t}^{-1}} + R\Delta_{\text{off}}$$

This implies that with probability at least $1 - \delta$, for any action $a$,

$$
\begin{aligned}
\phi(x_t,a)^\top \beta_\star &\le \max_a \phi(x_t,a)^\top \hat{U}\hat{U}^\top \hat{\beta}_{1,t} + \alpha_{1,t}\|\hat{U}^\top \phi(x_t,a)\|_{V_{1,t}^{-1}} + R\Delta_{\text{off}} \\
&= \phi(x_t,a_t)^\top \hat{U}\hat{U}^\top \hat{\beta}_{1,t} + \alpha_{1,t}\|\hat{U}^\top \phi(x_t,a_t)\|_{V_{1,t}^{-1}} + R\Delta_{\text{off}}
\end{aligned}
$$

**Leveraging optimism when** $\tau\Delta_{\text{off}}\sqrt{T} \le d_A$: Consider the standard regret decomposition, which holds with probability $1 - \delta$:

$$
\begin{aligned}
\text{Reg}_T &= \sum_{t=1}^{T} \phi(x_t, a_t^\star)^\top \beta_\star - \phi(x_t, a_t)^\top \beta_\star \\
&\le \sum_{t=1}^{T} \phi(x_t, a_t)^\top \hat{U}\hat{U}^\top \hat{\beta}_{1,t} - \phi(x_t, a_t)^\top \beta_\star + \alpha_{1,t}\|\hat{U}^\top \phi(x_t, a_t)\|_{V_{1,t}^{-1}} + R\Delta_{\text{off}}
\end{aligned}
$$

$$\leq \sum_{t=1}^{T} 2\alpha_{1,t}\|\hat{\mathbf{U}}^\top \phi(x_t, a_t)\|_{\mathbf{V}_{1,t}^{-1}} + 2R\Delta_{\text{off}} \tag{2}$$

$$\overset{(i)}{\leq} \sum_{t=1}^{T} 2\alpha'_{1,t} \min\left(\|\hat{\mathbf{U}}^\top \phi(x_t, a_t)\|_{\mathbf{V}_{1,t}^{-1}}, 1\right) + 2R\Delta_{\text{off}}$$

$$\leq \sum_{t=1}^{T} 2(R\sqrt{\mu} + C\sqrt{d_K \log(T/\delta)}) \min\left(\|\hat{\mathbf{U}}^\top \phi(x_t, a_t)\|_{\mathbf{V}_{1,t}^{-1}}, 1\right) + 2R\Delta_{\text{off}}$$

$$+ \sum_{t=1}^{T} 2R\Delta_{\text{off}}\kappa_t \min\left(\|\hat{\mathbf{U}}^\top \phi(x_t, a_t)\|_{\mathbf{V}_{1,t}^{-1}}, 1\right)$$

$$\overset{(ii)}{\leq} 2(R\sqrt{\mu} + C\sqrt{d_K \log(T/\delta)})\sqrt{\sum_{t=1}^{T} \min\left(\|\hat{\mathbf{U}}^\top \phi(x_t, a_t)\|_{\mathbf{V}_{1,t}^{-1}}^2, 1\right)} + 2R\Delta_{\text{off}}T$$

$$+ 2R\Delta_{\text{off}}\sqrt{\sum_{t=1}^{T}\kappa_t^2}\sqrt{\sum_{t=1}^{T} \min\left(\|\hat{\mathbf{U}}^\top \phi(x_t, a_t)\|_{\mathbf{V}_{1,t}^{-1}}^2, 1\right)}$$

$$\overset{(iii)}{\leq} O(d_K\sqrt{T\log(T/\delta)}) + 2R\Delta_{\text{off}}T + 2R\Delta_{\text{off}}\sqrt{\frac{\sum_{t=1}^{T}\kappa_t^2}{T}}\sqrt{d_K T} \tag{3}$$

$$= O\left(Rd_K\sqrt{T\log(T/\delta)}\left(1 + \Delta_{\text{off}}\left(\frac{\sqrt{T}}{d_K} + \sqrt{\frac{1}{Td_K}\sum_{t=1}^{T}\kappa_t^2}\right)\right)\right)$$

where $(i)$ holds since $\phi(x_t, a_t^\star)^\top \boldsymbol{\beta}_\star - \phi(x_t, a_t)^\top \boldsymbol{\beta}_\star \leq 2R$, $(ii)$ holds by the Cauchy Schwarz inequality and $(iii)$ holds by Lemma 7. So, we have that

$$\text{Reg}_T = O\left(Rd_K\sqrt{T\log(T/\delta)}\left(1 + \frac{1}{\lambda_A^3\lambda_\theta}\sqrt{\frac{d_A\log(d_A)}{Nd_K}}\left(\sqrt{T} + \sqrt{\frac{d_K}{T}\sum_{t=1}^{S}\kappa_t^2}\right)\right)\right)$$

$$= \widetilde{O}\left(Rd_K\sqrt{T}\left(1 + \frac{1}{\lambda_A^3\lambda_\theta}\left(\sqrt{\frac{d_AT}{Nd_K}} + \sqrt{\frac{d_A}{TN}\sum_{t=1}^{T}\kappa_t^2}\right)\right)\right)$$

$$= \widetilde{O}\left(Rd_K\sqrt{T}\left(1 + \frac{1}{\lambda_A^3\lambda_\theta}\left(\sqrt{\frac{d_AT}{Nd_K}} + \sqrt{\frac{d_A}{\min(S,T)N}\sum_{t=1}^{\min(S,T)}\kappa_t^2}\right)\right)\right)$$

where the last inequality holds since $T < S$ for the first timestep $S$ satisfying $\Delta_{\text{off}}\left(\sqrt{S} + \sqrt{\frac{d_K}{S}\sum_{t=1}^{S}\kappa_t^2}\right) \geq d_A$.

Additionally, since $\Delta_{\text{off}}\left(\sqrt{T} + \sqrt{\frac{d_K}{T}\sum_{t=1}^{T}\kappa_t^2}\right) \leq d_A$, we have using equation 3 that

$$\text{Reg}_T = \widetilde{O}(d_A\sqrt{T})$$

as well. So, we have that when $\tau\Delta_{\text{off}}\sqrt{T} \leq d_A$,

$$\text{Reg}_T = \widetilde{O}\left(\min\left(d_A\sqrt{T}, Rd_K\sqrt{T}\left(1 + \frac{1}{\lambda_A^3\lambda_\theta}\left(\sqrt{\frac{d_AT}{Nd_K}} + \sqrt{\frac{d_A}{\min(S,T)N}\sum_{t=1}^{\min(S,T)}\kappa_t^2}\right)\right)\right)\right)$$

**Bounding regret when** $\tau\Delta_{\text{off}}\left(\sqrt{T} + \sqrt{\frac{d_K}{T}\sum_{t=1}^{T}\kappa_t^2}\right) \geq d_A$**:** In this regime, after the first timestep $S$ satisfying $\Delta_{\text{off}}\left(\sqrt{S} + \sqrt{\frac{d_K}{S}\sum_{t=1}^{S}\kappa_t^2}\right) \geq d_A$, we run standard LinUCB with dimension $d_A$ and incur $\widetilde{O}(d_A\sqrt{T})$ regret with

probability $1 - \delta$. Until timestep $S - 1$, we run the projected and modified version of LinUCB and incur regret bounded by

$$\widetilde{O}(d_K\sqrt{S}) + 2R\Delta_{\text{off}}S + 2R\Delta_{\text{off}}\sqrt{\frac{\sum_{t=1}^{S}\kappa_t^2}{S}}\sqrt{d_K S} = \widetilde{O}\left(d_A\sqrt{S} + d_K\sqrt{S}\right)$$
$$= \widetilde{O}(d_A\sqrt{T} + d_K\sqrt{T})$$
$$= \widetilde{O}(d_A\sqrt{T})$$

Combining these, we incur $O(d_A\sqrt{T})$ regret during the whole method.

Also, since $d_A\sqrt{T} \leq \Delta_{\text{off}}\left(\sqrt{TS} + \sqrt{\frac{d_K T}{S}\sum_{t=1}^{S}\kappa_t^2}\right)$, we get that our regret is also bounded by

$$\text{Reg}_T = \widetilde{O}\left(\Delta_{\text{off}}\left(\sqrt{TS} + \sqrt{\frac{d_K T}{S}\sum_{t=1}^{S}\kappa_t^2}\right)\right)$$

$$= \widetilde{O}(d_K\sqrt{T} + \Delta_{\text{off}}\left(\sqrt{TS} + \sqrt{\frac{d_K T}{S}\sum_{t=1}^{S}\kappa_t^2}\right))$$

$$= \widetilde{O}\left(Rd_K\sqrt{T}\left(1 + \frac{1}{\lambda_A^3\lambda_\theta}\left(\sqrt{\frac{d_A T}{N d_K}} + \sqrt{\frac{d_A}{\min(S,T)N}\sum_{t=1}^{\min(S,T)}\kappa_t^2}\right)\right)\right)$$

So, our regret satisfies

$$\text{Reg}_T = \widetilde{O}\left(\min\left(d_A\sqrt{T}, Rd_K\sqrt{T}\left(1 + \frac{1}{\lambda_A^3\lambda_\theta}\left(\sqrt{\frac{d_A T}{N d_K}} + \sqrt{\frac{d_A}{\min(S,T)N}\sum_{t=1}^{\min(S,T)}\kappa_t^2}\right)\right)\right)\right)$$

### E.2.1. UNDERSTANDING $\kappa_t$

Letting $\mathbf{X}_t = \hat{\mathbf{U}}^\top[\phi(x_1, a_1), \dots \phi(x_t, a_t)]^\top$, recall that $\kappa_t := \|\hat{\mathbf{U}}\mathbf{X}_t^\top\mathbf{X}_t\|_{(\mu\mathbf{I}+\mathbf{X}_t^\top\mathbf{X}_t)^{-1}}$. In the worst case, $\kappa_t \leq \|\hat{\mathbf{U}}\mathbf{X}_t^\top\mathbf{X}_t\|_2 = O(t)$ since actions have norm 1. In that case, $\frac{1}{S}\sum_{t=1}^{S}\kappa_t = O(S^2)$. However, if $\mathbf{X}_t$ lies in the span of $\hat{\mathbf{U}}^\top$, then $\mathbf{X}_t\hat{\mathbf{U}}^\top\hat{\mathbf{U}} = \mathbf{X}_t$. This means that for $\mathbf{Y}_t := \mathbf{X}_t\hat{\mathbf{U}}^\top$,

$$\kappa_t = \sqrt{\hat{\mathbf{U}}\mathbf{X}_t^\top\mathbf{X}_t\hat{\mathbf{U}}^\top\hat{\mathbf{U}}(\mu\mathbf{I}+\mathbf{X}_t^\top\mathbf{X}_t)^{-1}\mathbf{U}^\top\hat{\mathbf{U}}\mathbf{X}_t^\top\mathbf{X}_t\hat{\mathbf{U}}^\top}$$
$$= \sqrt{\mathbf{Y}_t^\top\mathbf{Y}_t(\mu I + \mathbf{Y}_t^\top\mathbf{Y}_t)^{-1}\mathbf{Y}_t^\top\mathbf{Y}_t}$$
$$= \sqrt{(\mathbf{I} - \mu(\mu\mathbf{I}+\mathbf{Y}_t^\top\mathbf{Y}_t))\mathbf{Y}_t^\top\mathbf{Y}_t}$$
$$= \widetilde{O}(\sqrt{t})$$

So, $\frac{1}{S}\sum_{t=1}^{S}\kappa_t = O(S) = O(T)$.

$\square$

# F. Lower Bounds

We formally state and prove the lower bound below. Much like how we generate families of reward parameters in lower bound proofs for purely online regret, we are now generating a family of tuples of latent bandits (for the offline data) and reward parameters represented in the latent bandit (for the online interaction).

**Theorem 7** (Regret Lower Bound). *Let $d_A^2 H \leq 2T$, $d_A^2 \leq N$, $d_K > 1$. Consider the action set $\mathcal{A} = \{a \mid \|a\|_2 \leq 1\}$. For each regime, either $\frac{d_K T}{(d_A - d_K)N}$ being larger than $1$, or between $1$ and $1/2$, or less than $1/2$, there exists a family of tuples $(F, \boldsymbol{\beta})$, where $F$ is a latent bandit with a rank $d_K$ latent subspace and $\boldsymbol{\beta}$ is a reward parameter in its support, satisfying the following:*

*(i) For any offline behavior policy $\pi_b$, all $F$ have uniformly bounded $\lambda_\theta$ associated to the offline data.*

*(ii) For any offline behavior policy $\pi_b$ and any learner, there is a $(F, \boldsymbol{\beta})$ such that the regret $\mathrm{Reg}(T, \boldsymbol{\beta})$ of the learner under offline data from $\pi_b$ and $F$ and online reward parameter $\boldsymbol{\beta}$ is bounded below by*

$$\mathrm{Reg}(T, \boldsymbol{\beta}) \geq \Omega\left(\min\left(d_A\sqrt{T}, d_K\sqrt{T}\left(1 + \sqrt{\frac{d_A T}{d_K N}}\right)\right)\right)$$

**Remark 1.**   • We are stating a version with no contexts and only actions here for notational simplicity, the version for contextual bandits has the same proof verbatim. One just replaces $\mathcal{A}$ with $\{\phi(x, a) \mid x \in \mathcal{X}, a \in \mathcal{A}\}$.

• Note that the theorem statement is complicated to ensure that it is essentially the strongest version of the theorem possible. Condition $(i)$ is needed to ensure that we aren't cheating by ensuring that the offline data itself obscures the correct subspace. Condition $(ii)$ is the actual regret lower bound.

*Proof.* The proof is inspired by the proof of Theorem 24.2 in Lattimore and Szepesvári (2018), giving a regret lower bound for standard stochastic linear bandits with a unit ball action set. Without loss of generality, we can assume that $d_A \geq 1.01 d_K$, otherwise both terms in the minimum have the same order and the proof is complete. An astute reader will note that the regimes are separated based on whether $d_K\sqrt{T}\left(1 + \sqrt{\frac{(d_A - d_K)T}{d_K N}}\right) \leq d_A\sqrt{T}$. We will first address the difficult regime where $\frac{1}{2} \leq \sqrt{\frac{d_K T}{(d_A - d_K)N}} \leq 1$. Until stated otherwise in this proof, we will work in this regime and assume that $\frac{1}{2} \leq \sqrt{\frac{d_K T}{(d_A - d_K)N}} \leq 1$.

## F.1. Setup

Consider $\Delta_{\mathrm{in}} = \frac{1}{5\sqrt{3}}\sqrt{d_K/T}$ and $\Delta_{\mathrm{out}} = \frac{1}{4\sqrt{3}}\sqrt{d_K/N}$. Let $\mathcal{B} := \{\pm\Delta_{\mathrm{in}}\}^{d_K - 1} \times \{0\} \times \{\pm\Delta_{\mathrm{out}}\}^{d_A - d_K}$ and let $\boldsymbol{\beta} \in \mathcal{B}$. We set the $d_K^{th}$ coordinate to $0$ for technical reasons. For any bandit instance $\boldsymbol{\beta}$, let the rewards have Gaussian noise with variance $1$. Construct a family of latent bandit-online latent state pairs as follows. Define $F_{\boldsymbol{\beta}}$ to be a latent bandit with a uniform distribution over all $2^{d_K - 1}$ reward parameters obtained by negating any of the first $d_K - 1$ coordinates of $\boldsymbol{\beta}$. Notice that this latent bandit has $2^{d_K - 1}$ latent states sharing a $d_K$-dimensional subspace. Let us construct the family of pairs $(F_{\boldsymbol{\beta}}, \boldsymbol{\beta})$, where $F_{\boldsymbol{\beta}}$ is the latent bandit used to generate offline data and $\boldsymbol{\beta}$ is the reward parameter underlying the online trajectory.

Note that condition $(i)$ is satisfied by merely computing the matrix $\mathbb{E}_{\boldsymbol{\beta}}[\boldsymbol{\beta}\boldsymbol{\beta}^\top]$ and noticing that eigenvalues are merely norms $\frac{d_K}{T\|\boldsymbol{\beta}\|_2}$ or $\frac{d_K}{N\|\boldsymbol{\beta}\|_2}$, up to a constant. Now note that $\|\boldsymbol{\beta}\|_2 = \sqrt{\frac{d_K^2}{T} + \frac{d_K(d_A - d_K)}{N}}$ up to a constant, so $\lambda_\theta$ is bounded since $\frac{1}{2} \leq \frac{d_K T}{(d_A - d_K)N} \leq 1$

Notice that if $\boldsymbol{\beta}'$ is $\boldsymbol{\beta}$ with any of the first $d_K - 1$ coordinates negated, then $F_{\boldsymbol{\beta}'}$ is the same latent bandit as $F_{\boldsymbol{\beta}}$. Assume that a fixed behavior policy $\pi_b$ is used to produce offline data, producing a known dataset of contexts and actions shared across all latent bandit instances. We will repeatedly use the fact that $d_K - 1 \geq d_K/2$ in this proof.

## F.2. Proof Sketch and Intuition

Notice that we are working in the regime where $N$ is significantly larger than $T$. That means that we are treating the first $d_K$ coordinates as the main subspace and the rest of the coordinates as perturbations out of the subspace. This is represented in the notation $\Delta_{\mathrm{in}}$ and $\Delta_{\mathrm{out}}$. In our regime, where $\sqrt{\frac{d_K T}{(d_A - d_K)N}} \leq 1$, $\Delta_{\mathrm{out}}$ should be thought of as much smaller than $\Delta_{\mathrm{in}}$.

Eventually, we intend to lower bound the average regret over all pairs $(F_\beta, \beta)$ ranging over the vertices of a hypercuboid $\mathcal{B}$ of reward parameters. This will allow us to claim that there exists one parameter $\beta \in \mathcal{B}$ for which the regret is larger than this average. To lower bound this average, we first use change of measure inequalities, careful computation and clever design of $\mathcal{B}$ to get an intermediate lower bound, bounding the average regret over any pair of "adjacent" tuples $(F', \beta')$ and $(F, \beta)$. These are pairs where the sign of only one coordinate is flipped from $\beta$ to $\beta'$ and $F$ and $F'$ at most differ in their "out of subspace" perturbation. We can then average over all such pairs to lower bound the regret averaged over all $\beta \in \mathcal{B}$, as desired.

We will need two separate computations for the intermediate lower bound – one for when the pair of adjacent reward parameters corresponds to a coordinate $i < d_K$, and another for when it corresponds to a coordinate $i > d_K$. The proof ends with the averaging trick mentioned in the previous paragraph.

### F.3. Regret Lower Bound Decomposition
For $i < d_K$, define $\tau_i = T \wedge \min\{t : \sum_{s=1}^t A_{si}^2 \geq T/d_K\}$. For $i \geq d_K$, define $\tau_i = T \wedge \min\{t : \sum_{s=1}^t A_{si}^2 \geq T/(d_A - d_K)\}$. For now, let us denote the regret under parameter $\beta$ to be $\mathrm{Reg}(T, \beta)$. Now note the following decomposition of the lower bound.

$$
\begin{aligned}
\mathrm{Reg}(T, \beta) &= \mathbb{E}_\beta \Big[ \Delta_\mathrm{in} \sum_{t=1}^T \sum_{i=1}^{d_K-1} \Big( \frac{1}{\sqrt{d_K-1}} - A_{ti}\,\mathrm{sign}(\beta_i) \Big) \\
&\quad + \Delta_\mathrm{out} \sum_{t=1}^T \sum_{i=d_K+1}^{d_A} \Big( \frac{1}{\sqrt{d_A-d_K}} - A_{ti}\,\mathrm{sign}(\beta_i) \Big) \Big] \\
&\geq \mathbb{E}_\beta \Big[ \frac{\Delta_\mathrm{in} \sqrt{d_K-1}}{2} \sum_{t=1}^T \sum_{i=1}^{d_K-1} \Big( \frac{1}{\sqrt{d_K-1}} - A_{ti}\,\mathrm{sign}(\beta_i) \Big)^2 \\
&\quad + \frac{\Delta_\mathrm{out} \sqrt{d_A-d_K}}{2} \sum_{t=1}^T \sum_{i=d_K+1}^{d_A} \Big( \frac{1}{\sqrt{d_A-d_K}} - A_{ti}\,\mathrm{sign}(\beta_i) \Big)^2 \Big] \\
&\geq \frac{\Delta_\mathrm{in} \sqrt{d_K-1}}{2} \sum_{i=1}^{d_K-1} \mathbb{E}_\beta \Big[ \sum_{t=1}^{\tau_i} \Big( \frac{1}{\sqrt{d_K-1}} - A_{ti}\,\mathrm{sign}(\beta_i) \Big)^2 \Big] \\
&\quad + \frac{\Delta_\mathrm{out} \sqrt{d_A-d_K}}{2} \sum_{i=d_K+1}^{d_A} \mathbb{E}_\beta \Big[ \sum_{t=1}^{\tau_i} \Big( \frac{1}{\sqrt{d_A-d_K}} - A_{ti}\,\mathrm{sign}(\beta_i) \Big)^2 \Big]
\end{aligned}
$$

The first inequality holds by merely evaluating the square, simplifying and noting that $\|A_t\|_2^2 \leq 1$. The second inequality For $i < d_K$ and $x \in \{\pm 1\}$, define $U_i(x) := \sum_{t=1}^{\tau_i} \Big( \frac{1}{\sqrt{d_K-1}} - A_{ti}\,\mathrm{sign}(\beta_i) \Big)^2$. For $i > d_K$ and $x \in \{\pm 1\}$, define $U_i(x) := \sum_{t=1}^{\tau_i} \Big( \frac{1}{\sqrt{d_A-d_K}} - A_{ti}\,\mathrm{sign}(\beta_i) \Big)^2$.

Fix $i$ and let $\beta'$ be such that $\beta'_j = \beta_j$ for $j \neq i$ and $\beta'_i = -\beta_i$. Let $\mathbb{P}$ and $\mathbb{P}'$ be the joint laws of the offline data and the bandit/learner interaction measure for $\beta$ and $\beta'$ respectively. We will bound $\mathbb{E}_\beta[U_i(1)] + \mathbb{E}_{\beta'}[U_i(-1)]$ in the following subsections, treating $i < d_K$ and $i > d_K$ separately. This will allow us to bound $\sum_{\beta \in \mathcal{B}} \mathbb{E}_\beta[U_i(\mathrm{sign}(\beta_i))]$ later and apply an averaging trick.

### F.4. Bounding $\mathbb{E}_\beta[U_i(1)] + \mathbb{E}_{\beta'}[U_i(-1)]$ when $i < d_K$
Note that

$$
\begin{aligned}
\mathbb{E}_\beta[U_i(1)] &\overset{(i)}{\geq} \mathbb{E}_{\beta'}[U_i(-1)] - \Big( \frac{6T}{d_K} + 2 \Big) \sqrt{\frac{1}{2} D(\mathbb{P}, \mathbb{P}')} \\
&\overset{(ii)}{\geq} \mathbb{E}_{\beta'}[U_i(-1)] - \Delta_\mathrm{in} \Big( \frac{3T}{d_K} + 1 \Big) \sqrt{\sum_{t=1}^{\tau_i} A_{ti}^2}
\end{aligned}
$$

$$\overset{(iii)}{\geq} \mathbb{E}_{\boldsymbol{\beta}'}[U_i(-1)] - \Delta_{\text{in}} \left( \frac{3T}{d_K} + 1 \right) \sqrt{\frac{T}{d_K} + 1}$$

$$\overset{(iv)}{\geq} \mathbb{E}_{\boldsymbol{\beta}'}[U_i(-1)] - \frac{5\sqrt{3}\Delta_{\text{in}}T}{d_K} \sqrt{\frac{T}{d_K}} \tag{4}$$

where in $(i)$, we rely on the bound below and then use the TV distance change of measure inequality, followed by Pinsker's inequality. The bound below relies on the fact that $d_K - 1 \geq d_K/2$.

$$U_i(1) = \sum_{t=1}^{\tau_i} \left( \frac{1}{\sqrt{d_K - 1}} - A_{ti}\,\text{sign}(\beta_i) \right)^2 \leq 2\sum_{t=1}^{\tau_i} \frac{1}{d_K - 1} + 2\sum_{t=1}^{\tau_i} A_{ti}^2 \leq \frac{4T}{d_K} + \frac{2T}{d_K} + 2 = \frac{6T}{d_K} + 2$$

For the bound above, we use the definition of $\tau_i$. In $(ii)$, we use the chain rule for KL divergence under a stopping time, and crucially note that the offline data distributions is identical in this case since $F_{\boldsymbol{\beta}} = F_{\boldsymbol{\beta}'}$. Inequality $(iii)$ holds by the definition of $\tau_i$, and inequality $(iv)$ holds since $d_K \leq d_A \leq 2T$ since $d_A^2 H \leq 2T$.

So, we can conclude that

$$\mathbb{E}_{\boldsymbol{\beta}}[U_i(1)] + \mathbb{E}_{\boldsymbol{\beta}'}[U_i(-1)] \geq \mathbb{E}_{\boldsymbol{\beta}'}[U_i(1) + U_i(-1)] - \frac{5\sqrt{3}\Delta_{\text{in}}T}{d_K}\sqrt{\frac{T}{d_K}}$$

$$= 2\mathbb{E}_{\boldsymbol{\beta}'}\left[ \frac{\tau_i}{d_K - 1} + \sum_{t=1}^{\tau_i} A_{ti^2} \right] - \frac{5\sqrt{3}\Delta_{\text{in}}T}{d_K}\sqrt{\frac{T}{d_K}}$$

$$\geq \frac{2T}{d_K} - \frac{5\sqrt{3}\Delta_{\text{in}}T}{d_K}\sqrt{\frac{T}{d_K}}$$

$$= \frac{T}{d_K} \tag{5}$$

**F.5. Bounding $\mathbb{E}_{\boldsymbol{\beta}}[U_i(1)] + \mathbb{E}_{\boldsymbol{\beta}'}[U_i(-1)]$ when $i > d_K$**

Note the following computation, where we let $A_{r,h}$ be the action chosen at step $h$ of offline trajectories $d$, where $h = 1 \to H$ and $r = 1 \to N$.

$$\mathbb{E}_{\boldsymbol{\beta}}[U_i(1)] \overset{(i)}{\geq} \mathbb{E}_{\boldsymbol{\beta}'}[U_i(-1)] - \left( \frac{4T}{d_A - d_K} + 2 \right) \sqrt{\frac{1}{2}D(\mathbb{P}, \mathbb{P}')}$$

$$\overset{(ii)}{\geq} \mathbb{E}_{\boldsymbol{\beta}'}[U_i(-1)] - \Delta_{\text{out}} \left( \frac{2T}{d_A - d_K} + 1 \right) \sqrt{\sum_{t=1}^{\tau_i} A_{ti}^2 + \frac{d_K}{48N}\mathbb{E}_{\pi_b}[\sum_{r=1}^{N}\sum_{h=1}^{H} A_{r,h,i}^2]}$$

$$\overset{(iii)}{\geq} \mathbb{E}_{\boldsymbol{\beta}'}[U_i(-1)] - \Delta_{\text{out}} \left( \frac{2T}{d_A - d_K} + 1 \right) \sqrt{\frac{T}{d_A - d_K} + \frac{d_K H}{48}}$$

$$\overset{(iv)}{\geq} \mathbb{E}_{\boldsymbol{\beta}'}[U_i(-1)] - \frac{5\sqrt{3}\Delta_{\text{out}}T}{d_A - d_K}\sqrt{\frac{T}{d_A - d_K}}$$

where again in $(i)$, we rely on the bound below and use the TV distance change of measure inequality, followed by Pinsker's inequality.

$$U_i(1) = \sum_{t=1}^{\tau_i} \left( \frac{1}{\sqrt{d_A - d_K}} - A_{ti}\,\text{sign}(\beta_i) \right)^2 \leq 2\sum_{t=1}^{\tau_i} \frac{1}{d_A - d_K} + 2\sum_{t=1}^{\tau_i} A_{ti}^2 \leq \frac{4T}{d_A - d_K} + 2$$

For the bound above, we use the definition of $\tau_i$. In $(ii)$, we use the chain rule for KL divergence under a stopping time and include the non-zero KL divergence coming from the offline term this time, which appears as the second term

in the square root. Inequality $(iii)$ holds by the definition of $\tau_i$ and the fact that $A_{ri}^2 \leq 1$. Inequality $(iv)$ holds since $d_k(d_A - d_K)H \leq d_A^2 H \leq 2T$.

So, we can conclude that

$$
\begin{aligned}
\mathbb{E}_{\boldsymbol{\beta}}[U_i(1)] + \mathbb{E}_{\boldsymbol{\beta}'}[U_i(-1)] &\geq \mathbb{E}_{\boldsymbol{\beta}'}[U_i(1) + U_i(-1)] - \frac{4\sqrt{3}\Delta_{\text{out}}T}{d_A - d_K}\sqrt{\frac{T}{d_A - d_K}} \\
&= \mathbb{E}_{\boldsymbol{\beta}'}\left[\frac{\tau_i}{d_A - d_K} + \sum_{t=1}^{\tau_i} A_{ti^2}\right] - \frac{4\sqrt{3}\Delta_{\text{out}}T}{d_A - d_K}\sqrt{\frac{T}{d_A - d_K}} \\
&\geq \frac{2T}{d_A - d_K} - \frac{4\sqrt{3}\Delta_{\text{out}}T}{d_A - d_K}\sqrt{\frac{T}{d_A - d_K}} \\
&= \frac{2T}{d_A - d_K} - \frac{T}{d_A - d_K}\sqrt{\frac{d_K T}{(d_A - d_K)N}} \\
&\geq \frac{T}{d_A - d_K}
\end{aligned}
\tag{6}
$$

where crucially, the last inequality holds since $\sqrt{\frac{d_K T}{(d_A - d_K)N}} \leq 1$.

### F.6. Lower bounding regret using an averaging trick

For $i \leq d_K$, define by $\mathcal{B}_{-i} := \{\pm\Delta_{\text{in}}\}^{d_K - 2} \times \{0\} \times \{\pm\Delta_{\text{out}}\}^{d_A - d_K}$, which is the slice of $\mathcal{B}$ where all coordinates but $\beta_i$ vary. Similarly, for $i > d_K$, define the slice $\mathcal{B}_{-i} := \{\pm\Delta_{\text{in}}\}^{d_K - 1} \times \{0\} \times \{\pm\Delta_{\text{out}}\}^{d_A - d_K - 1}$. We will denote the tuple of coordinates of $\boldsymbol{\beta}$ other than $i$ by $\beta_{-i}$. We thus get the following lower bound on regret, using inequalities 4, 5 and 6.

$$
\begin{aligned}
\sum_{\boldsymbol{\beta}\in\mathcal{B}} \text{Reg}(T, \boldsymbol{\beta}) &\geq \frac{\Delta_{\text{in}}\sqrt{d_K - 1}}{2} \sum_{i=1}^{d_K - 1} \sum_{\boldsymbol{\beta}\in\mathcal{B}} \mathbb{E}_{\boldsymbol{\beta}}[U_i(\text{sign}(\beta_i))] \\
&\quad + \frac{\Delta_{\text{out}}\sqrt{d_A - d_K}}{2} \sum_{i=d_K + 1}^{d_A} \sum_{\boldsymbol{\beta}\in\mathcal{B}} \mathbb{E}_{\boldsymbol{\beta}}[U_i(\text{sign}(\beta_i))] \\
&= \frac{\Delta_{\text{in}}\sqrt{d_K - 1}}{2} \sum_{i=1}^{d_K - 1} \sum_{\beta_{-i}\in\mathcal{B}_{-i}} \sum_{\beta_i\in\{\pm\Delta_{\text{in}}\}} \mathbb{E}_{\boldsymbol{\beta}}[U_i(\text{sign}(\beta_i))] \\
&\quad + \frac{\Delta_{\text{out}}\sqrt{d_A - d_K}}{2} \sum_{i=d_K + 1}^{d_A} \sum_{\beta_{-i}\in\mathcal{B}_{-i}} \sum_{\beta_i\in\{\pm\Delta_{\text{out}}\}} \mathbb{E}_{\boldsymbol{\beta}}[U_i(\text{sign}(\beta_i))] \\
&\geq \frac{\Delta_{\text{in}}\sqrt{d_K - 1}}{2} \sum_{i=1}^{d_K - 1} \sum_{\beta_{-i}\in\mathcal{B}_{-i}} \frac{T}{d_K} + \frac{\Delta_{\text{out}}\sqrt{d_A - d_K}}{2} \sum_{i=d_K + 1}^{d_A} \sum_{\beta_{-i}\in\mathcal{B}_{-i}} \frac{T}{d_A - d_K} \\
&\geq \frac{\Delta_{\text{in}}\sqrt{d_K}}{2\sqrt{2}} \sum_{\beta_{-i}\in\mathcal{B}_{-i}} \frac{T}{2} + \frac{\Delta_{\text{out}}\sqrt{d_A - d_K}}{2} \sum_{\beta_{-i}\in\mathcal{B}_{-i}} T \\
&= \frac{2^{d_A}}{80\sqrt{6}} d_K \sqrt{T}\left(1 + 5\sqrt{2\frac{(d_A - d_K)T}{d_K N}}\right)
\end{aligned}
$$

That means that there exists $\boldsymbol{\beta} \in \mathcal{B}$ so that

$$
\text{Reg}(T, \boldsymbol{\beta}) \geq \frac{1}{80\sqrt{6}} d_K \sqrt{T}\left(1 + \sqrt{\frac{(d_A - d_K)T}{d_K N}}\right)
$$

As desired.

### F.7. The Other Two Regimes

In the regime $d_K T \geq (d_A - d_K)N$, one can simply use the $2^{d_A}$ bandit instances in the standard unit ball regret lower bound from Theorem 24.2 in Lattimore and Szepesvári (2018) with dimension $d_A$, and follow the proof essentially verbatim. The only difference is that we will be choosing pairs of tuples $(F', \beta')$ and $(F, \beta)$ instead of just pairs of reward parameters $\beta'$ and $\beta$. One can choose any latent bandit with $d_K$ reward parameters in its support, two of which are $\beta$ and $\beta'$, and set both $F$ and $F'$ to this. For this, it is convenient to choose the latent bandit to have a uniform distribution over $2^d_K$ reward parameters obtained by flipping signs of $d_K$ chosen coordinates, since then one can easily compute that $\lambda_\theta = 1$. This will ensure that offline data distributions are identical and the KL divergence contribution from the offline data distribution is 0, allowing us to follow the proof of Theorem 24.2 in Lattimore and Szepesvári (2018) essentially verbatim. This establishes condition $(ii)$, and we have also establis

Similarly, when $d_K T \ll (d_A - d_K)N$, we can use the standard lower bound from Theorem 24.2 in Lattimore and Szepesvári (2018) again, this time with dimension $d_K$. Fix $F$ to be the latent bandit with a uniform distribution over all $2^{d_K}$ reward parameters $\mathcal{B} = \{\pm \Delta_{\text{in}}\}^{d_K} \times \{0\}^{d_A - d_K}$, and consider the family $(F, \beta)$ of tuples with fixed $F$ and $\beta$ varying through $\mathcal{B}$. We can now follow the proof of Theorem 24.2 in Lattimore and Szepesvári (2018) verbatim. Again, the only difference is that we will be choosing pairs of tuples $(F, \beta')$ and $(F, \beta)$ instead of just pairs of reward parameters $\beta'$ and $\beta$. And yet again, we can check that $\lambda_\theta = 1$ and condition $(i)$ is thus satisfied. $\square$

## G. Additional Algorithms

We provide a version of SOLD that utilizes pseudoinverses. We use this within our experiments to avoid having to search for regularization parameters, and recommend that the user use this instead of Algorithm 1 when finding a suitable regularization parameter is a concern.

---

**Algorithm 4** Subspace estimation from Offline Latent bandit Data (SOLD) – Pseudoinverse Version

---

1: **Input:** Dataset $\mathcal{D}_{\text{off}}$ of collected trajectories $\tau_n = ((x_{n,1}, a_{n,1}, r_{n,1}), ..., (x_{n,H}, a_{n,H}, r_{n,1}))$ under a behavior policy $\pi_b$, dimension of latent subspace $d_K$.
2: **Divide** each $\tau_n$ into odd and even steps, giving trajectory halves $\tau_{n,1}$ and $\tau_{n,2}$.
3: **Estimate** reward parameters $\hat{\beta}_{n,i} \leftarrow \mathbf{V}_{n,i}^\dagger \mathbf{b}_{n,i}$, where $\mathbf{V}_{n,i} \leftarrow \sum_{(x,a,r) \in \tau_{n,i}} \phi(x,a)\phi(x,a)^\top$ and $\mathbf{b}_{n,i} \leftarrow \sum_{(x,a,r) \in \tau_{n,i}} \phi(x,a)r$ for $i = 1, 2$.
4: **Compute** $\mathbf{M}_n \leftarrow \frac{1}{2}(\hat{\beta}_{n,1}\hat{\beta}_{n,2}^\top + \hat{\beta}_{n,2}\hat{\beta}_{n,1}^\top)$ and compute $\overline{\mathbf{M}}_N \leftarrow \frac{1}{N}\sum_{n=1}^N \mathbf{M}_n$.
5: **Compute** $\mathbf{W}_{n,i}$, the eigenvectors of $\mathbf{V}_{n,i}$ corresponding to nonzero eigenvalues.
6: **Compute** $\overline{\mathbf{D}}_{N,i} \leftarrow \frac{1}{N}\sum_{n=1}^N (\mathbf{W}_{n,i}\mathbf{W}_{n,i}^\top)^\dagger$, $i = 1, 2$.
7: **Obtain** $\hat{\mathbf{U}}$, the top $d_K$ eigenvectors of $\overline{\mathbf{D}}_{N,1}^{-1}\overline{\mathbf{M}}_N\overline{\mathbf{D}}_{N,2}^{-1}$.
8: **return** Projection matrix $\hat{\mathbf{U}}\hat{\mathbf{U}}^\top$, $\Delta_{\text{off}}$ as in Theorem 1

---

We also provide a method of instantiating the ProBALL framework with linear Thompson sampling. Like ProBALL-UCB, ProBALL-TS operates within the estimated subspace until the online uncertainty is low enough. We therefore maintain two normal posterior distributions, one over the latent state parameter in the estimated subspace, and one over the high-dimensional reward parameter, and sample from them as such.

---

**Algorithm 5** Projection and Bonuses for Accelerating Latent bandit Thompson Sampling (ProBALL-TS)

---

1: **Input:** Projection matrix $\hat{\mathbf{U}}\hat{\mathbf{U}}^\top$, confidence bound $\Delta_{\text{off}}$. Hyperparameters $\alpha_{1,t}, \alpha_{2,t}, \tau, \tau'$.
2: **Initialize** $\mathbf{V}_1 \leftarrow I$, $\mathbf{b}_1 \leftarrow 0$, $\mathbf{C}_t \leftarrow 0$
3: **for** $t = 1, \ldots T$ **do**
4:     **if** $\Delta_{\text{off}}\tau\sqrt{t} + \Delta_{\text{off}}\tau'\sqrt{d_K \sum_{s=1}^t \kappa_s^2/t} \le d_A$ **then**
5:         **Compute** $\bar{\theta}_{1,t} \leftarrow (\hat{\mathbf{U}}^\top \mathbf{V}_t \hat{\mathbf{U}})^{-1}\hat{\mathbf{U}}^\top \mathbf{b}_t$
6:         **Sample** $\hat{\theta}_{1,t} \sim \mathcal{N}\left(\bar{\beta}_{1,t}, \alpha_{1,t}^2(\hat{\mathbf{U}}^\top \mathbf{V}_t \hat{\mathbf{U}})^{-1}\right)$
7:         **Play** $a_t \leftarrow \arg\max_a \phi(x_t, a)^\top \hat{\mathbf{U}}\hat{\theta}_{1,t}$
8:     **else**
9:         **Compute** $\bar{\beta}_{2,t} \leftarrow \mathbf{V}_t^{-1}\mathbf{b}_t$
10:       **Sample** $\hat{\beta}_{2,t} \sim \mathcal{N}\left(\bar{\beta}_{2,t}, \alpha_{2,t}^2\mathbf{V}_t^{-1}\right)$
11:       **Play** $a_t \leftarrow \arg\max_a \phi(x_t, a)^\top \hat{\beta}_{2,t}$
12:     **end if**
13:     **Observe** reward $r_t$ and update $\mathbf{b}_{t+1} \leftarrow \mathbf{b}_t + \phi(x_t, a)r_t$, $\mathbf{V}_{t+1} \leftarrow \mathbf{V}_t + \phi(x_t, a)\phi(x_t, a)^\top$
14:     **Update** $\mathbf{C}_{t+1} \leftarrow \mathbf{C}_t + \hat{\mathbf{U}}^\top\phi(x_t, a_t)\phi(x_t, a_t)^\top$, $\kappa_{t+1} \leftarrow \|\mathbf{C}_{t+1}\|_{(\hat{\mathbf{U}}^\top \mathbf{V}_{t+1}\hat{\mathbf{U}})^{-1}}$
15: **end for**

---

# H. Experimental Details and Additional Experiments

## H.1. Determining the Latent Rank from Offline Data

We note that as discussed in 3, we can use the eigenvalues of $\overline{\mathbf{D}}_{N,1}^{-1}\overline{\mathbf{M}}_N\overline{\mathbf{D}}_{N,2}^{-1}$ to determine the rank of our subspace. We use the version of this arising from pseudo-inverses instead of regularization, just like in the MovieLens experiments. We demonstrate that we can indeed determine that the $d_K = 18$ by finding the significant eigenvalues of the pseudo-inverse version of $\overline{\mathbf{D}}_{N,1}^{-1}\overline{\mathbf{M}}_N\overline{\mathbf{D}}_{N,2}^{-1}$ estimated from the offline dataset of 5000 samples. We show the plots and log plots of these eigenvalues. We also plot the eigenvalues of the completed ratings matrix for comparison. Notice that they match and both fall after 18 eigenvalues.

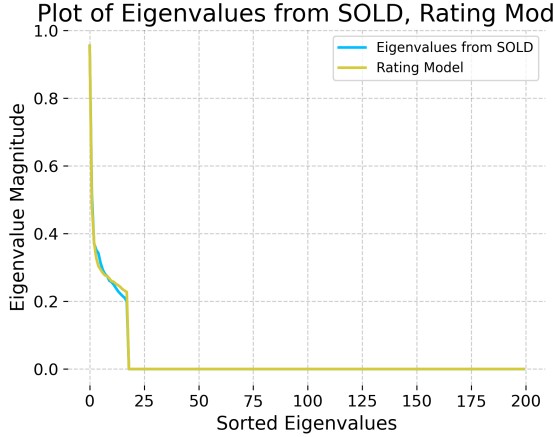

Figure 3: Plot of eigenvalues of aforementioned matrix. Notice the drop after 18 eigenvalues.

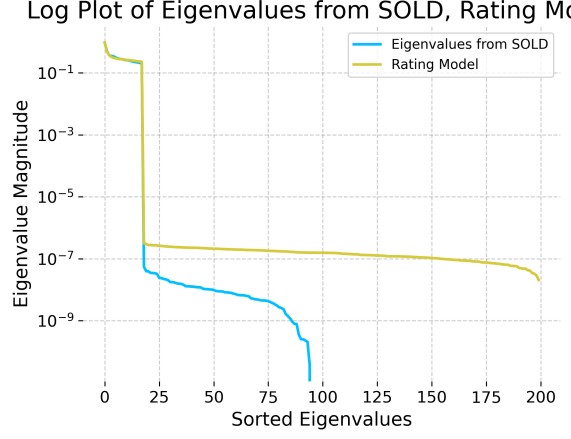

Figure 4: Log-plot of eigenvalues of aforementioned matrix. Notice the drop after 18 eigenvalues.

## H.2. Simulation Study

We generate $\mathbf{U}_*$ with $\mathbf{U}_{ij}$ i.i.d. $\mathrm{Unif}(0, \frac{2.5}{d_K d_A})$. We simulate the hidden labels $\boldsymbol{\theta}_n \sim \mathcal{N}(0, d_K^{-1}\mathbf{I}_{d_K})$, generate feature vectors $\phi(x_{n,h}, a_{n,h}) \sim \mathcal{N}(0, \mathbf{I}_{d_A})$ normalized to unit norm, and sample noise $\epsilon_{n,h}$ i.i.d. $\mathcal{N}(0, 0.5^2)$. We use SOLD to estimate $\hat{\mathbf{U}}$ from the offline dataset $\mathcal{D}_{\mathrm{off}}$, which consists of 5000 trajectories of length 20 each. In accordance with the confidence set determined by (Li et al., 2010), we choose $\alpha_{1,t} = 0.33\sqrt{d_K \log(1 + 10T/d_K)}$ and $\alpha_{2,t} = 0.33\sqrt{d_A \log(1 + 10T/d_A)}$, and share the LinUCB and ProBALL-UCB hyperparameters by assigning $\alpha_t = \alpha_{2,t}$. [11]

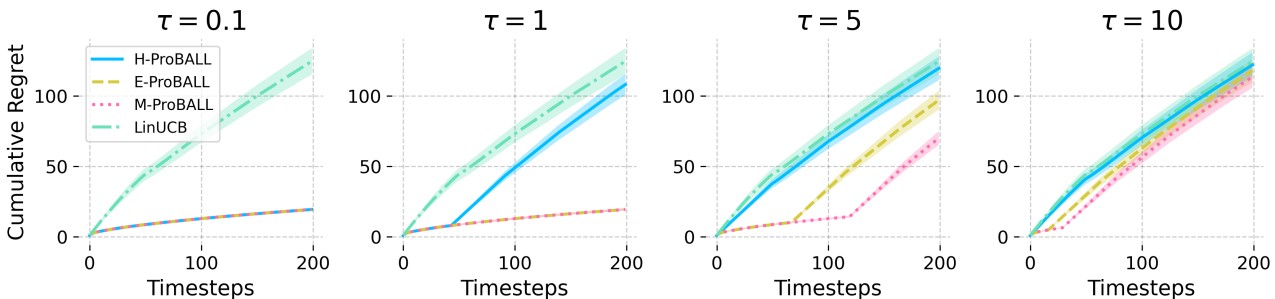

Figure 5: Comparison of ProBALL-UCB with LinUCB, for different choices of $\tau$ and confidence bound constructions. All variants perform no worse than LinUCB, with martingale Bernstein performing the best. The shaded area depicts 1-standard error confidence intervals over 30 trials.

---

[11]All experiments were run on a single computer with an Intel i9-13900k CPU, 128GB of RAM, and a NVIDIA RTX 3090 GPU, in no more than an hour in total.

## H.3. MovieLens

MovieLens (Harper and Konstan, 2015) is a large-scale movie recommendation dataset comprising 6040 users and 3883 movies, where each user may rate one or more movies. Like (Hong et al., 2020), we filter the dataset to include only movies rated by at least 200 users and vice-versa. We factor the sparse rating matrix into user parameters $\beta$ and movie features $\Phi$ using the probabilistic matrix factorization algorithm of (Mnih and Salakhutdinov, 2007b), using nuclear norm regularization so that the rank of $\beta$ is $d_K = 18$. However, we consider a much higher dimensional problem than (Hong et al., 2020) do – we let $d_A = 200$ so $\beta \in \mathbb{R}^{1589 \times 200}, \Phi \in \mathbb{R}^{200 \times 1426}$. At each round for user $i$, the agent chooses between 20 movies of different genres with features $\Phi_{a_1}, ..., \Phi_{a_{20}}$, and has to recommend the best movie presented to it to maximize the user's rating of the movie. We generate rewards for recommending movie $j$ to user $i$ by $\beta_i^T \Phi_j + \epsilon_{ij}, \epsilon_{ij}$ i.i.d. $\mathcal{N}(0, 0.5)$.

Our hyperparameters are chosen and varied just as in the simulation study. To reproduce the methods of (Hong et al., 2020), we cluster the user features into $d_K$ clusters using k-means, and provide mUCB and mmUCB with the mean vectors of each cluster as latent models. We initialize ProBALL-UCB with a subspace estimated with an unregularized variant of SOLD, that uses pseudo-inverses instead of inverses, because of difficulties in finding an appropriate regularization parameter for this large, noisy, and high-dimensional dataset. The subspace was estimated from 5000 trajectories of length 50 simulated from the reward model and the uniform behavior policy. Note that we assign $\Delta_{\text{off}}$ for $\epsilon$ in mmUCB, as this is their tolerance parameter for model misspecification.

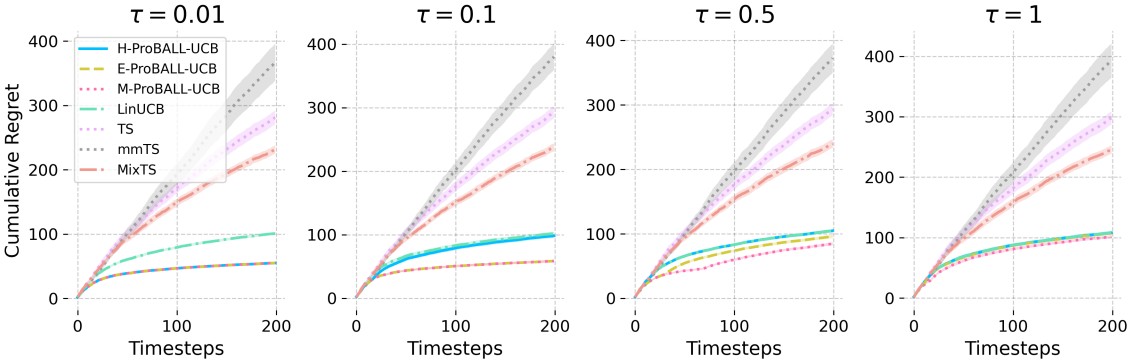

Figure 6: Comparison of ProBALL-UCB with LinUCB and TS algorithms, for different choices of $\tau$ and confidence bound constructions. All variants perform no worse than LinUCB and outperform the TS algorithms, with martingale Bernstein performing the best. The shaded area depicts 1-standard error confidence intervals over 30 trials.

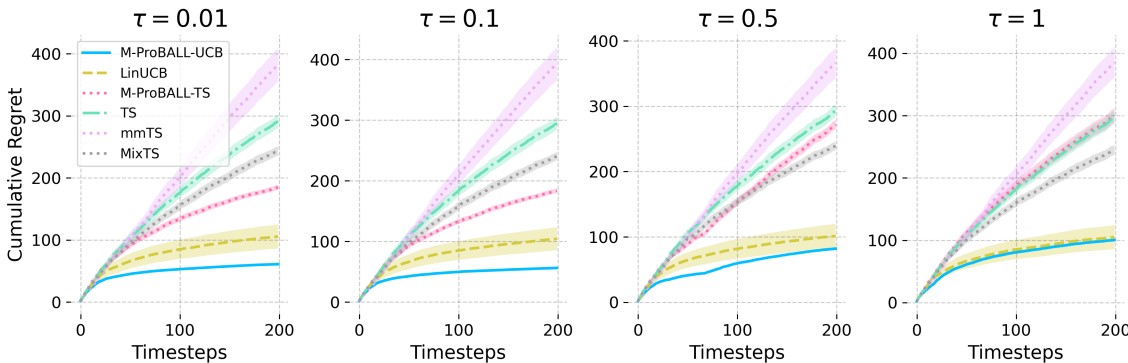

Figure 7: Comparison of ProBALL-UCB and ProBALL-TS initialized with SOLD against LinUCB, TS, MixTS, and mmTS, for different choices of $\tau$ and confidence bound constructions. ProBALL-UCB outperforms LinUCB, and ProBALL-TS outperforms MixTS and mmTS. Shaded area depicts 1-standard error confidence intervals over 30 trials with fresh $\theta$. The confidence intervals on regret thus account for the variation in frequentist regret for changing $\theta$.

### H.3.1. UCB ALGORITHMS

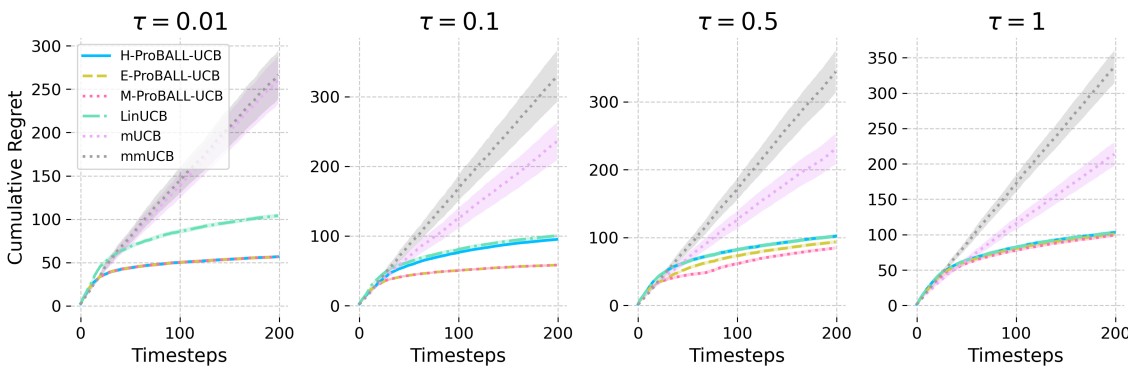

Figure 8: Comparison of ProBALL-UCB initialized with SOLD against LinUCB, mUCB, and mmUCB, for different choices of $\tau$ and confidence bound constructions, in terms of regret. All variants of ProBALL-UCB perform no worse than LinUCB, and outperform mUCB and mmUCB. Shaded area depicts 1-standard error confidence intervals over 30 trials with fresh $\theta$. The confidence intervals on regret thus account for the variation in frequentist regret for changing $\theta$.

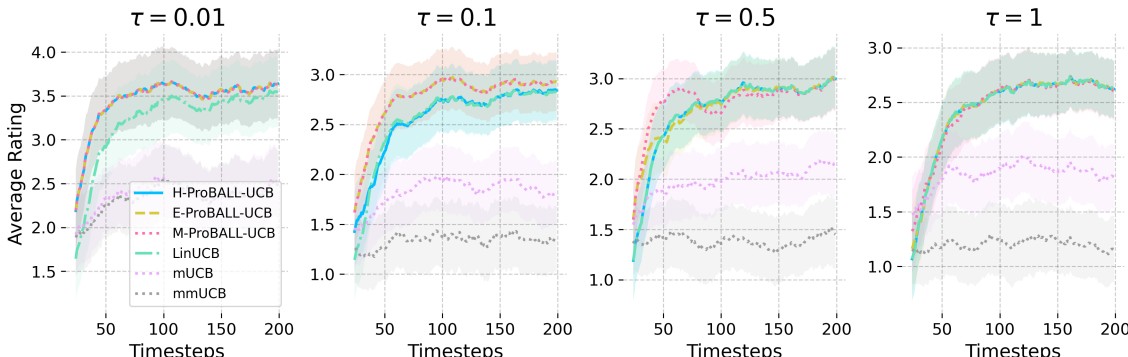

Figure 9: Comparison of ProBALL-UCB initialized with SOLD against LinUCB, mUCB, and mmUCB, for different choices of $\tau$ and confidence bound constructions, in terms of rolling average rating over 25 timesteps. ProBALL-UCB performs no worse than LinUCB, and outperforms mUCB and mmUCB.

## H.3.2. TS ALGORITHMS

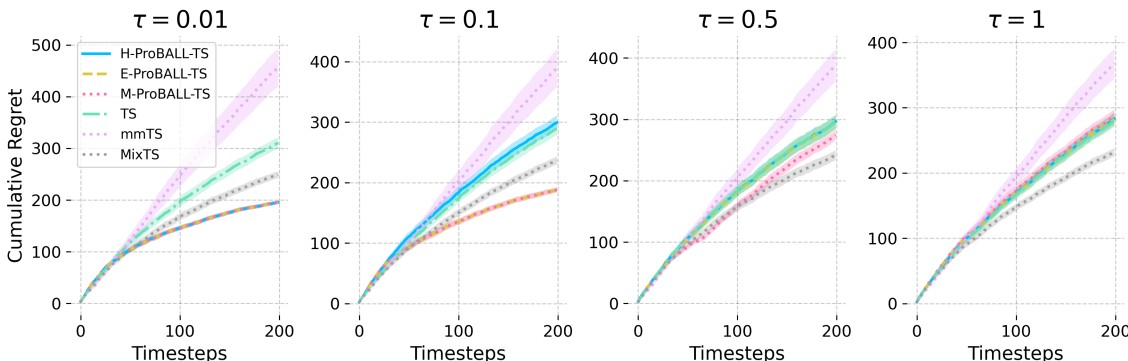

Figure 10: Comparison of ProBALL-TS initialized with SOLD against TS, mmTS, and MixTS, for different choices of $\tau$ and confidence bound constructions, in terms of regret. All variants of ProBALL-TS outperform TS, mmTS, and MixTS.

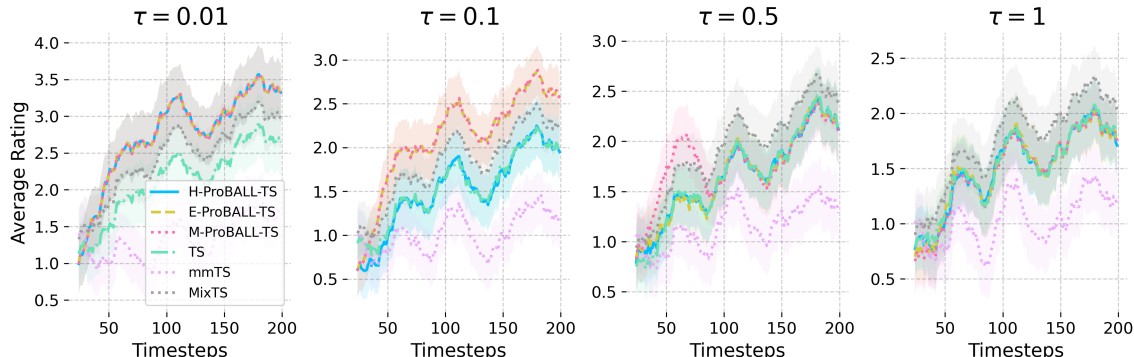

Figure 11: Comparison of ProBALL-TS initialized with SOLD against TS, mmTS, and MixTS, for different choices of $\tau$ and confidence bound constructions, in terms of rolling average rating over $25$ timesteps. All variants of ProBALL-TS outperform TS, mmTS, and MixTS. Shaded area depicts 1-standard error confidence intervals over 30 trials with fresh $\theta$. The confidence intervals on regret thus account for the variation in frequentist regret for changing $\theta$.

### H.3.3. COMPARISON AGAINST LOW-DIMENSIONAL GROUND TRUTH SUBSPACES

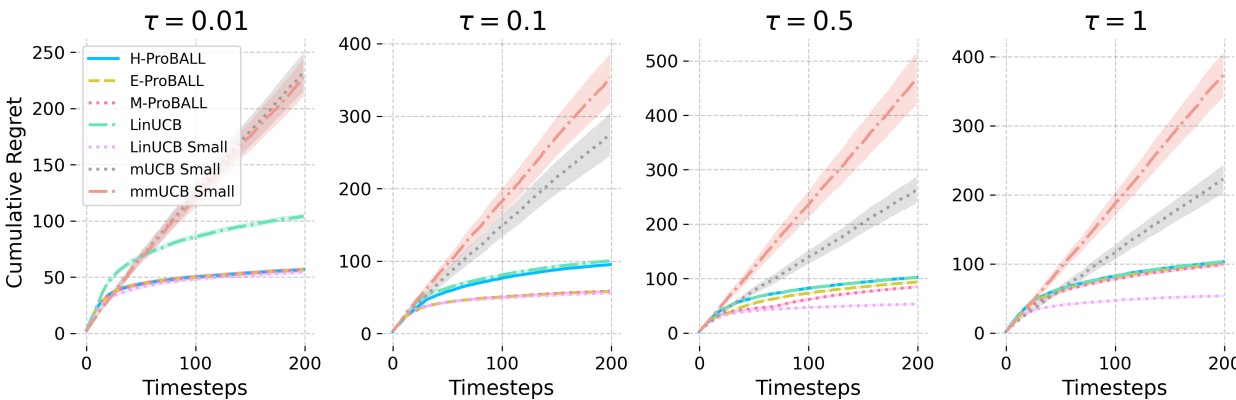

Figure 12: Comparison of ProBALL-UCB initialized with SOLD against LinUCB, mUCB, and mmUCB on low-dimensional ground-truth features, for different choices of $\tau$ and confidence bound constructions. When $\tau$ is small enough, all variants of ProBALL-UCB perform no worse than low-dimensional LinUCB, and outperform mUCB and mmUCB, on ground truth features. This showcases the efficacy of SOLD, and demonstrates that we recover subspaces that are just as good as ground-truth. Shaded area depicts 1-standard error confidence intervals over 30 trials with fresh $\theta$. The confidence intervals on regret thus account for the variation in frequentist regret for changing $\theta$.

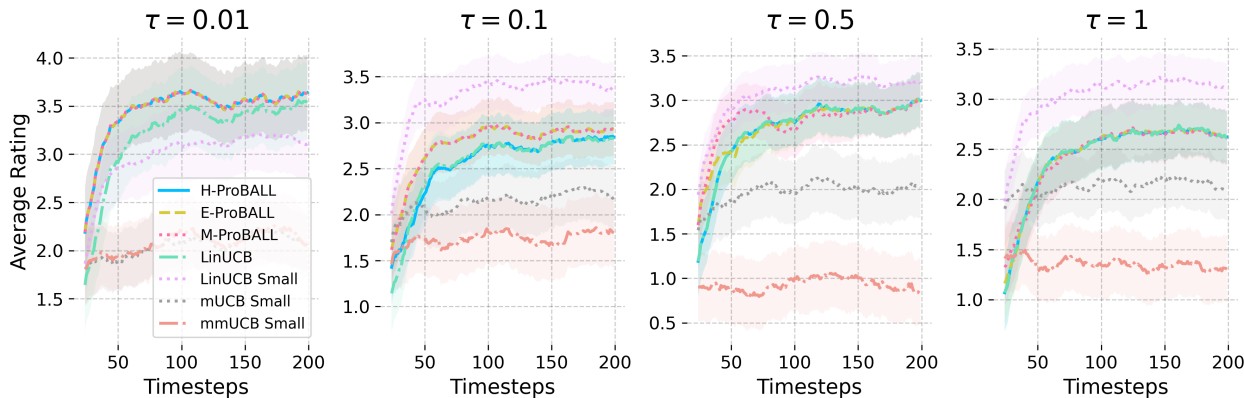

Figure 13: Comparison of ProBALL-UCB initialized with SOLD against LinUCB, mUCB, and mmUCB on low-dimensional ground-truth features, for different choices of $\tau$ and confidence bound constructions. When $\tau$ is small enough, all variants of ProBALL-UCB perform no worse than low-dimensional LinUCB, and outperform mUCB and mmUCB, on ground truth features. This showcases the efficacy of SOLD, and demonstrates that we recover subspaces that are just as good as ground-truth. Shaded area depicts 1-standard error confidence intervals over 30 trials with fresh $\theta$. The confidence intervals on regret thus account for the variation in frequentist regret for changing $\theta$.

### H.3.4. NO USAGE OF SOLD

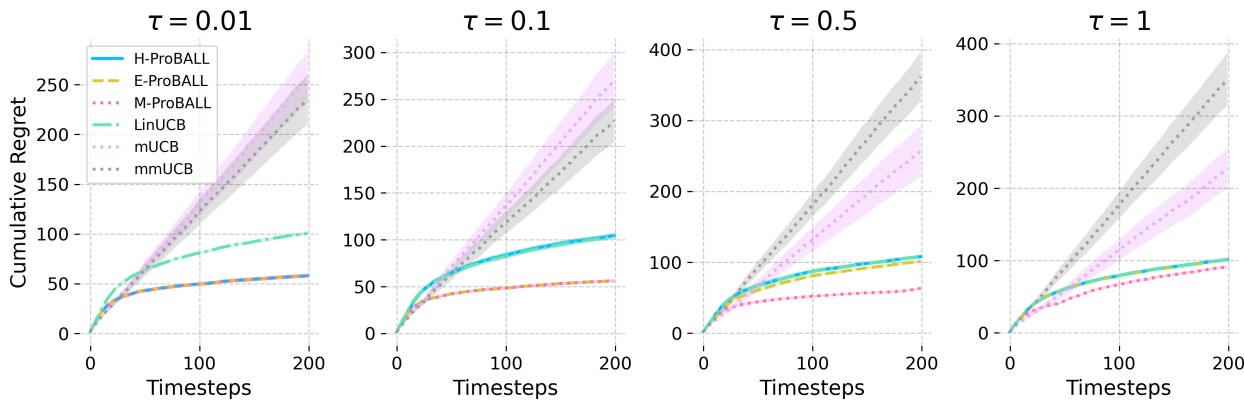

Figure 14: Comparison of ProBALL-UCB initialized with ground truth subspaces against LinUCB, mUCB, and mmUCB, for different choices of $\tau$ and confidence bound constructions. All variants of ProBALL-UCB perform no worse than LinUCB, and outperform mUCB and mmUCB. Shaded area depicts 1-standard error confidence intervals over 30 trials with fresh $\theta$. The confidence intervals on regret thus account for the variation in frequentist regret for changing $\theta$.

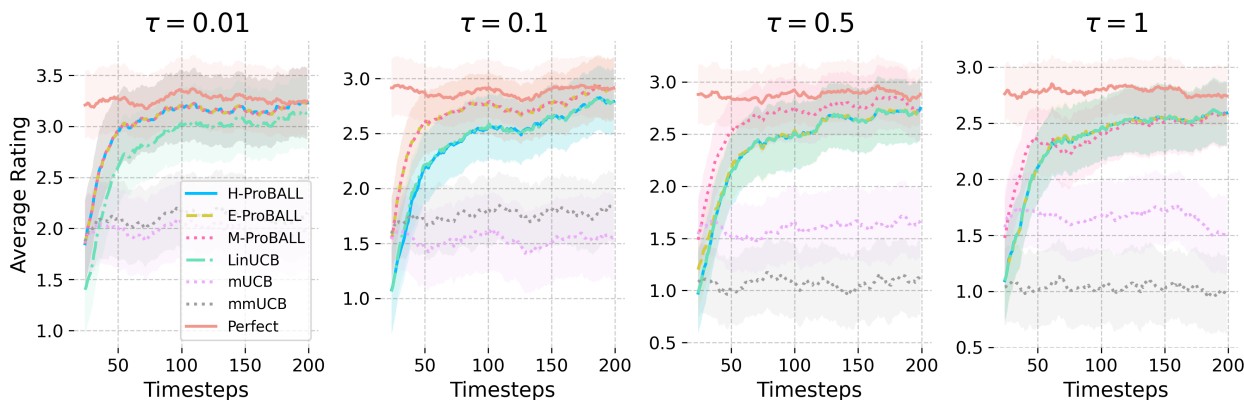

Figure 15: Comparison of ProBALL-UCB initialized with ground truth subspaces against LinUCB, mUCB, and mmUCB, for different choices of $\tau$ and confidence bound constructions, in terms of rolling average rating over 25 timesteps. All variants of ProBALL-UCB perform no worse than LinUCB, and outperform mUCB and mmUCB. Shaded area depicts 1-standard error confidence intervals over 30 trials with fresh $\theta$. The confidence intervals on regret thus account for the variation in frequentist regret for changing $\theta$.

## H.4. Sample Complexity of SOLD

We perform an empirical study of the sample complexity of SOLD on the MovieLens dataset. To do so, we compare the end-to-end regret at $T = 200$ timesteps of both ProBALL-UCB and ProBALL-TS, against LinUCB and Linear Thompson sampling using ground-truth low-dimensional features. When $\tau$ is small enough, we see that the end-to-end regret of both ProBALL-UCB and ProBALL-TS converges to that of LinUCB and Linear Thompson sampling using ground-truth low-dimensional features. This shows that we lose little from needing to estimate the subspace with SOLD when enough offline samples are present.

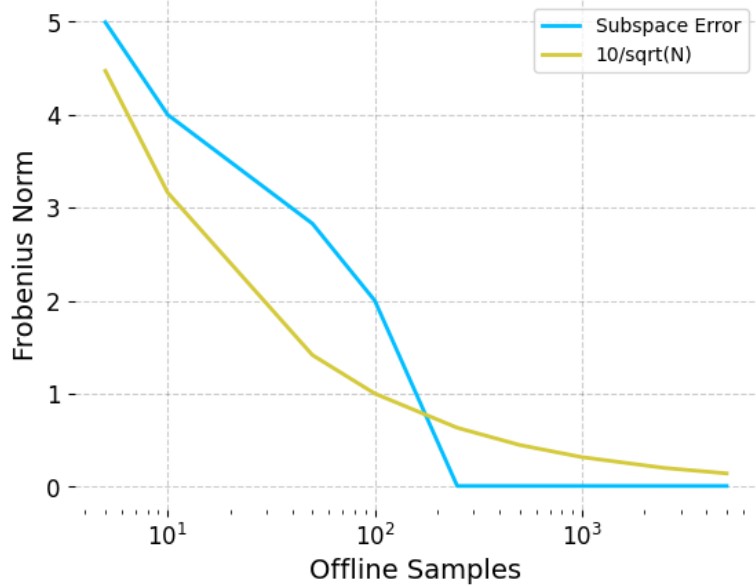

Figure 16: Subspace estimation error of SOLD against the number of offline samples, in the Frobenius norm. This was performed on the MovieLens dataset. We compare the error of SOLD against the parametric rate of $1/\sqrt{N}$. This shows that the error of SOLD indeed decreases very quickly in practice.

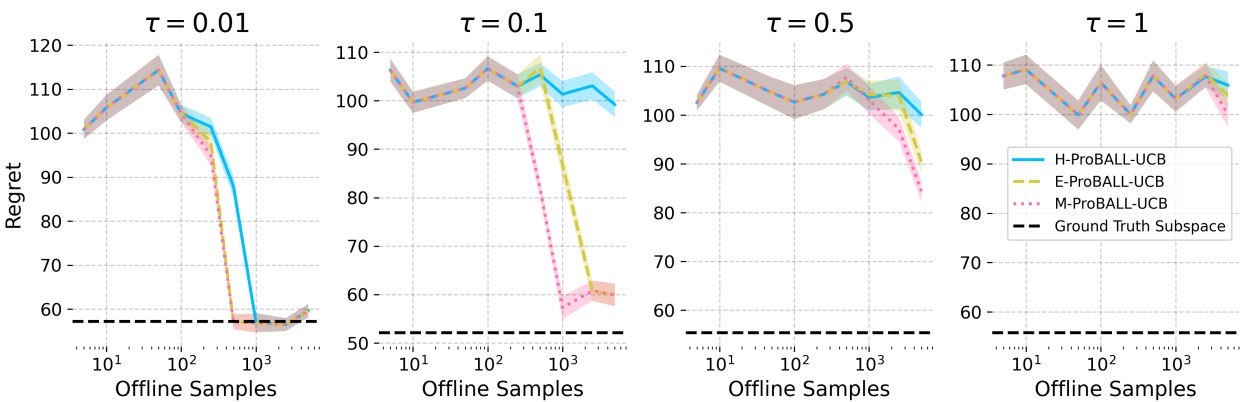

Figure 17: End-to-end regret at $T = 200$ timesteps of ProBALL-UCB initialized with SOLD, against the number of offline samples used in fitting SOLD. With a low enough $\tau$, the regret of ProBALL-UCB approaches the regret of LinUCB on ground-truth low-dimensional features, showing that we lose next to nothing from needing to estimate the subspace with SOLD. Shaded area depicts 1-standard error confidence intervals over 30 trials with fresh $\theta$.

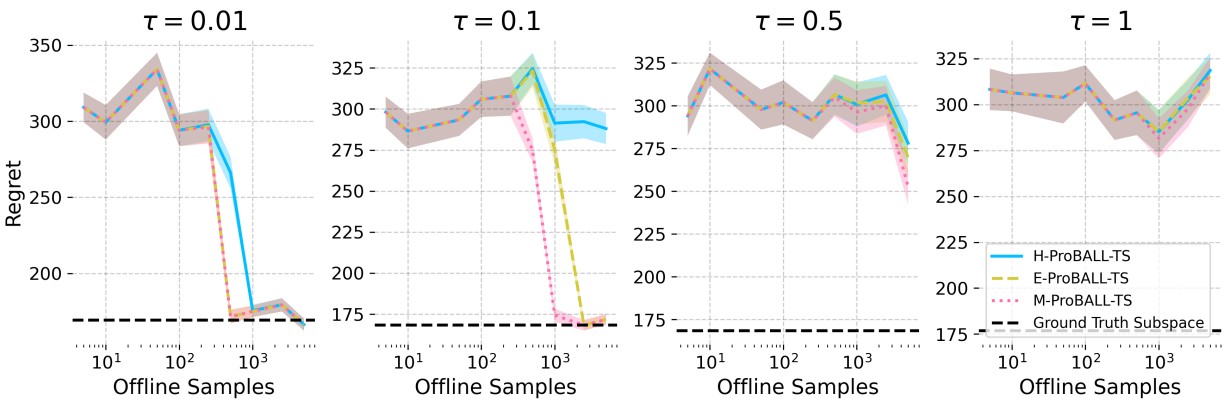

Figure 18: End-to-end regret at $T = 200$ timesteps of ProBALL-TS initialized with SOLD, against the number of offline samples used in fitting SOLD. With a low enough $\tau$, the regret of ProBALL-TS approaches the regret of TS on ground-truth low-dimensional features, showing that we lose next to nothing from needing to estimate the subspace with SOLD. Shaded area depicts 1-standard error confidence intervals over 30 trials with fresh $\theta$.

