# OpenReview forum: "Leveraging Offline Data in Linear Latent Contextual Bandits"
_ICML.cc/2025/Conference — ICML 2025 poster_

### Official Review · Reviewer_EkCv · 2025-02-25

**Overall Recommendation:** 5

**Summary:**

This paper introduces a linear version of latent bandit models, where the reward for user u and action a of feature $\phi_{u,a}$ at step t is $Y_{u,a} = \phi_{u,a}U\theta+\varepsilon_t$ where $\varepsilon_t$ is an iid subgaussian noise, U an unitary matrix and $\theta$ a low-dimensional latent vector. U and $\theta$ are unknown. Moreover, the authors assume the access to a prior offline set of N short trajectories to speed up online learning. The goal is to minimize the cumulative regret at fixed horizon T.
First, an offline algorithm called SOLD is introduced to learn a confidence set and an estimator of the latent subspace by estimating the matrix $UU^ T$ on the offline data. Second, a LinUB-inspired algorithm named LOCAL-UCB is proposed and uses arm indices leveraging the confidence set learned on the offline data and the iteratively updated confidence set on online data to get lower-bound matching guarantees on the regret, at the cost of computational tractability. Eventually, another online algorithm named ProBALL-UCB avoids solving the difficult optimization problem in LOCAL-UCB at the price of no longer matching regret upper bounds.
Experimental results are provided on several baselines and synthetic and real-life data sets.

## update after rebuttal
All my questions have been answered and I keep my very positive score on this paper.

**Claims And Evidence:**

To me, all claims (regret bounds, empirical results, conditions for stateless decision processes being a latent bandit) are backed by convincing evidence.

**Essential References Not Discussed:**

Lines 81-82 on page 2: “To the best of our knowledge, this is the first lower bound in a hybrid (offline-online) sequential decision-making setting”: a paper from 2023 [1] gives lower bounds on the cumulative regret for a structure type of latent bandits (with hidden clusters) where their algorithmic contribution leverages an offline matrix completion oracle during online learning. It is not exactly the same “hybrid” setting with offline collected trajectories and their lower bound does not take into account offline information, but, as it is a very similar setting, I would advise discussing (shortly) this paper and rephrasing the quoted sentence.

[1] Pal, S., Suggala, A. S., Shanmugam, K., & Jain, P. (2023, April). Optimal algorithms for latent bandits with cluster structure. In International Conference on Artificial Intelligence and Statistics (pp. 7540-7577). PMLR.

**Experimental Designs Or Analyses:**

See Methods and Evaluation Criteria.

**Methods And Evaluation Criteria:**

Experiments are performed both on synthetic and real-life data sets relevant to the recommendation task. The selected baselines (LinUCB for linear bandits without relying on offline, mTS and mUCB for latent bandits without the linear structure, different concentration bounds and hyperparameter values for ProBALL-UCB) are relevant.

**Other Comments Or Suggestions:**

None.

**Other Strengths And Weaknesses:**

Strengths
- The theoretical contributions are novel, strong and diverse (conditions for learning unconfounded estimators, regret lower bound, generality of latent bandits).
- The algorithmic contributions are novel and interesting (latent subspace and confidence set learning from offline data, lower-bound-matching algorithm, tractable counterpart). The latter can be extended to Bayesian approaches to regret minimization.
- The empirical results seem robust with an acceptable number of iterations (30), provide variations on hyperparameter values, and clearly show an improvement over the state-of-the-art in related settings, as the linear latent bandit was first introduced in this paper.
- The problem of cumulative regret minimization in structured latent bandits makes sense.
- The paper is well-written.
- The code is available, reusable (presence of a specific Python module) and reproducible (presence of notebooks for each data set).

Weaknesses
- I am surprised that LinUCB performs that well in Figure 2, second plot, as it does not rely on offline data. Could you explain this?

**Questions For Authors:**

See Weaknesses. This is a minor concern to me.

**Relation To Broader Scientific Literature:**

The idea of splitting trajectories to learn the latent subspace in the SOLD algorithm comes from a prior work [1]. Linearity is a popular structure in the bandit literature [2] and latent bandits with optimism have been investigated since at least 2014 [3].

[1]  Kausik, C., Tan, K., & Tewari, A. (2023, July). Learning mixtures of markov chains and mdps. In International Conference on Machine Learning (pp. 15970-16017). PMLR.

[2] Li, L., Chu, W., Langford, J., & Schapire, R. E. (2010, April). A contextual-bandit approach to personalized news article recommendation. In Proceedings of the 19th international conference on World wide web (pp. 661-670).

[3] Maillard, O. A., & Mannor, S. (2014, January). Latent Bandits. In International Conference on Machine Learning (pp. 136-144). PMLR.

**Theoretical Claims:**

I did not check the proofs in Appendix.

---

> ### Author Rebuttal · Authors · 2025-04-01
>
> We are very grateful to the reviewer for their kind and constructive comments. We are excited to hear that the reviewer highlights so many strengths of our work, including:
> 1. The strength of our theoretical contributions.
> 2. Our problem formulation.
> 3. Our novel and interesting algorithmic contributions.
> 4. The writing of the paper.
> 5. The reproducibility of our code (thank you for looking through the code!).
> 6. The robustness of our empirical results.
>
> Again, we appreciate your thorough review, and address your comments below.
>
> ## Essential References Not Discussed
> We were indeed not aware of the lower bound in [1] (Pal et al, 2023). As the reviewer remarks, there is a subtlety since this is also a purely online setting unlike our hybrid offline-online setting, although it relies on an offline matrix completion oracle. Nevertheless, we agree that this paper is relevant to ours and a remark should be added within our paper addressing it. We thank the reviewer for bringing this to our attention.
>
> ## Other Strengths and Weaknesses
> Note that in the second plot of Figure 2, LinUCB performs better than the algorithms mUCB and mmUCB from [2] (Hong et al, 2020), the same as H-ProBALL-UCB, and worse than E-ProBALL-UCB and M-ProBALL-UCB.
> 1. **Why LinUCB is better than mUCB and mmUCB:** The outperformance over mUCB and mmUCB makes sense, since these two algorithms only work with standard multi-arm bandits with finitely many latent states without a linear structure. LinUCB can leverage an approximately linear structure in the MovieLens data by working in a linear bandit setting. We know that given K arms in a linear bandit, standard UCB algorithms have Ksqrt{T} regret, while LinUCB has d\sqrt{T} regret, which can be lower if $d<K$. It is certainly still a bit surprising that even using offline data in mUCB and mmUCB is not enough to outperform LinUCB - a purely online algorithm that simply leverages a potential linear structure in rewards.
> 2. **Why LinUCB performs the same as H-ProBALL-UCB:** ProBALL-UCB performs best when we achieve tight subspace concentration in constructing our subspace confidence sets. Specifically, given loose confidence sets, ProBALL-UCB switches to LinUCB very early on. As the Hoeffding (H) confidence sets are not as tight as the empirical Bernstein (E) and martingale Bernstein (M) confidence sets, the performance of H-ProBALL-UCB accordingly is similar to that of LinUCB.
> 3. **Why LinUCB is worse than E-ProBALL-UCB and M-ProBALL-UCB:** As we mention above, ProBALL-UCB is able to leverage offline data better when we use tight subspace concentration bounds in constructing our subspace confidence sets. The empirical Bernstein (E) and martingale Bernstein (M) confidence sets are much tighter than Hoeffding confidence sets. So, ProBALL-UCB is able to work in the learnt low-dimensional subspace for long enough before having to switch to LinUCB.
>
> ## Refs
> 1. Pal et al, 2023. Optimal algorithms for latent bandits with cluster structure.
> 2. Hong et al, 2020. Latent Bandits Revisited.

---

> > ### Comment · Reviewer_EkCv · 2025-04-01
> >
> > Thank you for answering my question. I have no further comments and keep the score as it is.

---

### Official Review · Reviewer_D3RP · 2025-03-07

**Overall Recommendation:** 4

**Summary:**

This paper explores linear latent contextual bandit problems and how offline data can be used to speed up online learning. The authors introduce an offline algorithm that learns a low-dimensional latent subspace with provable guarantees. Building on this, they propose an online algorithm that achieves minimax-optimal regret, meaning its performance is as good as theoretically possible. They also present a more practical version of this algorithm that is computationally efficient but comes with a slightly weaker guarantee. Empirical results further support their theoretical findings, demonstrating the effectiveness of their approach.

**Claims And Evidence:**

The overall claims in the paper are clear and well-supported by both theoretical analysis and empirical results.

**Essential References Not Discussed:**

I think the paper covers the related work quite thoroughly.

**Experimental Designs Or Analyses:**

In Figure 2, the first plot (Simulation Study) shows that the regret curves of the proposed algorithm increase rapidly at certain points, with a noticeably steep slope. In particular, the red line has a steeper incline than LinUCB. This raises concerns about the practical effectiveness of the proposed methods for larger timesteps, as it is unclear whether they consistently outperform LinUCB in the long run.

**Methods And Evaluation Criteria:**

Overall, the  proposed methods make sense for the problem setting. However, one concern is that they assume the proposed algorithm (including in the empirical results) knows $d_K$, which is often difficult to determine in practice. The authors mention that $d_K$ can be estimated heuristically, but the empirical results do not include any experiments where $d_K$ is estimated rather than given. Including such results would have strengthened the evaluation and provided a clearer understanding of the method’s practical applicability.

**Other Comments Or Suggestions:**

No other comments.

**Other Strengths And Weaknesses:**

The lower bound analysis and empirical results clearly strengthen the paper, demonstrating both theoretical optimality and practical effectiveness of the proposed methods.

**Questions For Authors:**

1. Is the offline data generated from multiple different latent states, rather than just from the latent state $\theta^\star$? If so, could you explain this in more detail?

2. In the "simulation study" experiment, does your proposed algorithm outperform LinUCB when the number of timesteps is sufficiently large?

3. In the experiments, what happens if we estimate $d_K$ from the data instead of assuming it's known?

**Relation To Broader Scientific Literature:**

This paper expands the scope of linear bandits by incorporating latent states and hybrid offline-online learning settings. It introduces novel algorithms that leverage offline data to accelerate online decision-making while accounting for latent structures in user behavior or environments.

**Theoretical Claims:**

The theoretical claims are strong and well-reasoned. While I didn’t go through every detail of the proofs, they seem correct.

---

> ### Author Rebuttal · Authors · 2025-04-01
>
> Thank you for your review and confidence in our paper! We are grateful for your appreciation of:
> 1. The clarity of our claims and the strength of the theoretical and empirical support for them.
> 2. The expansion of the scope of linear bandits to include latent states and hybrid offline-online learning.
> 3. The novelty of our algorithms.
> 4. The thorough coverage of literature in our related work section.
> 5. The strengthening of the paper due to the lower bound analysis and the empirical results.
>
> We address your qualms below. **If our responses have adequately addressed your concerns, we would be delighted if you would consider raising your score.**
>
> ## Methods and Evaluation Criteria
>
> We are grateful for your appreciation of our proposed method and our heuristic for estimating $d_K$. We do in fact perform experiments for estimating $d_K$ in the Appendix, and we thank you for allowing us to reiterate our experiments involving this principled estimation method below.
>
> 1. **In practice, we don’t need to know $d_K$:** As you point out, in lines 206-209 (right column), we mention that one can use our theory to derive a principled estimate for $d_K$.
> 2. **Experiments estimating $d_K$ in Appendix H.1:** We direct the reviewer to Appendix H.1, where we provide experiments where we determine $d_K$ from offline data within the MovieLens experiments.
> 3. **Effect of $d_K$ estimation on downstream regret:** For your satisfaction, we also discuss the effect on downstream regret here. Overestimating $d_K$ is not a huge issue, as this simply leads to a slightly larger confidence set. Underestimating $d_K$, on the other hand, can lead to the learning of a misspecified subspace, and the regret bound then degenerates into the $d_A\sqrt{T}$ bound.
>
> We apologize for not explicitly mentioning appendix H.1 in the experiments section and will do so in the camera-ready version.
>
> ## Experimental Designs Or Analyses
>
> We appreciate your concern about the practical applicability of ProBALL-UCB, but we want to point out that we discuss in the paper how the experimental observations are consistent with our expectations and highlight the superiority of ProBALL-UCB over Lin-UCB.
>
> 1. As we discuss in lines 290-297 (left column) of the paper as well as show in the algorithm for ProBALL-UCB, ProBALL-UCB switches to standard Lin-UCB after a certain threshold is reached.
> 2. As we discuss in lines 366-368 (left column), the kinks or “rapid increases” that we see in Figure 2 correspond to switching to Lin-UCB once the aforementioned threshold is reached.
> 3. This means that the growths of these graphs after the rapid increases are _identical_ to Lin-UCB, not greater or lesser. An example of this can be found in Appendix H.2, Figure 5, in the right-most subfigure labeled $\tau=10$. Within this subfigure, ProBALL-UCB switches over to Lin-UCB earlier than in the illustration in Figure 2, with a similar initially steeper slope, but the regret of ProBALL-UCB is never worse than that of Lin-UCB.
>
> So, the warm-start provided by ProBALL-UCB makes it superior to vanilla Lin-UCB, and ProBALL-UCB is at least as good as Lin-UCB after the "rapid increase."
>
> ## Questions for Authors
> ### Question 1
> Indeed, as we mention in line 131 (left column) as well as in other parts of the paper, the offline data comes from multiple latent states. In fact, as encapsulated in Assumption 2, there have to be enough latent states in the offline data to cover the low-rank latent subspace.
> ### Question 2
> Yes, it does, as we clarify conceptually in our response to your qualm under “Experimental Designs or Analyses” above.
> ### Question 3
> As clarified in our response to your qualm under “Methods and Evaluation Criteria” above, we do in fact do so in Appendix H.1. We apologize for not mentioning this in the main paper, and we will do so in the camera-ready version.

---

### Official Review · Reviewer_qrXz · 2025-03-13

**Overall Recommendation:** 2

**Summary:**

In this paper, the authors study the linear latent contextual bandit problem. They consider a setting in which the latent reward vectors lie within a low-dimensional subspace. An offline dataset from tasks whose hidden reward vectors share the same subspace is assumed to be available. They first present an algorithm that estimates the subspace, which is proven to recover the subspace with a bounded error. Then, using the subspace estimation algorithm, they show that, under the assumption that the distribution of latent reward vectors spans the space, the algorithm can provably utilize the offline dataset to achieve a better regret bound. A complementary regret lower bound is then presented to show the near-optimality of the algorithm. As the previous algorithm is not computationally efficient, they present an algorithm that approximates the confidence set with a corresponding regret bound. Finally, experiments on both synthesized and real data are presented to demonstrate the practical performance of their algorithm.

**Claims And Evidence:**

The major problem is that the regret bound presented in Theorem 2 is not minimax optimal, as the regret upper bound depends on the coverage factors $\lambda_\theta$ and $\lambda_A$. In contrast, the lower bound presented in Theorem 3 has no dependency on either $\lambda_\theta$ or $\lambda_A$. As a result, in the case where the feature vector or the hidden latent reward vector does not span the whole space, the algorithm is not minimax optimal and has no advantage compared to the vanilla LinUCB algorithm for contextual bandits.

**Essential References Not Discussed:**

They have cited all relevant papers to my knowledge.

**Experimental Designs Or Analyses:**

They apply their algorithm to both synthesized and real data.

**Methods And Evaluation Criteria:**

N/A

**Other Comments Or Suggestions:**

See above

**Other Strengths And Weaknesses:**

Although ProBALL-UCB is presented as an approximation of the LOCAL-UCB algorithm to ensure computational efficiency and has been shown to leverage offline data in experiments, it is not provably established that the algorithm achieves a better regret bound in general. Specifically, since $\hat{U}$ is determined by the algorithm routine, it is unclear in which situations $\phi(x_t, a_t)$ lies in the span of $\hat{U}$. It would be valuable to formally establish the cases in which ProBALL-UCB can effectively leverage offline data.

**Questions For Authors:**

- Do you think the data coverage assumption (i.e., Assumption 2) is necessary? In comparison, it has been shown that in multi-task linear bandits, this assumption is not required to obtain a provably improved regret bound by leveraging data from other tasks (see, e.g., Yang et al., 2022).

Yang, Jiaqi, et al. "Nearly minimax algorithms for linear bandits with shared representation." arXiv preprint arXiv:2203.15664 (2022).

**Relation To Broader Scientific Literature:**

This paper extends the understanding of leveraging offline data in bandits problem, showing that the offline data can provably improve the performance of the algorithm.

**Theoretical Claims:**

All the claims are clear and proved.

---

> ### Author Rebuttal · Authors · 2025-04-01
>
> Thank you for your comments. We appreciate that you recognize:
>
> 1. The clarity and veracity of our theorems and proofs.
> 2. The application of our algorithms to both synthetic and real data.
> 3. Our contribution to the problem of leveraging offline data in bandits.
>
> We address your qualms below. **If our responses have adequately addressed your concerns, we implore you to consider raising your score.**
>
> ## Claims and Evidence
>
> While we agree there is subtlety in its nature compared to typical purely online bounds, we maintain that our regret bound is optimal. We appreciate the opportunity to clarify this and will include the discussion in the camera-ready appendix.
>
> 1. **Disappearance of $\lambda_\theta$ and $\lambda_A$ in the worst-case bound:** As you would agree, minimax lower bounds are worst-case bounds that optimize over “all problem instances.” So, instance-specific parameters like $\lambda_\theta$ and $\lambda_A$ naturally disappear in the worst-case expression. This is standard in lower bounds for bandits. However, it is crucial to carefully choose the space of “all problem instances,” especially in our offline+online setting.
> 2. **The insufficient offline coverage case is trivial, we need a more informative lower bound:** The key challenge is in selecting an instance space that yields an informative lower bound.
>
>     a. **Typical and trivial choice of problem-instance space:** In online linear bandits, it’s common to vary all parameters after fixing the dimension and bounding the reward parameter $\beta$. Applying that here yields a trivial $d_A\sqrt{T}$ bound, achieved when offline data has insufficient coverage (when offline data "does not span the whole space," as you said). Many algorithms (including ours) attain this, showing minimax optimality but not any benefit over Lin-UCB.
>
>     b. **Our more informative choice of problem-instance space:** To demonstrate a real advantage over Lin-UCB, we constrain the instance space further - we fix the offline data quality by assuming $\lambda_\theta$ is bounded below. This models scenarios with sufficient offline coverage, and our lower bound shows that even in these non-trivial settings, no algorithm can outperform ours. Further we establish a clear advantage over the $d_A\sqrt{T}$ bound for Lin-UCB.
> 3. **Potential for more instance-dependent lower bounds:** While we analyze worst-case performance over a meaningful class, there’s room for future work on sharper, instance-dependent lower bounds that reflect explicit dependence on both $\lambda_\theta$ and $\lambda_A$. However, we believe that our introduction of a nontrivial lower bound in this hybrid offline+online setting is already a significant step.
>
> ## Other Strengths and Weaknesses
>
> It is unclear what you mean by “better regret bound in general.” You might be comparing to either LOCAL-UCB or Lin-UCB - we address each of the two below:
> 1. **ProBALL-UCB vs LOCAL-UCB:** We explicitly state that ProBALL-UCB has weaker guarantees than LOCAL-UCB in general (lines 86 and 303). However, as noted in lines 303–310 and proven in Appendix E.2.1, ProBALL-UCB matches LOCAL-UCB in “good” cases, like when the feature set is an $\ell_2$ ball, since then $\phi(x_t, a_t)$ lies in the span of $\hat{\mathbf{U}}$. Such an assumption is standard in the literature, including [1] (Yang et al, 2022), which you referenced. We acknowledge not citing Appendix E.2.1 in the main text and will correct that in the camera-ready version.
> 2. **ProBALL-UCB vs Lin-UCB:** ProBALL-UCB is better than LinUCB, as we can see from Theorem 4. In fact, ProBALL can improve significantly on Lin-UCB, as we see both from Theorem 4 and the experiments.
>
> ## Questions for Authors
>
> ### Question 1
> Yes, this assumption is essential—due to a key difference between our setting and that of [1] (Yang et al, 2022).
> 1. **[1] is purely online:** In [1], the setting is purely online and the learner chooses actions in each of M concurrent bandit instances. This allows for coordinated exploration across tasks.
> 2. **Our “multi-task” dataset is collected offline:** We work with an offline dataset of trajectories spanning multiple bandit instances. The learner has no control over the behavior policy that collected the data. Without structural assumptions on the dataset, estimating a useful subspace becomes infeasible, and regret degenerates to the standard $d_A\sqrt{T}$.
>
> Coverage assumptions similar to ours are commonplace within the offline linear MDP literature, like in [2,3], and they all take the form of concentrability-type assumptions [4] within the broader offline RL literature proper.
>
> ## Refs
> 1. Yang et al. (2022), Nearly Minimax Algorithms for Linear Bandits with Shared Representation
> 2. Jin et al. (2021), Is Pessimism Provably Efficient for Offline RL?
> 3. Duan et al. (2020), Minimax-optimal off-policy evaluation with linear function approximation
> 4. Zhan et al. (2022), Offline Reinforcement Learning with Realizability and Single-policy Concentrability

---

### Official Review · Reviewer_iZXM · 2025-03-13

**Overall Recommendation:** 5

**Summary:**

This paper studies the setting of _linear latent contextual bandits_. If you are given multiple trajectory data under some unknown behavior policy, with possibly different latent states for each trajectory, how do you efficiently use it in an online setting? This paper proposes three algorithms and their analysis — an algorithm for estimating the linear subspace for the latent variables, a minimax optimal (under expected regret) online algorithm, and a computationally efficient algorithm that is almost optimal under some settings. Finally, the paper shows the generality of latent bandits by defining a notion of exchangeable and coherent stateless decision processes and showing that every such process is a latent bandit.

**Claims And Evidence:**

The paper is exceptionally clear and thorough. Almost every claim is convincingly supported. It also does an excellent job conveying the intuition behind the definitions, algorithms and the proofs.

**Essential References Not Discussed:**

Nothing comes to mind here.

**Experimental Designs Or Analyses:**

I checked the appendix detailing the experimental results and they look good to me.

**Methods And Evaluation Criteria:**

This is a paper of theoretical nature, but does a great job of demonstrating the practical utility of the proposed algorithms by doing some experiments demonstrating their efficiency against reasonable baselines.

**Other Comments Or Suggestions:**

There is a small typo when defining $\overline{\mathbf{D}}_{N,i}$ in the additional notation section.

**Other Strengths And Weaknesses:**

As I said before in the review this is a very nicely written piece of work. As a reviewer it is always appreciated to review a paper from which I learn a lot.

I really like the SOLD algorithm. The idea of trajectory splitting is very cool!

**Questions For Authors:**

1. Why is the map $U_\star$ assumed linear?
2. I did not understand the argument why $U_\star$ can be assumed to be orthogonal without loss of generality. It may not even be a square matrix! A change of basis enforced by $A^{-1}$ does not necessarily makes $U_\star$ orthogonal.
3. On line 123, it is stated that "permuting the labels and rewards". But $A$ is permuting latent states.
4. From the discussion in section 8, I am unable to see how a latent bandit is an SDP. The definition of latent bandit has an extra measure-valued function $F$ which an SDP doesn't have.

**Relation To Broader Scientific Literature:**

The paper studies a more general setting of latent bandits introduced by Hong et al. (2020).

**Theoretical Claims:**

No, I cannot attest to checking the proofs carefully. I only skimmed some of the proofs in the appendix. However, what I read made sense and it seems the authors have given full proofs of everything.

---

> ### Author Rebuttal · Authors · 2025-04-01
>
> We are grateful for your review and your confidence in our paper! We appreciate that you enjoyed our:
> 1. Clarity and thoroughness of evidence.
> 2. Presentation of intuition and writing quality.
> 3. Demonstration of practical utility through experiments.
> 4. Generality over existing work in leveraging offline data for bandits.
> 5. Technical ideas behind SOLD.
>
> We acknowledge the typo and will address it in the camera ready version!
>
> ## Questions
> ### Question 1
> Linearity is a standard structural assumption in bandits, and it is not unreasonable that a continuous latent state could have a linear effect on the reward parameters of a bandit instance. This neatly generalizes the tabular assumption that each latent state comes with a specific reward parameter while allowing us to tackle continuous latent states. We demonstrate the practical relevance of this assumption by evaluating our algorithms on real-life MovieLens data.
>
> However, we agree that more complex relations between the latent state and reward parameters are possible, and we leave the general function approximation case to future work.
>
> ### Question 2
> This seems to be a misunderstanding in choice of terminology, and we are glad you brought it up. $U_\star$ is indeed not a square matrix at all! By virtue of our setting, it is a $d_K \times d_A$ matrix, and so in fact has very skewed dimensions.
>
> By orthogonal, we simply mean that the columns of $U_\star$ are orthonormal, not the rows. That is, $U_\star^\top U_\star$ is the $d_A$-dimensional identity matrix, but indeed, $U_\star U_\star^\top$ is almost never the $d_K$-dimensional identity matrix. Naturally, we only rely on the first fact (that $U_\star^\top U_\star = I$) in our proofs, which does hold WLOG.
>
> We will clarify our language in the camera ready version.
>
> ### Question 3
> By "permuting labels and rewards together," we mean permuting the latent labels so that the reward trajectories assigned to a given label stay together. This is the same as permuting latent states.
>
> We recognize the confusion this language can create and will clarify by simply saying "just like in the case of finitely many latent states, observations are not changed by permuting the latent states. That is, observations are not changed by permuting latent trajectory labels while keeping trajectories with the same label together."
>
> ### Question 4
> The latent bandit is indeed a special case of an SDP, where the function $\mathcal{F}_H$ is induced by the latent state random variable $F$.
>
> Specifically, as we state in the definition of a latent bandit, a latent bandit is an SDP where the function $\mathcal{F}_H(a_1, \dots a_H) = Y_1, \dots Y_H$ for SDPs is defined by drawing $Y_1 \dots Y_H$ independently conditioned on $F$ according to the distributions $Y_h \sim F(a_H)$. So, an SDP does not have a latent state $F$, but a latent bandit is a special case of an SDP where the functions $\mathcal{F}_H$ are induced by the latent state $F$ associated with the latent bandit.
>
> Just to help clarify this point, special cases of a general object can and usually do have extra structure. So, a latent bandit has the extra structure $F$ on top of the SDP functions $\mathcal{F}_H$ induced by $F$, just like how a linear bandit comes with the extra feature map $\phi$ and reward parameter $\beta$ on top of the general bandit reward function $r$ induced by $\phi$ and $\beta$.

---

> > ### Comment · Reviewer_iZXM · 2025-04-02
> >
> > Thank you for the response. It is helpful.
> > I will keep my acceptance score.

---

### Decision · Program_Chairs · 2025-05-01

**Decision:**

Accept (poster)

**Comment:**

This paper proposes two contextual bandit algorithms for latent bandits. Both algorithms have the following structure: estimate the latent subspace offline from previously logged data and then act with respect to it online. The first algorithm is impractical but has a lower regret. The second algorithm is practical but has a higher regret. Both algorithms are analyzed and the practical one is also empirically evaluated. The initial scores of the paper were 2x Strong Accept, Accept, and Weak Reject, which reflects its quality. Congratulations! Please take the feedback of the reviewers into account when updating the paper.